# *Mixture-of-Visual-Thoughts*: Exploring Context-Adaptive Reasoning Mode Selection for General Visual Reasoning

**Zejun Li**[1,2,3†‡]**, Yingxiu Zhao**[2,3†]**, Jiwen Zhang**[1]**, Siyuan Wang**[4]**, Yang Yao**[2,3]
**Runzhou Zhao**[2,3]**, Jun Song**[2,3*]**, Bo Zheng**[2,3]**, Zhongyu Wei**[1,5*]
[1]Fudan University    [2]Alibaba Group Holding Limited    [3]Future Living Lab of Alibaba
[4]University of Southern California    [5]Shanghai Innovation Institute
{zejunli20,zywei}@fudan.edu.cn    jsong.sj@alibaba-inc.com
https://github.com/Future-Living-Lab/mixture-of-visual-thoughts

## Abstract

Current visual reasoning methods mainly focus on exploring specific reasoning modes. Although improvements can be achieved in particular domains, they struggle to develop general reasoning capabilities. Inspired by this, we propose a novel adaptive reasoning paradigm, Mixture-of-Visual-Thoughts (**MoVT**), which unifies different reasoning modes within a single model and guides it to select the appropriate mode based on the context. To achieve this, we introduce **AdaVaR**, a two-stage Adaptive Visual Reasoning learning framework: different modes are unified and learned during the supervised cold-start stage, and the mode selection capability is induced via an RL process with a carefully designed AdaGRPO algorithm. Extensive experiments show that AdaVaR effectively guides the model to learn and differentiate multiple modes and perform context-adaptive mode selection, achieving consistent improvement across various scenarios, highlighting MoVT as an effective solution for building general visual reasoning models.

## 1 Introduction

Recent research has revealed the reasoning potential of large language models (LLMs): by guiding LLMs to generate chain-of-thought (CoT) rationales in various forms before providing an answer, their performance on complex problems can be substantially improved (Kojima et al., 2022; Wei et al., 2022; Wang et al., 2023; Yao et al., 2024b). Furthermore, the reasoning capability of LLMs can be further enhanced through reinforcement learning (RL), ultimately pushing the frontier of model capability (Guo et al., 2025; Kimi-Team et al., 2025; Yu et al., 2025).

Motivated by these advancements, preliminary explorations of reasoning in multi-modal contexts have emerged. As illustrated in Figure 1a, existing visual reasoning methods fall into two types based on the form of CoTs (referred to as "reasoning mode" in this paper): (1) Text-based reasoning, which is consistent with LLMs, where large vision-language models (LVLMs) represent the reasoning process directly in natural language (Yang et al., 2025; Huang et al., 2025; Meng et al., 2025); (2) Visually-grounded reasoning, which anchors the reasoning process in the visual context. Typically, this involves guiding LVLMs to generate structured outputs (e.g., bounding boxes or points) that align textual concepts in the reasoning process with specific image regions (Chen et al., 2023; Lei et al., 2024; Fan et al., 2025). Additionally, several works propose to introduce associated local visual information into the generated sequence, enabling the model to focus on specific regions for subsequent inference (Li et al., 2025; Shao et al., 2024a; Zheng et al., 2025; OpenAI et al., 2025).

Different reasoning modes impose distinct inductive biases, yielding varying strengths and weaknesses across domains. As shown in Figure 1b, text-based models excel at abstract reasoning (e.g., mathematics) but are more prone to hallucinations due to overthinking and language bias. In contrast, visually-grounded models are better at leveraging visual information, curbing hallucination,

---

[†]Equal contribution; [‡] Work done during the internship at Future Living Lab of Alibaba.
[*]Corresponding authors.

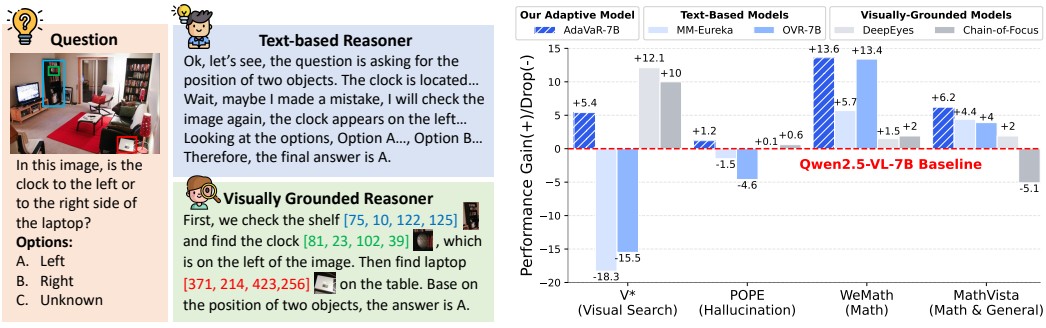

(a) Rollout examples of two reasoning modes.  (b) Performance of different reasoning modes.

Figure 1: Comparison between two reasoning modes. (b) Performance gains (positive values) and drops (negative values) of Qwen2.5-VL-7B–based reasoning models relative to the base model.

and handling problems with clear object information, yet show limited gains on mathematical benchmarks (since abstract concepts like length, size, etc. cannot be grounded to provide useful information). Overall, no single mode dominates across all tasks. This raises a critical question: can we integrate the complementary strengths of different modes to build a general visual reasoning model?

In light of this, we propose a novel visual reasoning paradigm, Mixture-of-Visual-Thoughts: the model is able to reason in different modes and adaptively choose the appropriate one based on the context. However, the greatest challenges lie in (i) how to uniformly represent different reasoning modes and enable a unified model to learn them; and (ii) how to develop the capability for context-adaptive mode selection. To tackle these challenges, we introduce **AdaVaR**, an Adaptive Visual Reasoning framework with a two-stage training paradigm: a supervised cold-start learning of different modes, and an adaptive RL to allow LVLMs to explore and acquire mode selection skills.

Specifically, AdaVaR begins with a cold-start stage to unify multiple reasoning modes within a model. We define a uniform sequence format for the reasoning paths of different modes and distinguish them with mode-specific prefix tokens. This allows us to mix reasoning data across modes and train the model through a unified supervised fine-tuning (SFT) stage. Although different reasoning modes can be effectively learned, guiding the selection of appropriate modes via SFT remains challenging, since we cannot pre-estimate the relative performance of different modes for each problem.

Therefore, to address the second challenge, a subsequent RL stage is introduced to induce context-adaptive mode selection skills while enhancing the reasoning abilities. The idea is straightforward: we encourage the model to explore different modes for each question and reward the mode that is more likely to yield correct answers. Nonetheless, standard GRPO (Shao et al., 2024b) poses obstacles to achieving this goal. First, the policy model may under-explore reasoning modes without explicit guidance (e.g., producing rollouts from only one mode per question). Second, the rollout-level advantage in GRPO fails to explicitly capture preferences among different modes. To address these issues, we propose AdaGRPO, which steers exploration and comparison between reasoning modes through three key components: (1) Prefix-guided mode exploration: by fixing the mode prefix, we compel the policy model to explore evenly across multiple modes for the same sample. (2) Adaptive advantage mechanism: besides the rollout-level advantage, we introduce a mode-relative advantage to explicitly guide the model towards optimal mode selection. (3) Curriculum-based data scheduling: we initiate training with easier data to learn coarse-grained distinctions between modes, then gradually transition to harder questions for fine-grained mode selection capability.

Extensive experiments demonstrate that AdaVaR enables the learning of different reasoning modes and context-adaptive mode selection capabilities. As shown in Figure 1b, our AdaVaR-7B achieves consistent improvements across scenarios. In terms of average performance over 8 various benchmarks, AdaVaR-3B matches Qwen2.5-VL-7B, and AdaVaR-7B surpasses GPT-4o. Our code, models, and data will be open-sourced to the community. The contributions of this work are threefold:

- Unlike existing works that focus on specific reasoning modes, we propose a novel visual reasoning paradigm, MoVT: unifying different modes within a model and guiding it to select the appropriate one based on context, pushing the frontier of general visual reasoning.

- We introduce AdaVaR, a two-stage adaptive reasoning training framework: different modes are unified and learned during the supervised cold-start stage, and adaptive selection of appropriate modes is learned during the RL process using the proposed AdaGRPO algorithm.

- Extensive experiments demonstrate the effectiveness of AdaVaR in guiding LVLMs to learn and distinguish different reasoning modes in a unified manner, and to dynamically select modes based on context. Our adaptive reasoning method brings general improvements across various scenarios and exhibits considerable potential for future exploration.

## 2    RELATED WORKS

### 2.1    LANGUAGE REASONING MODELS

Early work reveals the reasoning potential of LLMs, well-designed prompts (Kojima et al., 2022) and in-context samples (Wei et al., 2022) can be adopted to guide LLMs to generate CoTs, enhancing the ability to solve complex questions. Subsequent research focuses on designing complex forms of CoT (Yao et al., 2024b; Besta et al., 2024) and introducing mechanisms like majority voting (Wang et al., 2023), reflection (Madaan et al., 2023; Shinn et al., 2023), and step-level search (Lample et al., 2022; Xie et al., 2023; Tian et al., 2024). Recently, works represented by DeepSeek-R1 (Guo et al., 2025) have demonstrated that the reasoning abilities of LLMs can be effectively incentivized and expanded under a highly scalable RL environment (Qwen Team, 2025; Yu et al., 2025).

### 2.2    VISION-LANGUAGE REASONING MODELS

**Text-Based Reasoning Models**    Mainstream visual reasoning models directly express thinking processes in natural language. Early methods first learn reasoning patterns via SFT, building datasets through complex distillation (Xu et al., 2024; Thawakar et al., 2025) or collective search frameworks (Yao et al., 2024a), and then enhance the reasoning capability with search-based mechanisms (Xu et al., 2024; Thawakar et al., 2025) or DPO (Zhang et al., 2024b; Dong et al., 2025). Recent works mainly follow DeepSeek-R1, using high-quality data to compute verifiable rewards and employing GRPO to incentivize the reasoning abilities (Liu et al., 2025; Ma et al., 2025; Peng et al., 2025; Meng et al., 2025; Xu et al., 2025). Meanwhile, A cold-start SFT phase before RL, trained on high-quality data, can bootstrap the model's reasoning and enable more effective exploration during RL (Yang et al., 2025; Huang et al., 2025; Wei et al., 2025; Chen et al., 2025a).

**Visually-Grounded Reasoning Models**    Another line of research proposes to explicitly ground the reasoning process onto images. The prevailing approach is to introduce bounding box coordinates in CoTs, either through prompting (Mitra et al., 2024) or SFT (Chen et al., 2023; Wu et al., 2025c). Furthermore, the located regions can be cropped, zoomed, and then fed back into the model to assist it in referring to fine-grained visual information for subsequent reasoning (Li et al., 2025; Shao et al., 2024a; Shen et al., 2024; OpenAI et al., 2025). In addition, tool-based (Zhou et al., 2024) and visual prompting (Lei et al., 2024; Yang et al., 2023) methods can be used to supplement images with coordinate systems or segmentation maps, which can be utilized by LVLMs for explicit grounding. Recent works further demonstrate that grounded reasoning abilities can be activated and enhanced through RL by designing rewards (Sarch et al., 2025; Fan et al., 2025) and methods to guide grounded rollout generation (Cao et al., 2025; Zheng et al., 2025).

Existing models typically focus on one of the above modes and a narrow domain: text-based reasoning targets mathematical problems, while the visually-grounded mode is mainly used for object-centric tasks. Inspired by this, we propose an adaptive reasoning paradigm, MoVT, that integrates the complementary strengths of different modes to build a general-purpose visual reasoning model.

## 3    ADAVAR: AN ADAPTIVE VISUAL REASONING FRAMEWORK

This section details how the proposed AdaVaR framework endows LVLMs with adaptive reasoning abilities. First, to integrate multiple reasoning modes in a unified model, we introduce a reasoning paradigm in § 3.1 to represent reasoning paths from different modes in a unified manner, merging mode selection and reasoning into a single autoregressive generation process. Building on this, the

model undergoes an SFT phase to learn multiple reasoning modes simultaneously (as described in § 3.2). Based on the SFT model, we design an RL algorithm (as detailed in § 3.3) to help the model acquire the ability to select the appropriate mode according to the provided context.

## 3.1 REASONING MODE DEFINITION AND UNIFICATION

As introduced in the introduction, we regard different forms of CoT as distinct reasoning modes. In this paper, we mainly consider the integration of two commonly applied visual reasoning modes: (i) **Text-based reasoning**, without introducing additional inductive bias, represents all reasoning processes only in language; (ii) **Visually-grounded reasoning** requires models to generate structural information, such as coordinates, to align key concepts–primarily objects–with corresponding regions in the image during reasoning. In this work, we adopt the widely used bounding-box-enhanced format (Li et al., 2025; Fan et al., 2025), guiding the model to describe objects in the form of "object $[x_1, y_1, x_2, y_2]$"* to explicitly align the reasoning process with the image.

**Reasoning Mode Unification** To enable the model to learn and distinguish different modes within a unified framework, we define a uniform format of reasoning sequences: each mode is assigned a unique prefix token, which is placed at the beginning of reasoning paths to serve as an in-context indicator, guiding the model to differentiate between modes. We define the reasoning template as:

> **System**: You are a helpful assistant. You need to first think about the reasoning process and then answer the question raised by the user, in the format of <think> reasoning process here </think> <answer> answer here </answer>. You have two modes of thinking:
> 1. *Grounded Reasoning*: ..., begin your response with <ground> when using this mode.
> 2. *Text-based Reasoning*: ..., begin your response with <text> when using this mode.
> **User**: prompt. **Assistant**: <mode prefix> <think> ... </think> <answer> ... </answer>

The system prompt specifies the reasoning requirements and introduces the different reasoning modes (omitted for brevity; the full prompt is provided in Appendix B.1). The model response includes the mode prefix and the reasoning process. Leveraging the autoregressive property, generating reasoning paths in this format can be naturally divided into two steps, i.e., $P(a, t, m|i, q) = P(m|i, q) \times P(a, t|m, i, q)$: (1) $P(m|i, q)$ selecting the reasoning mode by generating the mode prefix $m$ based on the input image $i$ and question $q$, and (2) $P(a, t|m, i, q)$ generating the corresponding thinking process $t$ and answer $a$ based on the selected mode $m$ and the input context $i, q$. These two steps can be accomplished sequentially within a single sequence generation process.

## 3.2 STAGE 1: COLD-START MODE LEARNING WITH SFT

Building on the unified reasoning format, we adopt an SFT stage as a cold start to enable the base model, Qwen2.5-VL (Bai et al., 2025), to learn reasoning abilities across different modes simultaneously. To this end, we mix expert reasoning trajectories from different modes: (i) For text-based reasoning, we follow DeepSeek-R1 (Guo et al., 2025) to construct data by distilling a text-based visual reasoning model (Ma et al., 2025) and applying rejection sampling. (ii) Regarding visually-grounded reasoning, we directly utilize high-quality SFT data constructed in existing works (Li et al., 2025). The proportion of the two parts of data is controlled to 1:1 to prevent introducing undesired bias on mode selection. Please refer to Appendix B.2 for more details.

## 3.3 STAGE 2: ADAPTIVE MODE SELECTION WITH RL

Although Stage 1 helps the model learn reasoning abilities in different modes, it is difficult to learn mode selection under the supervised learning paradigm because we cannot pre-construct effective supervision on which mode to select. Moreover, recent studies have shown that the reasoning abilities acquired during the SFT stage can be further enhanced through RL (Yang et al., 2025; Wei et al., 2025). Therefore, we introduce an RL stage, which simultaneously improves the reasoning ability and guides the model to learn adaptive mode selection based on the proposed AdaGRPO algorithm.

---

*Absolute coordinates of the top-left and bottom-right corners, which are aligned with Qwen2.5-VL.

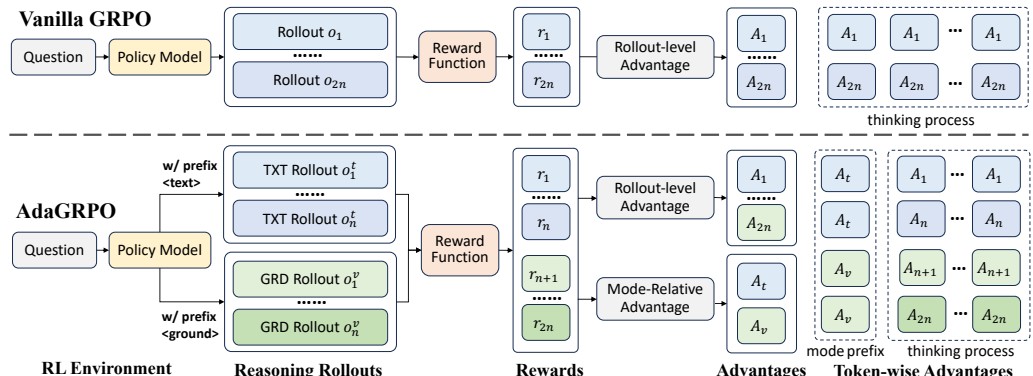

Figure 2: Demonstration of GRPO and AdaGRPO. In AdaGRPO, we use mode prefixes to guide exploration across different modes (TXT and GRD respectively represent text-based and visually-grounded reasoning), calculate both rollout-level advantages $\{A_j\}_{j=1}^{2n}$ and mode-relative preference $\{A_t, A_v\}$, and explicitly guide mode selection and the enhancement of reasoning abilities.

### 3.3.1 RL ALGORITHM: ADAGRPO

**Revisiting GRPO** For a better illustration on the design rationale of AdaGRPO, we review the details of GRPO (Shao et al., 2024b). As shown in the upper part of Figure 2, the old policy model $\pi_{\theta_{old}}$ generates $2n$ reasoning rollouts $\{o_1, o_2, ..., o_{2n}\}$ for question $q$, obtaining rewards $\{r_1, r_2, ..., r_{2n}\}$ which are utilized to optimize the current policy model $\pi_\theta$ with the following objective:

$$\mathcal{J}_{GRPO}(\theta) = \mathbb{E}\left[q \sim P(Q), \{o_j\}_{j=1}^{2n} \sim \pi_{\theta_{old}}(O|q)\right]$$

$$\frac{1}{2n}\sum_{j=1}^{2n}\frac{1}{|o_j|}\sum_{t=1}^{|o_j|}\left(\min\left[\frac{\pi_\theta(o_j|q, o_{j,<t})}{\pi_{\theta_{old}}(o_j|q, o_{j,<t})}A_{j,t}, \text{clip}\left(\frac{\pi_\theta(o_j|q, o_{j,<t})}{\pi_{\theta_{old}}(o_j|q, o_{j,<t})}, 1-\epsilon, 1+\epsilon\right)A_{j,t}\right]\right) \quad (1)$$

where $\epsilon$ is the clipping hyper-parameter and the KL penalty is omitted for brevity. $A_{j,t} = A_j = \frac{r_j - \text{mean}(\{r_1,...,r_{2n}\})}{\text{std}(\{r_1,...,r_{2n}\})}$ are the rollout-level advantages estimated by comparing rewards of all rollouts.

Although effective, the vanilla GRPO is limited in the adaptive reasoning scenarios in terms of: (i) the policy model, especially after SFT, may exhibit preferences for certain reasoning modes. As a result, the generated $2n$ rollouts could come from the same mode, leading to uneven exploration on different reasoning modes; (ii) GRPO merely considers rollout-wise advantages without explicit modeling on preference between different reasoning modes to guide mode selection.

Building on the insights above, we propose AdaGRPO, a variant of GRPO that encourages exploration across reasoning modes and guides mode selection with mode-relative advantage estimation.

**Prefix-Guided Mode Exploration** First, AdaGRPO enforces uniform exploration on different modes during the rollout sampling process. The unified reasoning format defined in Section 3.1 allow us to guide the model to perform reasoning in a specific mode by fixing the corresponding mode prefix in the input sequence. We divide the $2n$ rollouts into two sub-groups: $\{o_j^t\}_{j=1}^n \sim \pi_{\theta_{old}}(O|i, q, m_t)$ consists of $n$ text-based reasoning rollouts with prefix $m_t = $ <text>, and $\{o_j^v\}_{j=1}^n \sim \pi_{\theta_{old}}(O|i, q, m_v)$ consists of $n$ visually-grounded rollouts with prefix $m_v = $ <ground>.

**Reward Function** We follow the reward setup utilized in DeepSeek-R1 (Guo et al., 2025), which includes format rewards and accuracy rewards. The former controls the output format, while the latter assigns a reward of 1 or 0 based on rule-based evaluation of the produced answer. The same reward functions are applied to all rollouts from different modes $\{o_1^t, ..., o_n^t, o_1^v, ..., o_n^v\}$ to support subsequent computation of advantages both between reasoning modes and among rollouts.

**Adaptive Advantage Calculation** To explicitly guide the mode selection, we characterize the mode-relative advantage based on the rewards $\{r_j\}_{j=1}^n$ and $\{r_j\}_{j=n+1}^{2n}$, corresponding to the two sub-groups of rollouts $\{o_j^t\}_{j=1}^n$ and $\{o_j^v\}_{j=1}^n$. Following the Gaussian assumption in GRPO, we

obtain two normal distributions of rewards for the two sub-groups, $\{r_j\}_{j=1}^n \sim P_t$ and $\{r_j\}_{j=n+1}^{2n} \sim P_v$, then we estimate the advantage of mode A over mode B with the probability that a rollout sampled from mode A outperforms a rollout sample from mode B (and vice versa), formulated as:

$$A_v = P_{X \sim P_v, Y \sim P_t}(X > Y); \quad A_t = P_{X \sim P_v, Y \sim P_t}(X < Y)$$
$$\text{where} \quad P_t = N\left(\mu_t = \text{mean}(\{r_j\}_{j=1}^n), \sigma_t^2 = \text{Var}(\{r_j\}_{j=1}^n)\right)$$
$$P_v = N\left(\mu_v = \text{mean}(\{r_j\}_{j=n+1}^{2n}), \sigma_v^2 = \text{Var}(\{r_j\}_{j=n+1}^{2n})\right) \tag{2}$$

Furthermore, we can reasonably assume that $X$ and $Y$ are independent, so $X - Y \sim N(\mu_v - \mu_t, \sigma_v^2 + \sigma_t^2)$. In this way, $A_v$ and $A_t$ can be directly calculated with the cumulative distribution function (CDF) $\Phi$ of the standard Gaussian distribution as $A_v = \Phi\left(\frac{\mu_v - \mu_t}{\sqrt{\sigma_v^2 + \sigma_t^2}}\right) = 1 - A_t$.

As in GRPO, we also compute rollout-wise advantages $\{A_j\}_{j=1}^{2n}$. However, unlike GRPO, we assign different advantages to different tokens within the same sequence. As illustrated in the lower part of Figure 2, mode-relative advantages, $A_t$ and $A_v$, are assigned to mode prefix tokens, guiding models to select the better mode. Rollout-wise advantages $\{A_j\}_{j=1}^{2n}$ are applied to tokens in the thinking processes, enhancing the reasoning capability of the policy model. The specific formulation is:

$$A'_{j,t} = \begin{cases} \mathbb{1}_{\{o_j \in \{o_k^v\}_{k=1}^n\}} A_v + \mathbb{1}_{\{o_j \in \{o_k^t\}_{k=1}^n\}} A_t & \text{if } o_{j,t} \in m \\ A_j = \frac{r_j - \text{mean}(\{r_1, r_2, \ldots, r_{2n}\})}{\text{std}(\{r_1, r_2, \ldots, r_{2n}\})} & \text{otherwise} \end{cases} \tag{3}$$

In summary, the optimization objective of the proposed AdaGRPO is formulated as:

$$\mathcal{J}_{AdaGRPO}(\theta) = \mathbb{E}\left[i, q \sim P(I, Q), \{o_j\}_{j=1}^n \sim \pi_{\theta_{old}}(O|q, i, m_t), \{o_j\}_{j=n+1}^{2n} \sim \pi_{\theta_{old}}(O|q, i, m_v)\right]$$

$$\frac{1}{2n} \sum_{j=1}^{2n} \frac{1}{|o_j|} \sum_{t=1}^{|o_j|} \left( \min\left[ \frac{\pi_\theta(o_j|i, q, o_{j,<t})}{\pi_{\theta_{old}}(o_j|i, q, o_{j,<t})} A'_{j,t}, \text{clip}\left( \frac{\pi_\theta(o_j|i, q, o_{j,<t})}{\pi_{\theta_{old}}(o_j|i, q, o_{j,<t})}, 1 - \epsilon, 1 + \epsilon \right) A'_{j,t} \right] \right)$$

### 3.3.2 DATASET COLLECTION

For RL, we collect a diverse dataset consisting of two parts: (1) existing datasets with verifiable answers, including Geo170K (Gao et al., 2023), OmniCount (Mondal et al., 2025), and MM-Eureka (Meng et al., 2025); (2) a curated subset from the SFT data of LLaVA-OneVision (Li et al., 2024) and InternVL (Chen et al., 2024b), focusing on math, OCR, object counting, science, grounding, and document-related tasks. The subset is filtered based on the answer verifiability and difficulty, and down-sampled to balance different tasks. Please refer to Appendix B.2 for more details.

**Curriculum Learning** To help the model progressively learn reasoning and mode selection in complex scenarios, we design two data mixing strategies: binary mixture, which includes only the OmniCount and Geo170K (with Geo170K being relatively simple geometry problems); and diverse mixture, which consists of all remaining data beyond binary mixing, covering multiple tasks and featuring higher difficulty. Our model first learns from the binary mixed data, then from the diverse mixed data, with both the difficulty and task distribution progressing from simple to complex.

## 4 EXPERIMENTS

### 4.1 EXPERIMENTAL SETUP

**Implementation Details** Based on the proposed framework, we develop two reasoning models, AdaVaR-3B and AdaVaR-7B, which are constructed from Qwen2.5-VL-3B and Qwen2.5-VL-7B (Bai et al., 2025), respectively. Please refer to Appendix B.3 for detailed hyperparameter settings.

**Evaluation Benchmarks** Unlike existing works that focus on specific domains, we comprehensively evaluate the reasoning abilities of AdaVaR across diverse scenarios. For mathematical reasoning, we consider MathVista (Lu et al., 2023), MathVision (Wang et al., 2024), MathVerse (Zhang et al., 2024a), and WeMath (Qiao et al., 2024). In additional, for general-purpose scenarios, we adopt V* (Wu & Xie, 2023), SpatialScore (Wu et al., 2025b), and MMStar (Chen et al., 2024a)

Table 1: Evaluation results on various benchmarks. * indicates results reproduced in this paper. Within each model group with different parameter sizes, the best and second-best results are highlighted in **bold** and underlined, respectively. We use background colors to distinguish text-based , visually-grounded , and our adaptive reasoning models. "Acc." is short for accuracy.

| Model | Math-Oriented | | | | General Scenarios | | | | Average Acc. |
|---|---|---|---|---|---|---|---|---|---|
| | MathVista testmini | MathVision | MathVerse vision-only | WeMath strict | MMStar | V* | POPE all | SpatialScore hard | |
| SFT & Close-source Models | | | | | | | | | |
| GPT-4o (2024) | 63.8 | 30.4 | 39.9 | 42.9 | 64.7 | 66.0 | 86.9 | 30.6 | 53.20 |
| LLaVA-OV-7B (2024) | 62.6 | 17.6 | 19.3 | 17.7 | - | 75.4 | 88.4 | 15.6 | - |
| InternVL3-8B (2025) | 70.5 | 28.6 | 39.8 | 37.5 | 66.3 | - | 91.1 | 12.9 | - |
| Reasoning Models based on Qwen2.5-VL-3B | | | | | | | | | |
| Qwen2.5-VL-3B (2025) | 62.3 | 21.2 | 31.2* | 21.2* | 53.1* | 73.5* | 87.2* | 16.2 | 45.74 |
| LMM-R1 (2025) | 63.2 | **25.2** | **37.3*** | 29.2* | 55.0* | 74.9* | 86.7* | 18.9* | 48.80 |
| VLAA-Thinker-3B (2025a) | 61.0 | 24.4 | 36.4 | 33.8 | 54.8* | 56.0* | 87.8* | 19.4* | 46.70 |
| GRIT-3B (2025) | 59.8 | 21.2* | 28.4* | 29.1* | 52.7* | 72.3* | 83.0* | **19.7*** | 45.80 |
| ViGoRL-3B (2025) | 55.2* | 18.9* | 24.2* | 28.5* | 51.4* | **78.1** | 88.0* | 18.2* | 45.70 |
| AdaVaR-3B (Ours) | **69.8** | 24.5 | 35.2 | **33.8** | **59.3** | 77.0 | **88.2** | 18.9 | **50.84** |
| Reasoning Models based on Qwen2.5-VL-7B | | | | | | | | | |
| Qwen2.5-VL-7B (2025) | 68.2 | 25.1 | 41.1 | 31.2 | 60.3* | 78.0* | 87.8* | 15.2 | 50.90 |
| VLAA-Thinker-7B (2025a) | 68.0 | 26.4 | 48.2 | 41.5 | 62.6* | 78.5* | 86.2* | 22.3* | 54.20 |
| MM-Eureka (2025) | 72.6 | 28.1 | 45.4 | 36.9 | **64.0*** | 59.7* | 86.3* | **27.1*** | 52.52 |
| OVR-7B (2025) | 72.1 | **46.4*** | **54.6** | 44.6 | 62.7 | 62.5* | 83.2 | 20.0* | 55.76 |
| Orsta-7B (2025) | 72.5 | 28.2* | 42.9* | 31.8* | 59.6* | 78.0* | 86.9* | 16.4* | 52.04 |
| DeepEyes (2025) | 70.1 | 26.6 | 40.7* | 32.7* | 61.3* | **90.1** | 87.9* | 20.3* | 53.72 |
| Chain-of-Focus (2025) | 63.1* | 22.9* | 32.1* | 33.1* | 59.3* | 88.0 | 88.4 | 20.6* | 50.94 |
| ViGoRL-7B (2025) | 63.5* | 23.3* | 32.6* | 36.3* | 54.3* | 86.4 | 88.3* | 19.5* | 50.53 |
| AdaVaR-7B (Ours) | **74.4** | 28.5 | 43.0 | **44.8** | 63.0 | 83.4 | **89.0** | 20.4 | **55.82** |

to evaluate the capabilities of models in visual search, spatial reasoning, and general perceptual reasoning, respectively, while POPE (Li et al., 2023) is used to diagnose hallucination risk.

During evaluation, our AdaVaR model naturally incorporates a mode-switching mechanism: if the model gets stuck in repetitive logic and cannot conclude an answer (between <answer> and </answer>) while reasoning in a particular mode, we switch to another mode and try again. More evaluation details, including descriptions of the compared models, are included in Appendix B.4.

## 4.2 OVERALL PERFORMANCE

Table 1 presents a thorough evaluation of visual reasoning models. Several insights can be gleaned: (1) As argued in Section 1, models based on a single reasoning mode tend to become experts in specific domains. Text-based models, such as VLAA-Thinker-3B and OVR-7B, achieve notable improvement on mathematical problems but show significant degradation in V* and POPE. In contrast, visually-grounded models consistently improve performance on V* and control hallucination, but struggle with math problems–DeepEyes makes a slight improvement while other methods fail to maintain the original mathematical capability of Qwen2.5-VL. (2) AdaVaR-3B and AdaVaR-7B, on the other hand, combine the strengths of two modes in an adaptive manner, and are the only models that outperform Qwen2.5-VL across all datasets. Specifically, AdaVaR achieves the best performance on MathVista, WeMath, and POPE, while excelling on MMStar, MathVision, and V*. (3) In terms of the overall performance measured by average accuracy, AdaVaR achieves the best results in both the 3B and 7B groups. Notably, AdaVaR-3B reaches a level very close to Qwen2.5-VL-7B, while AdaVaR-7B surpasses GPT-4o. In summary, our adaptive reasoning framework provides an effective and feasible solution for building general visual reasoning models.

## 4.3 DELVING INTO ADAPTIVE VISUAL REASONING

Beyond the overall improvement, we delve into the mechanism of adaptive reasoning based on AdaVaR-3B and several ablated models, aiming to answer the questions raised in Section 1.

### 4.3.1 QUESTION 1: CAN WE UNIFY DIFFERENT REASONING MODES IN ONE MODEL?

As shown in Table 2, after SFT and RL, two reasoning modes induce different prediction patterns in terms of performance: the text-based mode excels at math, while the grounded mode is better at object-centric tasks. Please also see cases in Appendix C.8 for a qualitative comparison.

Table 2: Performance of AdaVaR with different reasoning modes at various stages. AdaVaR without a subscript performs adaptive reasoning. AdaVaR with the subscripts $G$ and $T$ denote the performance using visually-grounded mode and text-based mode, respectively. GRD% indicates the proportion of times AdaVaR chooses the grounded mode in the corresponding dataset.

| Model | Math-Oriented | | | | General Scenarios | | | | Average Acc. |
|---|---|---|---|---|---|---|---|---|---|
| | MathVista | MathVision | MathVerse | WeMath | MMStar | V* | POPE | SpaScore | |
| Qwen2.5-VL-3B | 62.3 | 21.2 | 31.2 | 21.1 | 53.1 | 73.5 | 86.1 | 16.2 | 45.6 |
| Stage 1 SFT-based Models | | | | | | | | | |
| AdaVaR-SFT-3B | 60.4 | 23.2 | 33.0 | 29.5 | 54.7 | 75.5 | 89.1 | 16.8 | 47.8 |
| AdaVaR-SFT-3B$_G$ | 56.4 | 20.3 | 28.5 | 29.5 | 53.2 | 75.5 | 89.2 | 17.6 | 46.3 |
| AdaVaR-SFT-3B$_T$ | 62.5 | 22.9 | 31.4 | 31.1 | 55.9 | 65.6 | 83.1 | 17.8 | 46.3 |
| GRD% | 31% | 5% | 1% | 78% | 90% | 98% | 94% | 96% | - |
| Stage 2 RL-based Models | | | | | | | | | |
| AdaVaR-3B | 69.8 | 24.5 | 35.2 | 33.8 | 59.3 | 77.1 | 88.2 | 18.9 | 50.8 |
| AdaVaR-3B$_G$ | 66.1 | 22.1 | 33.9 | 32.1 | 56.3 | 76.4 | 88.2 | 18.5 | 49.3 |
| AdaVaR-3B$_T$ | 68.1 | 23.0 | 34.9 | 32.6 | 58.5 | 58.6 | 84.6 | 17.7 | 47.3 |
| Upper Bound | 74.8 | 30.9 | 41.7 | 38.3 | 67.6 | 78.5 | 95.6 | 27.8 | 56.8 |
| GRD% | 2% | 0% | 0% | 0% | 51% | 99% | 100% | 59% | - |
| Baseline Models based on a Single Reasoning Mode | | | | | | | | | |
| Grounded-SFT-RL | 62.6 | 22.2 | 34.4 | 33.0 | 55.4 | 77.0 | 87.4 | 18.4 | 48.7 |
| Text-SFT-RL | 67.2 | 23.5 | 36.1 | 32.2 | 56.0 | 70.7 | 85.6 | 18.8 | 48.8 |
| Mix-SFT-RL | 65.0 | 23.2 | 33.6 | 28.8 | 56.2 | 76.9 | 85.4 | 18.6 | 48.5 |

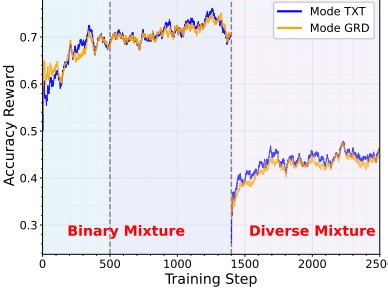

(a) Training rewards on math problems.

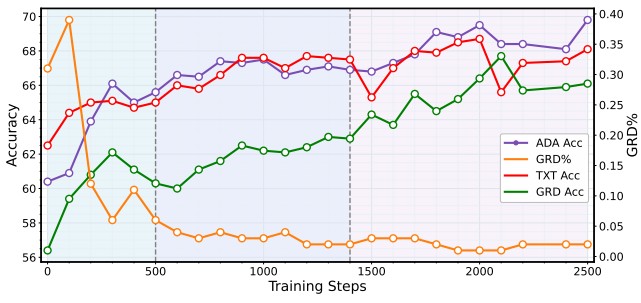

(b) Performance evolution of different modes on MathVista.

Figure 3: Math-related training dynamics and evaluation metrics during Stage 2. ADA, TXT, and GRD are short for the three reasoning modes: adaptive, text-based, and grounded, respectively.

**Comparison with Single-Mode Baselines**    Furthermore, we construct two single-mode baselines, Grounded-SFT-RL and Text-SFT-RL, under the same training settings: using SFT data corresponding to each mode and RL data consistent with AdaVaR. We find that: (1) The difference between the single-mode baselines and the respective modes of AdaVaR is minimal, suggesting that consolidating both modes into one model does not hinder the improvement of either individual mode; (2) Both baselines perform worse than AdaVaR, indicating that, compared to simply training with diverse data, adaptive reasoning is more effective for developing general-purpose reasoning capabilities.

**Necessity of Mode-Specific Prefix**    Another question is whether we need to distinguish between different modes. To investigate this, we remove the prefix token and mix the SFT data from two modes together, creating Mix-SFT-RL. Its performance is even inferior to the single-mode baselines. This suggests that simply mixing data from different modes is insufficient, and that the mode-specific prefix token introduced in AdaVaR helps the model differentiate between modes and facilitate uniform exploration across modes during the RL stage, resulting in a better overall performance.

**The Upper Bound of Adaptive Reasoning**    Table 2 also presents the upper bound of adaptive reasoning: the performance when a prediction is considered correct if either mode yields the correct answer. The upper bound clearly surpasses Qwen2.5-VL-3B and single-mode models. Even in mathematical problems, where text-based reasoning is more dominant, the grounded mode can solve some problems that the text-based mode cannot. This phenomenon demonstrates the complementary nature between two modes and highlights the great potential of the adaptive MoVT paradigm.

Table 3: Ablation study. Ada-Adv and PG-Exp are abbreviation for adaptive advantage and prefix-guided exploration, respectively. The definition of diverse mixed data is provided in Section 3.3.2.

| Model | Math-Oriented | | | | General Scenarios | | | | Average Acc. |
|---|---|---|---|---|---|---|---|---|---|
| | MathVista | MathVision | MathVerse | WeMath | MMStar | V* | POPE | SpaScore | |
| AdaVaR-3B | 69.8 | 24.5 | 35.2 | 33.8 | 59.3 | 77.1 | 88.2 | 18.9 | 50.8 |
| w/o Ada-Adv + PG-Exp | 66.3 | 23.8 | 34.9 | 31.3 | 56.6 | 75.4 | 89.1 | 19.7 | 49.6 |
| w/o Ada-Adv | 68.4 | 24.7 | 34.4 | 33.7 | 58.9 | 77.4 | 88.0 | 17.4 | 50.3 |
| w/o Diverse Mixed Data | 67.4 | 23.7 | 33.1 | 33.4 | 57.3 | 76.4 | 82.1 | 18.5 | 49.0 |
| w/o Curriculum Learning | 66.8 | 24.4 | 34.9 | 33.4 | 57.8 | 76.9 | 88.2 | 18.5 | 50.1 |

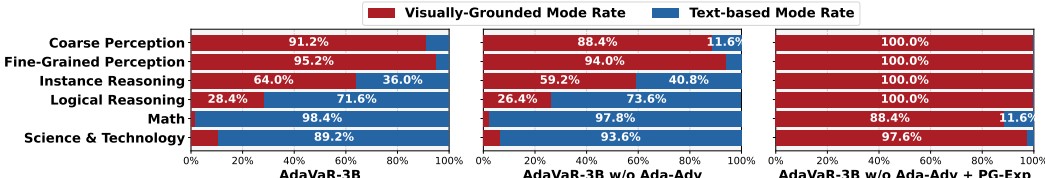

Figure 4: Proportion of each mode selected by different models across categories in MMStar.

### 4.3.2 QUESTION 2: CAN ADAVAR SELECT APPROPRIATE MODE?

Comparing the SFT and RL results in Table 2, we find that (1) the SFT model exhibits unexpected behavior: although the text mode performs significantly better on MathVista, it still chooses the grounded mode 31% of the time, resulting in worse performance than the text mode alone. This also supports our argument in Section 1: it is hard to guide the model to learn effective mode selection during the SFT stage. (2) After Stage 2, AdaVaR exhibits reasonable mode selection: it prefers the text mode for math problems and the grounded mode for V* and POPE tasks (as well as fine-grained choices shown in Figure 4); its performance consistently surpasses single-mode reasoning across all benchmarks, indicating that the RL stage helps the model acquire effective mode selection abilities.

**Learning Curves of Mode Selection**   How is the mode-selection capability acquired? Figure 3 illustrates the learning process using the mathematical scenario as an example. It is observed that the relative performance across modes (as reflected by the relative reward levels of two modes in Figure 3a) guides the mode-selection preference, which is illustrated by the GRD% curve in Figure 3b. The RL process can be divided into three phases, which are delineated in the figure using different background colors in Figure 3: (1) Exploration phase: at the beginning, performance estimates for different modes obtained via RL sampling are still unstable, and the GRD mode even outperforms the TXT mode for a period, with the choice of mode also fluctuating. (2) Stabilization phase: as the text mode demonstrates a clear advantage in RL sampling, the adaptive model gradually tends to select the text mode. However, it does not yet achieve optimal performance, and the adaptive performance is slightly weaker than the TXT mode. (3) Refinement phase: by introducing more difficult and diverse data, the capabilities of each mode steadily improve, and the model learns which types of math problems benefit from choosing the GRD mode, ultimately pushing the ADA mode to reach optimal performance. Please refer to Appendix C.5 for the training dynamics in other domains, and to Appendix C.1 for a more comprehensive analysis of the learned mode selection behavior.

**Effectiveness of AdaGRPO**   Table 3 and Figure 4 present the results of ablation experiments related to key designs in AdaGRPO. (1) When both Ada-Adv and PG-Exp are removed, which is equivalent to the original GRPO, we observe that RL reinforces the SFT model's unreasonable preference for the grounded mode, resulting in overall inferior performance. (2) With PG-Exp included, the model is able to fully explores different modes and compares rollouts between them. In datasets where the advantages between modes are clear, such as V* and MathVision, the model makes reasonable mode selections and achieves good performance. (3) With further introduction of adaptive advantage, AdaVaR learns more refined mode selection abilities, leading to improvements on complex datasets like MathVista and MMStar, as well as further boosting overall performance.

**Training Data Matters**   Apart from demonstrating the importance of AdaGRPO, the results in Table 3 demonstrate that diverse data enables the model to learn more generalizable reasoning and

mode selection abilities. Additionally, the curriculum learning strategy is also highly effective during the RL phase. Both factors contribute to the improved model performance.

## 5 CONCLUSION

This paper presents an adaptive visual reasoning paradigm, MoVT, for building general visual reasoning models. We propose AdaVaR, a framework that unifies the reasoning capabilities of different modes within a single model through SFT, and employ a tailored RL algorithm, AdaGRPO, to train the model to choose the appropriate mode based on the context. Experimental results demonstrate that, unlike existing models that excel only in specific domains, AdaVaR achieves consistent improvements across multiple scenarios–with AdaVaR-7B surpassing GPT-4o in average performance.

## ETHICS STATEMENT

As described in Appendix B.2, all data sources used in this paper are open-source and available for research purposes. We strictly comply with the corresponding terms of use to ensure our data can be used for scientific research. The original datasets have been widely used in related studies and are confirmed to contain no harmful content. Building on these datasets, we performed certain reconstruction and filtering steps, as detailed in Appendix B.2; this process only involves modifying the problem format and providing additional information for filtering. Through sampling-based spot checks, we have verified that this data construction process does not introduce harmful content, and our filtering is entirely for research purposes, without any discriminatory bias.

Our reasoning models, AdaVaR-3B and AdaVaR-7B, currently focus on English scenarios, and we aim to expand to more languages in the future. Our work and artifacts are designed with the principle of universality and fairness, without any preference for specific demographic groups.

## REPRODUCIBILITY STATEMENT

We have taken rigorous steps to ensure that relevant researchers can easily reproduce this work. First, our approach is built upon the open-source Qwen2.5-VL-Instruct model. Second, in Section 3, we describe our AdaVaR framework in detail, including how to represent reasoning generation sequences in a unified way (Section 3.1), how to perform cold-start SFT (Section 3.2), and how to train the model's mode selection capability via RL (Section 3.3). Furthermore, we provide extensive additional details, including the implementation details and formulas of AdaGRPO (Section 3.3.1), detailed model prompts (Appendix B.1), data construction details (Section 3.3.2 and Appendix B.2), specific training parameter settings (Appendix B.3), and evaluation details (Appendix B.4). In summary, we believe researchers can readily reproduce the findings of this paper. Our data, model weights, and code are available at https://github.com/Future-Living-Lab/mixture-of-visual-thoughts.

## ACKNOWLEDGMENT

This work was supported by Alibaba Group through Alibaba Research Intern Program. The support was also provided by AI for Science Program, Shanghai Municipal Commission of Economy and Informatization (2025-GZL-RGZN-BTBX-02028), and National Key R&D Program of China (2023YFF1204800). Computational resources were provided in part by CFFF.

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

## A  APPENDIX ROADMAP

To assist the reader in navigating our supplementary materials, we provide a brief overview of the Appendix below. Appendix B provides supplementary implementation details, including:

- Detailed prompts in Appendix B.1.

- How to collect the training data (Appendix B.2).

- Training details in Appendix B.3.

- Details of the evaluation process in Appendix B.4.

Appendix C presents supplementary experiments and analysis:

- Appendix C.1 analyzes the mode-selection behavior learned by AdaVaR, including fine-grained selection, the impact of images on mode selection, and attention visualization.

- Appendix C.2 explores whether the adaptive reasoning ability can be induced through SFT.

- A detailed error analysis is provided in Appendix C.3.

- Computational efficiency analysis is conducted in Appendix C.4.

- Appendix C.5 presents training dynamics during the two stages.

- The scalability of MoVT and AdaVaR, as well as the direction for future extension, are discussed in Appendix C.6.

- Appendix C.7 provides additional ablation studies on several designs.

- A comprehensive qualitative case study is included in Appendix C.8.

## B  SUPPLEMENTED IMPLEMENTATION DETAILS

### B.1  DETAILED PROMPTS

Here we provide the complete prompt template used in our adaptive reasoning model:

> **System**: You are a helpful assistant. The user asks a question related to an image, you need to solve it. Please first think about the reasoning process in the mind and then provide the user with the answer. The reasoning process and answer are enclosed within <think> </think> and <answer> </answer> tags, respectively, i.e., </think> reasoning process here </think> <answer> answer here </answer>. You have two modes of thinking, and you should choose the appropriate one based on the question:
> 1. *Grounded Reasoning*: Use this mode when you need to locate specific objects in the visual input. In your reasoning path, identify key objects and provide their corresponding bounding box coordinates in the format 'object[x1, y1, x2, y2]'. When using this mode, begin your response with the tag <ground>.
> 2. *Text-based Reasoning*: Use this mode for general reasoning based solely on textual thoughts. No object localization or coordinate output is required in this mode. When using this mode, begin your response with the tag <text>.
> Choose the mode that best fits the task, and structure your response accordingly.
> **User**: {QUESTION}. Please first select the appropriate reasoning mode based on the question, using <ground> or <text> to indicate the type, then follow the corresponding format to output the reasoning process, and finally provide the correct answer according to the user requirement.
> **Assistant**: <mode prefix> <think> ... </think> <answer> ... </answer>

where the {QUESTION} will be replaced by the input question. Please note that we have added a post prompt after each question to emphasize mode selection, in order to prevent the model from ignoring the system prompt due to the long input sequence of images and questions.

## B.2 Details on Training Data Collection

### B.2.1 SFT Data Collection

**Visually-Grounded Reasoning Data** Regarding the visually-grounded mode, we utilize the data constructed in VoCoT (Li et al., 2025), we find that the dataset lacks multiple-choice samples, to avoid potential bias induced by such a distribution gap, we employ GPT-4o to rewrite a subset into multiple-choice format (by providing the original question and the correct option). To control the total size, we down-sample the GQA subset, resulting in a final dataset of 119K examples.

During the re-formulation process, we strictly adhere to copyright regulations. We would like to clarify that VoCoT data is released under the Apache 2.0 license, which permits free reconstruction, modification, and creation of derivative datasets. We strictly adhere to these requirements and will include the appropriate VoCoT attributions in our future open-sourcing process, clearly stating in our documentation that our dataset is derived from VoCoT with modifications.

For quality assessment, we sample 100 reconstructed samples to check for formatting errors or the generation of false-negative options. None of the samples exhibited any issues, which is primarily due to the simplicity of the reconstruction task and GPT-4o's strong instruction-following capability. Therefore, we believe the quality of the reconstructed data is reliable.

**Text-based Reasoning Data** For the text-based mode, we aim to avoid introducing other inductive bias; therefore, instead of using an R1-OneVision–style approach (Yang et al., 2025) built from DeepSeek-R1, we distill Orsta (Ma et al., 2025), a reasoning model obtained by applying RL directly to Qwen2.5-VL. The specific procedure is as follows: We randomly sample a total of 250K questions from R1-OneVision (Yang et al., 2025), llava-CoT (Xu et al., 2024), NuminaMath1.5 (LI et al., 2024), and Virgo (Du et al., 2025). For each question, Orsta generates eight responses with temperature of 1. We then use a rule-based method to determine correctness and retain only the correct instances; if multiple reasoning traces are correct, we randomly select one during training. We also observe that Orsta might skip the reasoning process and directly output an option, so we distinguish reasoning data from direct-answer data based on whether the model's reply contains the <think> and <answer> tags, and apply different system prompts for these two data types. After the rejection sampling, we obtain 115K reasoning examples and 95K direct-answer examples.

### B.2.2 RL Data Collection

**Difficulty-based Filtering** Since overly simple questions cannot provide effective supervision during the RL process, we filter them out from the RL data. Following prior work (Ma et al., 2025; Zheng et al., 2025), we adopt a model-based evaluation strategy: questions that are answered correctly in all 8 random trials by Qwen2.5-VL-7B are considered too simple and are excluded.

**RL Data Collection** As introduced in Section 3.3.2, our RL dataset consists of two parts: (1) existing data with verifiable answers, including math-related Geo170K (Gao et al., 2023) and MM-Eureka (Meng et al., 2025), as well as OmniCount (Mondal et al., 2025) for object counting. We further filter MM-Eureka following the process: we first perform uniform sampling across the data sources (capping any category exceeding 1,500 instances at 1,500) and remove overly simple questions using the difficulty-based filtering method described in the previous paragraph, ultimately retaining 13K examples. (2) Additionally constructed data for general scenarios. The questions and answers come from the SFT data used in LLaVA-OneVision (Li et al., 2024) and InternVL (Zhu et al., 2025). We retain only the subsets that are related to mathematics, OCR, object counting, STEM, grounding, and table/document understanding. For this part of data, we use an offline Qwen2.5-VL-72B (Bai et al., 2025) model to assign task categories to each samples, using the same categorization scheme as in Orsta (Ma et al., 2025).

To ensure that answers could be verified by a rule-based reward function, we first use Qwen2.5-VL-72B to determine the answer type. Depending on the task and question type, we collect and evaluate responses from multiple models via offline experiments, and, guided by the question type, we augment some input queries with prompts specifying the required answer type. Ultimately, we retain only those questions that could be judged accordingly. Finally, after filtering by difficulty to remove overly easy questions, we retain 22K examples.

Table 4: The detailed training hyper-parameters of AdaVaR.

| Configuration | Stage 1 SFT | Stage 2 RL |
|---|---|---|
| Model Initialization | Qwen2.5-VL-Instruct | Stage1 |
| Trainable Modules | MLP + LLM | All Modules |
| Optimizer | AdamW | AdamW |
| Optimizer Hyperparameters | $\beta_1 = 0.9, \beta_2 = 0.95, \epsilon = 1e^{-6}$ | $\beta_1 = 0.9, \beta_2 = 0.95, \epsilon = 1e^{-6}$ |
| Global batch size | 256 | 32 |
| Peak learning rate | 5e-6 | 1e-6 |
| Learning rate scheduler | Cosine | Linear |
| Training Epochs | 1 | 1 |
| Warm-up ratio | 0.01 | 0.0 |
| Weight decay | 0.1 | 0.0 |
| Max Sequence Length (SFT) | 4096 | - |
| Max Prompt Length (RL) | - | 2048 |
| Max RL Completion Length (RL) | - | 2048 |
| KL Penalty Coefficient (RL) | - | 0.04 |
| Generation Temperature (RL) | - | 0.9 |
| # Rollouts per Sample (RL) | - | 8 |
| Numerical precision | bfloat16 | bfloat16 |
| GPU Usage | 8 NVIDIA A100 | 32 NVIDIA A100 |
| DeepSpeed Configure | zero3 | zero3 |
| Training Time | 4h | 29h |

**Multi-Modal Data Processing**  Since our model is based on Qwen2.5-VL (Bai et al., 2025), our data processing methods strictly follow those of Qwen2.5-VL: we apply the same dynamic resolution strategy for patchifying images and use the corresponding tokenizer for tokenizing text.

**Summary on Data Efficiency**  Our AdaVaR framework operates under a data-efficient setting, in terms of (1) Data sources: We rely entirely on existing datasets, and the processes of data reconstruction and filtering do not involve any additional human annotation. (2) Data volume: We use 329K samples in the SFT phase and 55K in the RL phase, totaling 384K—comparable to other RL methods such as OVR-7B (Wei et al., 2025) (428K), and significantly less than the SFT data used for mainstream LVLMs, such as 1.8M samples used in LLaVA-OneVision (Li et al., 2024).

### B.3 TRAINING DETAILS

We list the hyper-parameter settings in Table 4, and here we discuss the rationale behind several key parameter choices.

**The Number of Training Epochs in the SFT Stage**  We follow the commonly adopted setup in existing works (Yang et al., 2025; Chen et al., 2025a) to train one epoch for SFT. We validate the rational behind this setting in Appendix C.5.

**The Number of Rollouts Generated per Sample**  In the RL stage, "# Rollouts per Sample" refers to the number of rollouts generated for each question. Generating more rollouts per sample can help provide better RL supervision signals and improve performance, but it also incurs higher computational costs. Therefore, to ensure a fair comparison with other models in Table 1, we set it to 8, which corresponds to $n = 4$ in Figure 2 and the equations in Section 3.3, meaning that the policy model generates 4 rollouts with each reasoning mode, for a total of 8. Similarly, for a fair comparison with ablated models, when training the ablated models that are based on a single reasoning mode in Section 4.3, we set the number of rollouts generated in GRPO to 8.

**Coefficient for KL Penalty**  For brevity, the KL penalty are omitted in equations in Section 3.3. In practice, we introduce the KL penalty term to preserve the knowledge of reasoning modes learned during the SFT phase. The coefficient is set to 0.04, following the settings used in existing works and frameworks (Chen et al., 2025b; Shen et al., 2025; Hugging Face, 2025).

**Training Dynamics Monitor**  During training, we implement a monitor to track the performance of different modes on various types of data, allowing us to better observe the relative strengths and weaknesses between modes, the detailed training dynamics are provided in Appendix C.5.

---

**Algorithm 1** Mode-relative advantage calculation.

---

**Require:** $r[1..2n]$ # rewards of all rollouts where $r[1:n]$ and $r[n+1:2n]$ correspond to the text-based mode and the visually-grounded mode, respectively.
**Require:** $\Phi(x)$ # CDF of the standard Gaussian distribution $N(0, 1)$
**Ensure:** $A_t, A_v$
  1: # mean and standard deviation of text-based mode
  2: $\mu_t \leftarrow \text{mean}(r[1:n])$, $\sigma_t \leftarrow \text{std}(r[1:n])$
  3: # mean and standard deviation of visually-grounded mode
  4: $\mu_v \leftarrow \text{mean}(r[n+1:2n])$, $\sigma_v \leftarrow \text{std}(r[n+1:2n])$
  5: # Advantage of text-based mode over visually-grounded mode
  6: $A_t = \Phi\left(\frac{\mu_t - \mu_v}{\sqrt{\sigma_v^2 + \sigma_t^2}}\right)$
  7: # Advantage of visually-grounded mode over text-based mode
  8: $A_v = 1 - A_t$
  9: **return** $A_t, A_v$

---

**Algorithm for Advantage Calculation**    Specifically, in Algorithm 1, we provide the pseudo code for calculating the mode-relative advantage $A_v$ and $A_t$ in Section 3.3.1.

## B.4   EVALUATION DETAILS

**Benchmark Details**    As indicated in the header of Table 1, we use the testmini split in Math-Vista (Lu et al., 2023), the test split in MathVision (Wang et al., 2024), the vision-only portion of the testmini split in MathVerse (Zhang et al., 2024a), the val split in MMStar (Chen et al., 2024a), the test split in V* (Wu & Xie, 2023), the testmini split in WeMath (Qiao et al., 2024), all four splits of POPE (Li et al., 2023), and the hard split of SpatialScore (Wu et al., 2025a). For the eight benchmark datasets used in this paper, the evaluated model is required to produce structured outputs (e.g., option letters or numbers), we prompt the model to generate the desired format, and evaluate the results offline using rule-based procedures. Except for WeMath, for which we report the strict accuracy as specified in the original paper (Qiao et al., 2024).

**Average Schema**    Since each benchmark represents a different domain, but the number of samples in each is inconsistent, all benchmarks are weighted equally for the average accuracy. We aim for the average performance to comprehensively reflect capabilities across multiple domains.

**Mode-Switching Mechanism**    Specifically, the phrase "getting stuck in repetitive logic" mentioned in Section 4.1 refers to the situation where the mode reaches the maximum number of tokens (8192, significantly longer than the typical reasoning length of AdaVaR) without generating an answer in the form of <answer></answer>. Please note that our mode-switching mechanism does not leverage any additional information, ensuring a fair comparison with other models; rather, mode switching is a new feature enabled by our MoVT reasoning paradigm.

**Compared Models**    As discussed in Section 1, the compared models can be divided into two groups: text-based models including LMM-R1 (Peng et al., 2025), VLAA-Thinker (Chen et al., 2025a), MM-Eureka (Meng et al., 2025), OVR-7B (Wei et al., 2025), and Orsta-7B (Ma et al., 2025); as well as the visually-grounded models including GRIT (Fan et al., 2025), ViGoRL (Sarch et al., 2025), DeepEyes (Zheng et al., 2025), and Chain-of-Focus (Zhang et al., 2025). Since most compared models focus on specific domains and do not provide evaluation results across all benchmarks, we conduct evaluation to fill in the missing results for a comprehensive comparison. Specifically, to ensure a fair comparison, we do not retrain the corresponding models; instead, we strictly follow the original papers, using the officially released pretrained checkpoints and evaluating them with the recommended parameters according to their respective evaluation protocols.

**Evaluation Environment and Cost**    All evaluation is conducted on a single node with 8 A100 GPUs, accelerated by the vLLM framework (Kwon et al., 2023). As measured in Appendix C.4.2, a complete evaluation of the 3B-scale model across all benchmarks requires 17.7 GPU hours

Table 5: The proportion of mode selected by AdaVaR-3B across different image types in MMStar.

| Mode Rate | Natural Images | Art | Document | Diagram | Geometry | Others |
|---|---|---|---|---|---|---|
| Grounded | 92.20% | 39.40% | 8.10% | 4.50% | 0% | 0% |
| Text-based | 7.80% | 60.60% | 91.90% | 95.50% | 100% | 100% |

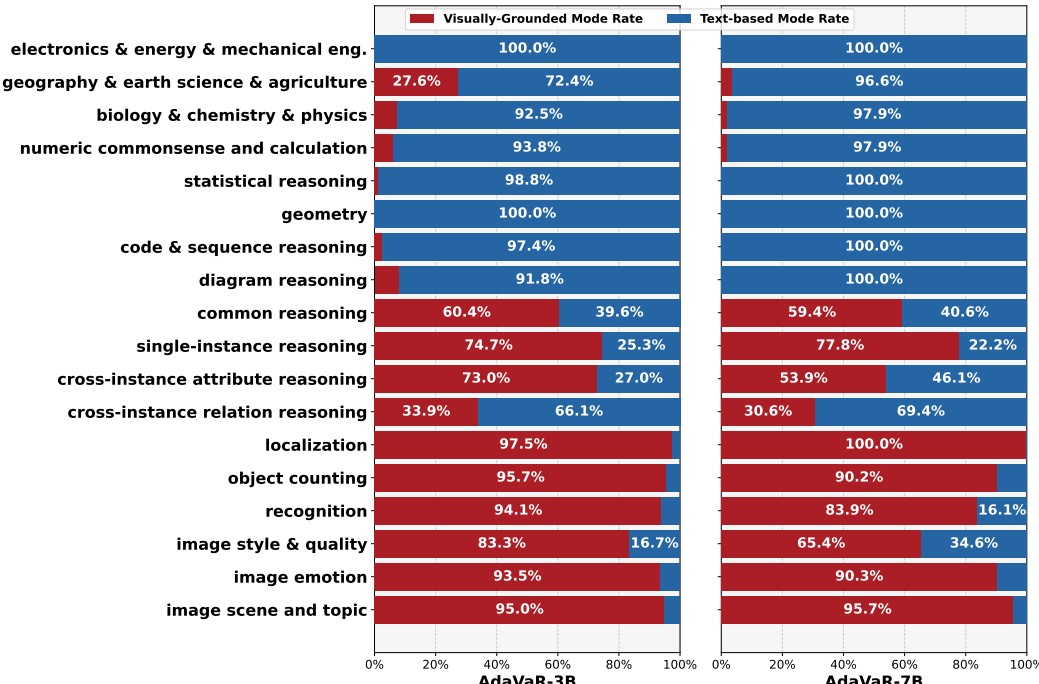

Figure 5: Proportion of each mode selected by different models across sub-categories in MMStar.

(on A100), making frequent performance monitoring during evaluation highly resource-intensive. Therefore, we only track MathVista performance, as shown in Figure 3.

### B.5 LLM Usage

Our use of LLMs in this paper mainly comprises two parts: writing assistance and tools for data construction. During writing, we use GPT-4o to help polish the exposition. In the data construction process, as described in Appendix B.2, we employ Qwen2.5-VL and GPT-4o to assist with data reconstruction, annotation, and filtering.

## C Supplementary Experimental Results and Analysis

### C.1 Delving into Mode-Selection Behaviors

In the main text, we have discussed mode selection at the dataset level and the coarse-grained task level within MMStar. This section provides additional analysis to complement that discussion.

### C.1.1 Mode Selection at the Fine-Grained Task Level

Figure 5 illustrate the fine-grained mode selection tendencies of AdaVaR-3B and AdaVaR-7B. We observe that, since the 7B model has richer knowledge, AdaVaR-7B prefers the text-based mode for knowledge-related categories, such as geography & earth science & agriculture. This also highlights the necessity of the RL process: we need to sample during RL to better estimate the strengths and weaknesses of each mode, which are different for different models (there is no universal ground truth of mode selection that is applicable for all models).

Table 6: Comparison between AdaVaR and simple mode ensemble strategies.

| Model | MathVista | MathVision | MathVerse | WeMath | MMStar | POPE | V* | SpaScore | Avg. Acc |
|---|---|---|---|---|---|---|---|---|---|
| AdaVaR-3B | 69.8 | 24.5 | 35.2 | 33.8 | 59.3 | 77.1 | 88.2 | 18.9 | 50.8 |
| Expert Pick | 68.1 | 23.0 | 34.9 | 32.6 | 56.3 | 76.4 | 88.2 | 18.5 | 49.8 |
| Max(GRD, TXT) | 68.1 | 23.0 | 34.9 | 32.6 | 58.5 | 76.4 | 88.2 | 18.5 | 50.1 |

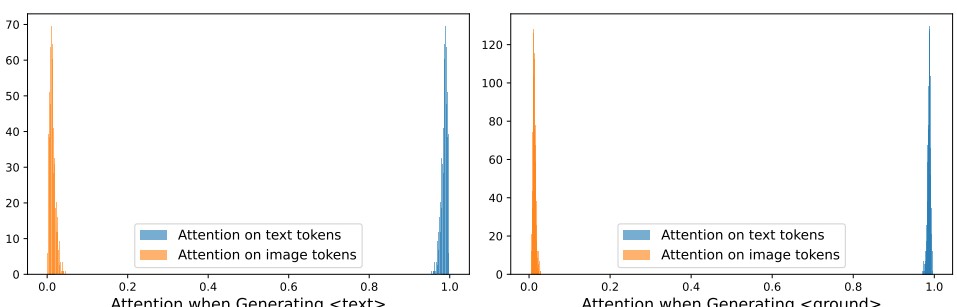

Figure 6: Distribution of attention mass over text tokens (blue) and image tokens (orange) when generating the `<text>` (left) and `<ground>` (right) mode prefixes on MMStar.

### C.1.2 The Impact of Image Types on Mode Selection

Furthermore, we conduct an analysis from the perspective of images. We employ GPT-5 to annotate the image types in MMStar and analyze their impact on mode selection. As presented in Table 5, image type also has a significant impact on mode selection: (i) for natural images, the model tends to use the grounded mode to precisely locate and analyze key objects; (ii) when images contain more abstract or harder-to-localize concepts—such as artistic styles in artwork, content in documents and diagrams, or angles in geometry problems—the model tends to favor the text-based reasoning mode.

### C.1.3 Attention Visualization for Mode Selection

To better understand how MoVT selects between the text-based and grounded modes, we visualize the attention patterns used when predicting the mode prefix. Concretely, for each sample we probe the attention scores from the token immediately preceding the mode prefix (i.e., the token that "decides" the mode) to all previous tokens. Following (Kaduri et al., 2025), we select the middle 25% of layers and average the attention weights across these layers and across all attention heads.

We first aggregate the attention mass over all image tokens and all text tokens on the MMStar dataset. As shown in Figure 6, when generating either the `<text>` or `<ground>` prefix, the model places the vast majority of attention on text tokens (0.97 in average) rather than image tokens (0.03 in average). This is consistent with the nature of MMStar, where the question type is primarily encoded in the textual prompt and multiple questions can be asked about the same image.

We further examine how attention is distributed over different textual tokens. Figure 7 visualizes token-level attention maps for representative examples from six benchmarks. We observe that the attention concentrates on tokens that define the question type, such as the interrogative words "which", "what"; key mathematical terms like " perimeter", "area" and object-oriented concepts of "background", "bus" and "person".

### C.1.4 Comparison with Naive Mode Ensemble Strategies

To validate the effectiveness of the mode selection learned by AdaVaR, we compare it with two naive strategies to use external knowledge for mode ensemble at the dataset level: (1) Expert pick: selection based on expert knowledge—here, we choose the text-based mode for math benchmarks and the grounded mode for others. Note that, in practical applications, even experts would find it difficult to make reasonable judgments on mixed-task datasets like MMStar and SpatialScore. (2) Max(GRD, TXT): for each dataset, we select the mode that achieves higher accuracy. Note that this strategy uses additional information from the dataset, making it inherently unfair for comparison.

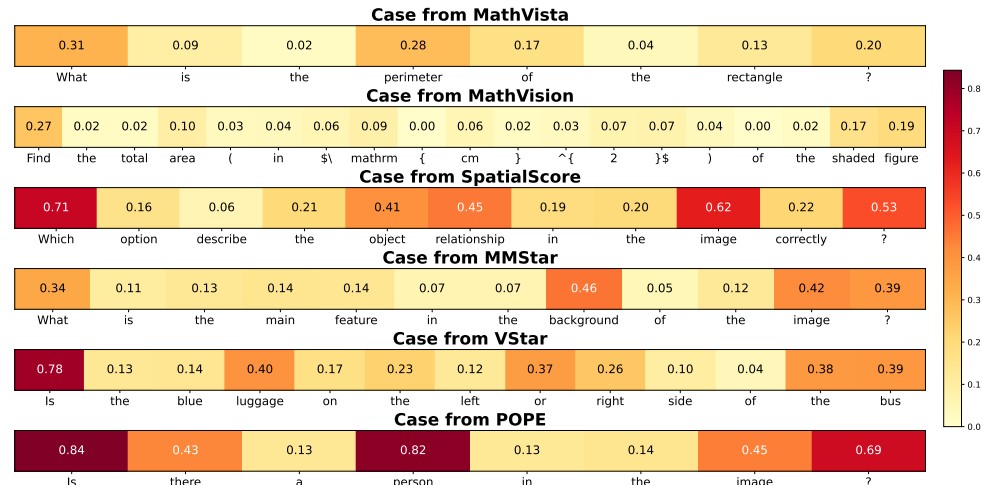

Figure 7: Token-level attention heatmaps over the question when generating the mode prefix.

Table 7: Performance of adaptive reasoning models.

| Model | Math-Oriented | | | | General Scenarios | | | | Average Acc. |
|---|---|---|---|---|---|---|---|---|---|
| | MathVista | MathVision | MathVerse | WeMath | MMStar | V* | POPE | SpaScore | |
| Qwen2.5-VL-3B | 62.3 | 21.2 | 31.2 | 21.1 | 53.1 | 73.5 | 86.1 | 16.2 | 45.6 |
| *Stage 1 SFT-based Models* | | | | | | | | | |
| AdaVaR-SFT-3B | 60.4 | 23.2 | 33.0 | 29.5 | 54.7 | 75.5 | 89.1 | 16.8 | 47.8 |
| GRD% | 31% | 5% | 1% | 78% | 90% | 98% | 94% | 96% | - |
| *Stage 2 RL-based Models* | | | | | | | | | |
| AdaVaR-3B | 69.8 | 24.5 | 35.2 | 33.8 | 59.3 | 77.1 | 88.2 | 18.9 | 50.8 |
| GRD% | 2% | 0% | 0% | 0% | 51% | 99% | 100% | 59% | - |
| *SFT-Guided Adaptive Reasoning Model* | | | | | | | | | |
| SFT-Adaptive-3B | 59.1 | 21.2 | 30.7 | 29.1 | 52.8 | 77.5 | 89.5 | 17.6 | 47.2 |
| GRD% | 86% | 90% | 59% | 100% | 100% | 100% | 100% | 100% | - |

The results in Table 6 show that both strategies—even with additional information—are outperformed by the selection made by AdaVaR. This indicates that: (i) mode selection at the individual sample level is necessary, especially for diverse queries in datasets like MMStar and in real-world applications; and (ii) our model is capable of making highly reasonable and effective mode choices.

## C.2    CAN THE ABILITY TO ADAPTIVELY SELECT MODES BE LEARNED VIA SFT?

To further validate the claim made in Section 1—that it is difficult to control the model's ability to choose among different modes during the SFT stage—and to demonstrate the necessity of the RL stage, we conduct an supplementary experiment. Without relying on RL at all, we train the model using two stages of SFT. The first stage is identical to AdaVaR Stage 1 and is used to learn the reasoning capabilities of different modes; the second stage is based on sampling from the model trained in Stage 1 to offline estimate the relative performance of the different modes.

Specifically, using the same RL data as AdaVaR, we estimate for each question with AdaVaR-SFT by running inference eight times with each of the two modes (temperature=1), estimating the accuracy of both modes, and retaining the rollouts of the mode with higher accuracy; if the accuracies are equal, we keep both. We then aggregate the reasoning trajectories corresponding to the estimated better mode(s) across all questions to serve as the SFT data for the second stage.

As presented in Table 7, SFT-Adaptive-3B indicates the SFT-guided adaptive reasoning baseline. It can be seen that SFT-Adaptive-3B exhibits a clear preference for the grounded mode; according to the evaluation results, this is unwarranted, because its performance on mathematical tasks and its overall performance are even worse than the model after Stage 1 (except on POPE and VStar, which are dominated by the grounded mode). A possible reason is that, as shown in Figure 3a, after

Table 8: Proportions of different response types from AdaVaR-3B across different datasets.

| Response Type | MathVista | MathVision | MathVerse | WeMath | MMStar | VStar | POPE | SpaScore |
|---|---|---|---|---|---|---|---|---|
| Correct | 69.80% | 24.50% | 35.20% | 33.80% | 59.30% | 77.10% | 88.20% | 18.90% |
| Selection Error | 4.80% | 6.30% | 7.20% | 9.10% | 8.80% | 2.60% | 2.90% | 10.50% |
| Reasoning Error | 25.40% | 69.20% | 57.60% | 57.10% | 31.90% | 20.30% | 8.90% | 70.60% |
| SE/(SE+RE) | 15.89% | 8.34% | 11.11% | 13.75% | 21.62% | 11.35% | 24.58% | 12.95% |

Table 9: Distribution of selection errors across different question types and selected modes.

| Selected Mode | Science | Perception | Math | Logical Reasoning | Instance Reasoning | Total |
|---|---|---|---|---|---|---|
| GRD Mode | 3 | 43 | 0 | 12 | 10 | 68 |
| TXT Mode | 33 | 0 | 14 | 12 | 12 | 71 |
| Total | 36 | 43 | 14 | 24 | 22 | 139 |

SFT the performances of the two modes have not yet stabilized (the text mode still has substantial room for improvement). Due to the limitations of the SFT framework, we can only perform offline sampling, which further reinforces this preference, as reflected in SFT-Adaptive-3B.

Overall, as an offline learning framework, SFT finds it is difficult to both strengthen different modes and guide mode selection, and it cannot correct unreasonable biases that emerge during iteration (and may even exacerbate them). In contrast, RL, as an online learning framework, uses dynamic sampling and estimation to effectively help the model enhance its reasoning ability and assess the relative merits of the modes. Therefore, the RL stage in the AdaVaR framework is necessary. However, as a promising future direction, it is worth exploring a multi-round SFT+RL iterative framework to further improve the model's capabilities.

## C.3 Error Analysis

First, we categorize errors into two types: (i) mode selection error (SE): the model fails to select the mode that could have produced the correct answer; (ii) reasoning error (RE): neither mode is able to produce the correct answer. As shown in Table 8, across all datasets, mode selection errors constitute only a small portion of the total errors (all less than 1/4).

Furthermore, we investigate the distribution of mode selection errors across different question types and image types in MMStar. Results in Table 9 and Table 10 show that: (1) most errors occur in domains where one mode has a clear advantage, such as perception tasks and natural images, where the grounded mode performs better, and science or math problems, where the text-based mode holds the advantage. (2) These exceptions illustrate that the complementarity between the two modes exists not only at the domain level, but also at the sample level. As a result, an optimal mode selection strategy cannot be predetermined and must be learned through the AdaGRPO algorithm.

Overall, our results indicate: (i) the context-adaptive mode selection in MoVT is reasonable, and (ii) our AdaVaR framework is capable of learning effective selection strategies, (iii) while still leaving room for further optimization.

## C.4 Analysis of the Computational Efficiency

### C.4.1 Training Efficiency: GRPO versus AdaGRPO

First, as stated in Section 3.3.1 and Appendix B.3, to ensure a fair comparison between GRPO, Ada-GRPO generates $n = 4$ rollouts per mode (8 per sample), while GRPO generates $2n = 8$ rollouts per sample. In this section, we verify the training efficiency both theoretically and empirically.

From a theoretical perspective, the main computational cost is incurred during rollout generation and the calculation of the training objective: (i) In rollout generation, simply generating rollouts separately for each mode would require two KV cache computations, leading to additional cost. However, in our design, the two modes share the same input sequence except for the mode-specific prefixes. By sharing the KV cache across modes, we achieve nearly the same computational cost as

Table 10: Distribution of selection errors across different image types and selected modes.

| Selected Mode | Natural Images | Diagram | Art | Geometry | Document | Others | Total |
|---|---|---|---|---|---|---|---|
| GRD Mode | 59 | 4 | 5 | 0 | 0 | 0 | 68 |
| TXT Mode | 8 | 46 | 6 | 6 | 4 | 1 | 71 |
| Total | 67 | 50 | 11 | 6 | 4 | 1 | 139 |

Table 11: Comparison between the training efficiency of GRPO and AdaGRPO. Theoretical FLOPs are estimated within a single optimization step: based on Qwen2.5-VL-3B, using an image of size 1024×768, a question of 260 tokens, and KV cache enabled. Empirical wall-clock time is measured by running 100 steps during the training process and averaging the results.

| Algorithm | Theoretical FLOPs | | | | | Empirical Wall-lock Time (seconds) | | | | |
|---|---|---|---|---|---|---|---|---|---|---|
| | Rollout Generation | Reward Calculation | Advantage Calculation | Objective Forward | Total | Rollout Generation | Reward Calculation | Advantage Calculation | Objective Forward | Total |
| AdaGRPO | 20.31T | 0 | 0 | 111.9T | 132.2T | 18.45 | 1.40E-02 | 2.20E-04 | 1.12 | 19.58 |
| GRPO | 20.47T | 0 | 0 | 113.9T | 134.4T | 19.18 | 1.20E-02 | 1.10E-03 | 1.17 | 20.36 |

GRPO. (ii) Regarding the training objective, both algorithms require forward passes of the policy model and reference model on 8 rollouts, so their computational costs are also nearly identical.

For empirical validation, we implement a FLOPS tracker using the deepspeed library and monitor the wall-clock time during training. As presented in Table 11, both theoretical computation and actual execution statistics show that the efficiency of AdaGRPO and GRPO is nearly identical (with approximately 4% additional overhead).

### C.4.2 INFERENCE EFFICIENCY

As for inference, theoretically, as described in Section 3.1, our AdaVaR model naturally integrates mode selection and reasoning into a single sequence generation step. During inference, unlike the training phase, there is no need to estimate mode-wise advantages. Therefore, similar to single-mode reasoning models, AdaVaR only requires a single call to the "model.generate()" method for each sample. Empirically, we compare the inference efficiency of AdaVaR and single-mode baselines in Table 12. As illustrated, the inference efficiency of different reasoning models is very similar on average, demonstrating that our framework does not introduce additional inference cost.

### C.5 TRAINING DYNAMICS

**Loss Curve in the SFT Stage**   Figure 9a shows the training loss during the SFT stage, demonstrating that the training process has already converged after one epoch.

**Rewards in the RL Stage**   Figure 8 shows the training dynamic of the two reasoning modes across different task during Stage 2 (the diverse mixture), illustrating the relative strengths and weakness between two modes. Based on the comparative performance of the two modes, we find that the text-based mode predominates on Math and Science, while the grounded mode excels on Detection and Grounding; for the other tasks, the two modes perform similarly. Overall, these results align with our expectations and further guide the learning of adaptive mode selection across tasks.

**Rollout Length in the RL Stage**   Figure 9b illustrates the change in rollout length for two modes during the RL process. It can be observed that the lengths of rollouts in two modes are relatively close, with a difference of less than 20 tokens.

### C.6 DISCUSSION ON SCALABILITY AND FUTURE WORK

Although this paper demonstrates that the adaptive reasoning paradigm MoVT is a feasible and effective solution for building general-purpose visual reasoning capabilities, our results further underscore its substantial potential. In this section, we discuss directions for future extension.

Table 12: Inference time (seconds) per sample across different models and datasets, measured with 8 A100 GPUs and accelerated by the vLLM framework (Kwon et al., 2023).

| Model | MathVista | MathVision | MathVerse | WeMath | MMStar | POPE | V* | SpaScore | Avg. |
|---|---|---|---|---|---|---|---|---|---|
| AdaVaR-3B | 0.32 | 0.95 | 0.45 | 0.48 | 0.18 | 0.07 | 0.25 | 0.87 | 0.36 |
| Text-SFT-RL | 0.32 | 0.91 | 0.44 | 0.53 | 0.21 | 0.11 | 0.23 | 0.74 | 0.37 |
| Grounded-SFT-RL | 0.31 | 0.85 | 0.39 | 0.4 | 0.25 | 0.07 | 0.27 | 0.84 | 0.34 |
| # samples | 1000 | 3040 | 3940 | 1740 | 1500 | 9000 | 191 | 1400 | - |

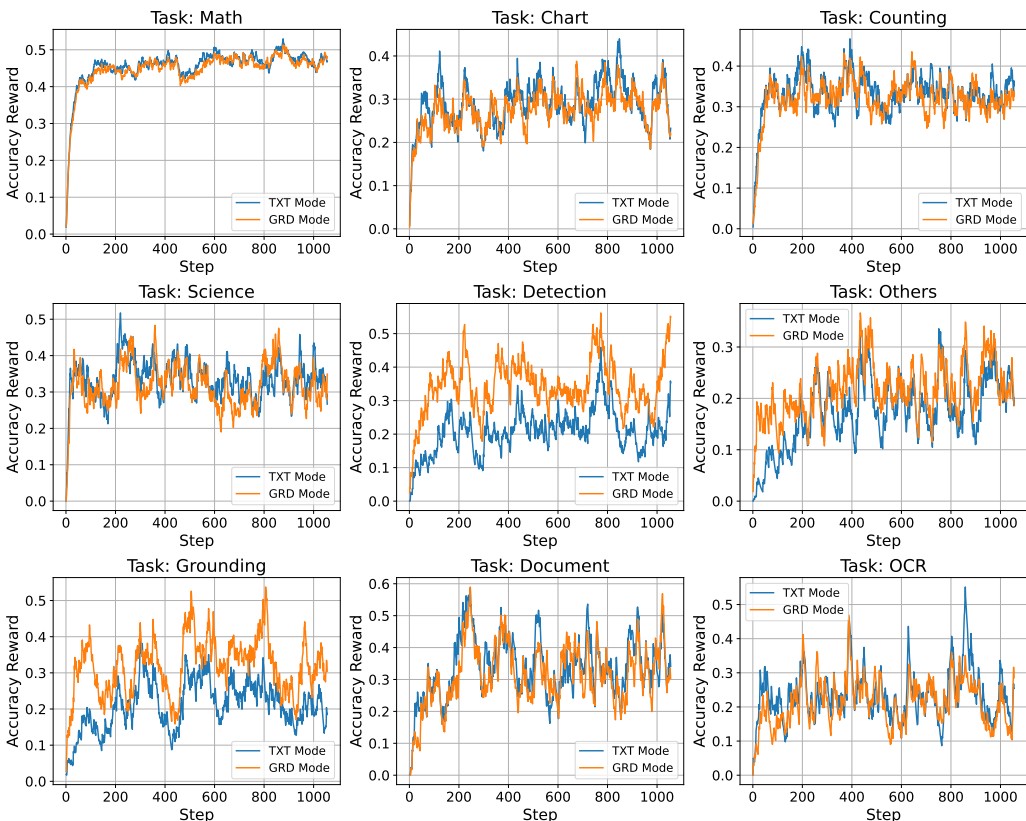

Figure 8: Accuracy reward curves for different modes across different tasks during RL.

### C.6.1 THE SCALABILITY OF MoVT TO MORE REASONING MODES

In the main paper, we instantiate MVoT with two mainstream reasoning modes: a *text-based* mode and a *grounded* mode. This design already validates the effectiveness of MoVT as a unified paradigm for adaptive mode selection. Then, to naturally extend MVoT to support richer reasoning patterns, we consider two complementary directions illustrated in Figure 10: **width extension** by adding more reasoning modes and **depth extension** by composing modes over multiple steps.

**Width Extension** Beyond the current text-based and grounded modes in Figure 10(a), MoVT can incorporate additional complementary modes by assigning each mode a dedicated prefix. For example, as shown in Figure 10(b), we may introduce a **tool-use mode**, where the model is encouraged to invoke external image-related tools before reasoning, e.g., `<tool> <think> ...</think> <answer> ...</answer>`; or a **no-think mode**, where the model directly produces an answer without explicit intermediate reasoning, e.g., `<answer> ...</answer>`. By constructing corresponding SFT data and performing RL to compare the new modes against existing ones under the MoVT framework, new reasoning skills can be seamlessly integrated into the model while preserving the unified mode-selection mechanism.

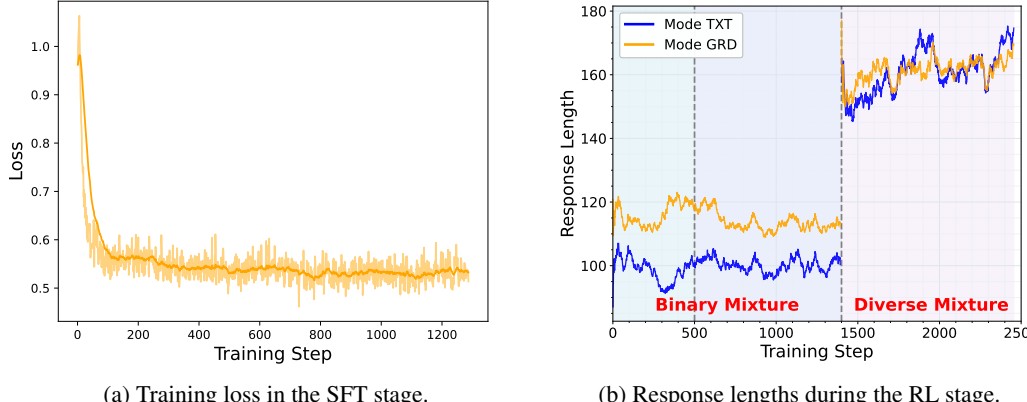

(a) Training loss in the SFT stage.

(b) Response lengths during the RL stage.

Figure 9: Supplementary training curves.

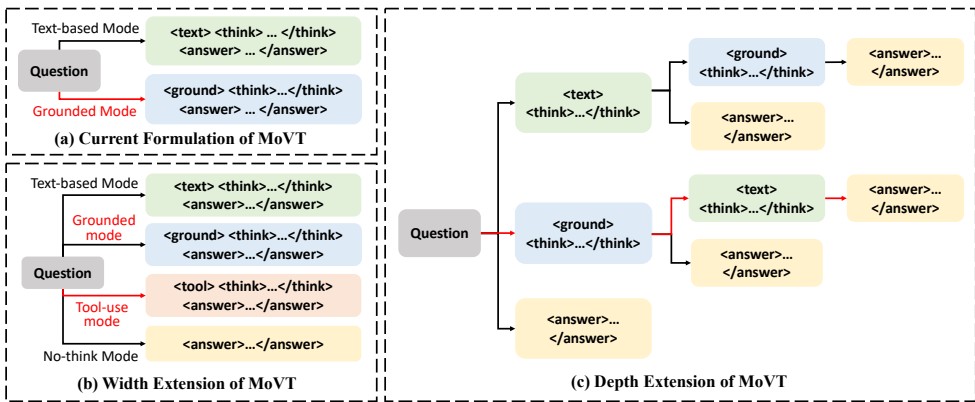

Figure 10: Extensions of MoVT to additional reasoning modes.

**Depth Extension**  MoVT can also be extended in depth by allowing multi-step mode selection and reasoning, as depicted in Figure 10(c). Concretely, at each step the model can either select a mode (e.g., text-based or grounded), performs reasoning within that mode or optionally decides to terminate by producing an answer based on the current information. This yields a flexible controller that can realize traditional single-mode reasoning as a special case but with more complex reasoning patterns constructed via sequential mixing and mode switching (e.g., grounded → text-based, or text-based → grounded). Under this formulation, inference and training can be further enhanced by search strategies and RL objectives that explicitly explore the space of multi-step mode sequences.

### C.6.2 THE SCALABILITY OF THE ADAVAR LEARNING FRAMEWORK

The AdaVaR learning framework can also be naturally adapted to the two extension directions introduced in Appendix. While the SFT extension only requires constructing the corresponding data, here we primarily discuss the extension strategy for the AdaGRPO algorithm.

**Scalability of Prefix-Guided Mode Exploration**  For width extension to k>2 modes, we can improve efficiency by selecting j<k modes to explore for each sample: (i) randomly selecting modes in the early training stages, and (ii) once the model develops certain mode selection capability, selecting the top-j modes based on the model's predicted probabilities. For depth extension, we can intervene after the end of each thought process (i.e., after the </think> token) by append all feasible prefixes to construct rollouts with hierarchical and thorough exploration.

Table 13: The impact of data proportions among different modes during the SFT stage. Each cell is presented in the format 'accuracy / proportion of selecting the grounded mode'.

| Model | SFT Ratio | MathVista | MathVision | MathVerse | WeMath | MMStar | V* | POPE | SpatialScore | Avg. Acc |
|---|---|---|---|---|---|---|---|---|---|---|
| Qwen2.5-VL-3B | - | 62.3 | 21.2 | 31.2 | 21.1 | 53.1 | 73.5 | 86.1 | 16.2 | 45.6 |
| SFT-based Models | | | | | | | | | | |
| AdaVaR-SFT-3B | GRD:TXT=1:1 | 60.4 / 31% | 23.2 / 5% | 33.0 / 1% | 29.5 / 78% | 54.7 / 90% | 75.5 / 98% | 89.1 / 94% | 16.8 / 96% | 47.8 |
| AdaVaR-SFT-3B | GRD:TXT=3:1 | 57.1 / 52 % | 21.5 / 10% | 31.2 / 3% | 29.4 / 84% | 51.4 / 96% | 75.3 / 100% | 89.6 / 100% | 17.0 / 100% | 46.6 |
| AdaVaR-SFT-3B | GRD:TXT=1:3 | 61.0 / 21% | 23.1 / 3% | 32.5 / 0% | 30.0 / 40% | 55.0 / 75% | 75.0 / 95% | 86.1 / 81% | 16.5 / 92% | 47.2 |
| RL-based Models | | | | | | | | | | |
| AdaVaR-3B | GRD:TXT=1:1 | 69.8 / 2% | 24.5 / 0% | 35.2 / 0% | 33.8 / 0% | 59.3 / 51% | 77.1 / 99% | 88.2 / 100% | 18.9 / 59% | 50.8 |
| AdaVaR-3B | GRD:TXT=3:1 | 66.8 / 6% | 24.1 / 1% | 34.8 / 0% | 32.4 / 2% | 57.7 / 64% | 76.8 / 100% | 88.9 / 100% | 17.4 / 80% | 49.9 |
| AdaVaR-3B | GRD:TXT=1:3 | 68.8 / 1% | 24.8 / 0% | 35.2 / 0% | 34.3 / 0% | 58.1 / 43% | 76.4 / 99% | 87.2 / 100% | 16.6 / 40% | 50.2 |

Table 14: The impact of mode prefix formatting. For AdaVaR models, each cell is presented in the format 'accuracy / proportion of selecting the grounded mode'.

| Model | Mode Prefix | MathVista | MathVision | MathVerse | WeMath | MMStar | V* | POPE | SpatialScore | Avg. Acc |
|---|---|---|---|---|---|---|---|---|---|---|
| Qwen2.5-VL-3B | - | 62.3 | 21.2 | 31.2 | 21.1 | 53.1 | 73.5 | 86.2 | 16.2 | 45.6 |
| AdaVaR-3B | <text>, <ground> | 69.8 / 2 % | 24.5 / 0% | 35.2 / 0% | 33.8 / 0% | 59.3 / 51% | 77.1 / 99% | 88.2 / 100% | 18.9 / 59% | 50.8 |
| AdaVaR-3B | <mode1>, <mode2> | 69.6 / 2 % | 23.8 / 0% | 35.0 / 0% | 33.5 / 0% | 59.5 / 45% | 77.5 / 100% | 89.0 / 100% | 17.6 / 55% | 50.7 |

**Advancing Mode Selection** As shown in Table 2 and Appendix C.3, although AdaVaR achieves substantial performance gains, there remains a gap to the upper bound of adaptive reasoning, indicating that the model's mode-selection capability can be further improved. In the future work, we can explore to construct richer datasets, or introduce a dedicated reasoning mechanism for mode selection itself—having the model first reason about which mode to use and then reason about the problem—to strengthen mode selection.

### C.6.3 EXTENSION TO WIDER DOMAINS

**Extension to Multilingual Context** As stated in Ethics Statement, following mainstream visual reasoning models, our work primarily focuses on English scenarios. The main limitation preventing thorough exploration of multilingual visual reasoning is the lack of high-quality data, both for training and evaluation. In future work, we aim to explore several key directions: (1) how to efficiently construct multilingual reasoning-oriented datasets; (2) the cross-lingual transferability and generalization of reasoning capabilities; and (3) how to develop adaptive reasoning methods for multilingual scenario, in order to extend MoVT and AdaVaR to a broader range of languages.

**Extension to Other Modalities** Our method primarily focuses on images. The improvements observed on video and multi-view 3D scenes in SpatialScore indicate that our approach can be extended to additional modalities. When dealing with diverse modalities—such as video, 3D data, and medical imaging—we believe adaptive reasoning will offer even greater value, as different modalities often require distinct reasoning modes. In future work, we aim to explore grounded reasoning methods in high-dimensional scenarios, expand the repertoire of reasoning modes, and investigate the transferability of these modes across different modalities.

### C.7 SUPPLEMENTARY ABLATION STUDY

### C.7.1 THE 1:1 DATA RATIO BETWEEN TWO MODES DURING SFT

As stated in Section 3.2, we chose the 1:1 ratio to avoid introducing a preference for any specific mode during the SFT stage. To validate this design, in Table 13, we experiment with different data ratios under the premise of keeping the total amount of data constant.

Several findings can be gleaned: (1) Higher proportion of a specific mode in the SFT data makes the model more inclined to select that mode: this tendency is particularly evident in the SFT model. After RL, different models converge toward similar trends, but the influence of the SFT stage still persists. (2) Such a preference may be effective on specific datasets (e.g., the preference for the text mode improves performance on WeMath); however, from the perspective of generalization (average performance), although all biased settings outperform the baseline, none perform as well as the 1:1 setting. (3) In addition, a major limitation of applying a skewed data ratio is that, in practical applications, facing various queries, it is difficult to know in advance which mode would be better.

Table 15: Ablation study on the SFT stage.

| Model | MathVista | MathVision | MathVerse | WeMath | MMStar | V* | POPE | SpaScore | Avg. Acc |
|---|---|---|---|---|---|---|---|---|---|
| Qwen2.5-VL-3B | 62.3 | 21.2 | 31.2 | 21.1 | 53.13 | 73.52 | 86.16 | 16.24 | 45.6 |
| AdaVaR-3B | 69.8 | 24.5 | 35.2 | 33.8 | 59.3 | 77.1 | 88.2 | 18.9 | 50.8 |
| w/o SFT | 63.2 | 21.7 | 31.3 | 26.9 | 54.1 | 73.2 | 82.4 | 18.6 | 46.5 |

Table 16: The impact of different system prompts. Each cell is presented in the format 'accuracy / proportion of selecting the grounded mode'.

| Model | train prompt | eval prompt | MathVista | MathVision | MathVerse | WeMath | MMStar | V* | POPE | SpaScore | Avg. Acc |
|---|---|---|---|---|---|---|---|---|---|---|---|
| Qwen2.5-VL-3B | - | - | 62.3 | 21.2 | 31.2 | 21.1 | 53.1 | 73.5 | 86.2 | 16.2 | 45.6 |
| AdaVaR-3B | full | full | 69.8 / 2% | 24.5 / 0% | 35.2 / 0% | 33.8 / 0% | 59.3 / 51% | 77.1 / 99% | 88.2 / 100% | 18.9 / 59% | 50.8 |
| AdaVaR-3B | full | think-only | 69.1 / 3 % | 24.3 / 0% | 36.1 / 0% | 33.0 / 1% | 57.8 / 59% | 76.4 / 100% | 88.6 / 100% | 18.6 / 62% | 50.5 |
| w/o full prompt | think-only | think-only | 69.5 / 2% | 25.1 / 0% | 36.1 / 0% | 33.2 / 0% | 58.4 / 52% | 77.0 / 100% | 88.3 / 100% | 18.5 / 59% | 50.8 |

In summary, although the RL phase helps reduce the sensitivity to the data in the SFT stage, an unbiased design between different modes is optimal for achieving superior general performance.

### C.7.2 THE FORMAT OF MODE PREFIXES

Currently, we utilize two mode prefixes with explicit semantics. To investigate the sensitivity to the format of prefixes, we conduct an ablation experiment, using semantically neutral prefixes, namely <mode1> and <mode2>. As presented in Table 14, models with different prefix formats show similar performance, indicating that, within our mode unification design, different prefix formats have minimal impact, and the model is able to learn distinct reasoning modes after SFT.

### C.7.3 THE EFFECTS OF SYSTEM PROMPT DESIGN

In our framework, we learn different reasoning modes during the SFT stage, where supervision primarily come from the SFT data, and the system prompt serves as an auxiliary guide. To verify the role of the system prompt, we conduct two ablation studies, aiming to address two questions: (1) Can different reasoning modes be elicited solely through the system prompt without relying on SFT data? and (2) Does the system prompt require detailed descriptions of the modes?

For the first question, we ablate AdaVaR-3B by skipping the SFT stage. Results in Table 15 show that: (i) Even provided with the description of the grounded mode, the model does not elicit the ability to perform grounded reasoning during the RL phase—only 0.03% of rollouts contain coordinates. (ii) In evaluations, the model shows no clear distinction between modes and fails to generate responses in the grounded reasoning format, achieving only a marginal improvement over Qwen2.5-VL-3B. Therefore, these results show that it is difficult to directly elicit multiple distinct reasoning modes merely by designing system prompts, which highlights the necessity of the SFT phase in our framework—regardless of whether it is applied to Qwen2.5-VL or other LVLMs.

To investigate the sensitivity to the system prompt, we experiment with two variants: *full*, which includes both the thinking instructions and the descriptions of the modes, and *think-only*, which removes the mode descriptions. Table 16 presents the results obtained using different system prompts during training and evaluation: (i) Our model, that is trained with the full prompt, can directly transfer to the think-only prompt and still maintain the capabilities of both reasoning modes, performing reasonable mode selection. (ii) When training with the think-only prompt, the model achieves performance nearly on par with that of the model trained using the full prompt. These results indicate that, after SFT, the model exhibits low sensitivity to the prompt. When extending to more modes, there is no need to design complex prompts, which enhances the scalability of our framework.

### C.7.4 DATA ORDER FOR CURRICULUM LEARNING

Results in Table 3 validate the effectiveness of the easy-to-hard data order. In this section, we investigate whether a hard-to-easy order works. As illustrated in Table 17, the model trained with a hard-to-easy data order consistently underperforms compared to the easy-to-hard setting, validating the rationality of our design.

Table 17: The impact of data order in curiculum learning.

| Model | Data Order | MathVista | MathVision | MathVerse | WeMath | MMStar | V* | POPE | SpaScore | Avg. Acc |
|---|---|---|---|---|---|---|---|---|---|---|
| AdaVaR-3B | easy-to-hard | 69.8 | 24.5 | 35.2 | 33.8 | 59.3 | 77.1 | 88.2 | 18.9 | 50.8 |
| AdaVaR-3B-R | hard-to-easy | 68.1 | 24.2 | 35.2 | 32.3 | 58.5 | 76.4 | 87.3 | 18.1 | 50.0 |

Table 18: Robustness against image noise. For AdaVaR, each cell is presented in the format 'accuracy / proportion of selecting the grounded mode'.

| Model | Gaussian Blur | MathVista | MathVision | MathVerse | WeMath | MMStar | V* | POPE | SpaScore | Avg. Acc |
|---|---|---|---|---|---|---|---|---|---|---|
| Qwen2.5-VL-3B | No | 62.3 | 21.2 | 31.2 | 21.1 | 53.1 | 73.5 | 86.1 | 16.2 | 45.6 |
| AdaVaR-3B | No | 69.8 / 2% | 24.5 / 0% | 35.2 / 0% | 33.8 / 0% | 59.3 / 51% | 77.1 / 99% | 88.2 / 100% | 18.9 / 59% | 50.8 |
| AdaVaR-3B | Yes | 68.8 / 2% | 24.3 / 0% | 35.1 / 0% | 33.7 / 0% | 58.3 / 45% | 75.8 / 99% | 88.2 / 98% | 18.5 / 54% | 50.5 |
| Text-SFT-RL | No | 67.2 | 23.52 | 36.1 | 32.19 | 56 | 70.7 | 85.6 | 18.8 | 48.8 |
| GRD-SFT-RL | No | 62.6 | 22.17 | 34.4 | 33 | 55.4 | 77 | 87.4 | 18.4 | 48.7 |

### C.7.5 ROBUSTNESS AGAINST IMAGE NOISE

To investigate whether the adaptive mode selection ability in AdaVaR is sensitive to image noises, we conduct an experiment by applying Gaussian blur to images during evaluation, we set the radius of blur to 0.5, in order to prevent excessive degradation of image information. The results are shown in Table 18, we find that: (1) introducing noise does not significantly affect mode selection; (2) the overall performance of AdaVaR shows only a slight decline; (3) even with added noise, AdaVaR-3B still maintains a clear advantage over baselines. These supports the robustness of our model.

### C.8 QUALITATIVE ANALYSIS

To illustrate the characteristics of different reasoning modes and how our AdaVaR model select the appropriate mode, we provide examples across multiple datasets, including image-related questions, the reasoning trajectories of different modes, and the model's mode selections.

**General Notation and Findings** In this section, all figures follow the same notation scheme: we underline the correct answers and visualize in the images the bounding boxes generated by the visually-grounded mode; different colors indicate the correspondence between the bounding boxes in the images and in the text; "TXT Mode" and "GRD Mode" are respectively short for the text-based mode and the visually-grounded mode. By examining cases across different datasets, we observe clear differences in the reasoning trajectories produced by the two modes, which also validates our discussion in Section 4.3.1: the two modes induce distinct reasoning patterns. In general, text-based reasoning is better at constructing complex, abstract logic, whereas the grounded mode tends to locate and search for key information and analyze it.

**MathVista** MathVista includes both math-oriented and general-scenario questions. As presented in Figure 11, we make several observations: (1) For more general scenarios—for example, the first case—the grounded mode, which places greater emphasis on visual information, performs better; (2) On tasks with relatively simple solution methods, such as the second case, the grounded mode also succeeds; (3) For problems requiring complex and abstract logic, like the third case, the textual mode is superior; (4) Moreover, when the grounded mode recognizes that there are no concepts requiring localization—for instance, the geometry problem in the fourth case—it also exhibits a certain degree of abstract reasoning ability.

**Math-Oriented Benchmarks** Figure 12 shows examples from three other math-oriented datasets. We observe a pattern: in these datasets, for nouns referring to abstract concepts (e.g., numbers), the grounded reasoning mode may invoke the "find" mechanism yet still cannot locate a concrete corresponding bounding box in the image. This corroborates our assumption that performing visual grounding is difficult in abstract scenarios. In such cases, the grounded mode can handle simple questions (e.g., the fourth case), but fails on those requiring step-by-step computation, such as the first case. Another observation is that the grounded mode is weak at numerical computation. For example, in the second and third cases it produces correct initial derivations but makes mistakes during the numerical calculations; in the second case, it even gets stuck in a repetitive generation loop and fails to solve the corresponding equation.

**V\*** V\* and POPE are representative of tasks that hinge on fine-grained object-related details. V\* in particular tests the model's ability to localize small objects. We provide several cases in Figure 13. As seen in the first two cases, text-based reasoning in such scenarios yields reasoning that lacks interpretability: without coordinates, users cannot tell whether the model actually localized the target object. Text-based reasoning is also prone to hallucinations; for example, in the second case the model produces incorrect information: it fails to find the bicycle and assumes in advance that the dog is to the left of the bicycle. In contrast, grounded reasoning can precisely locate the target object in such complex images, and the correct coordinates help the model answer the question correctly.

**POPE** Considering the hallucination task in POPE, the third case in Figure 13 shows that text-based reasoning mode struggles to locate the target object (especially when it is small, this limitation is also evident in V\*). Moreover, text-based reasoning can lead to overthinking—for example, the misunderstanding of a sandwich in the fourth case, and the introduction of irrelevant information and a repetitive generation loop in the fifth case. In contrast, the grounded mode thoroughly checks the objects in the image and provides correct answers.

**MMStar** As a benchmark including samples across different scenarios, We find that in many scenarios, both reasoning modes can arrive at the correct answer, so in Figure 14 we present several examples where the two modes differ. According to the first three examples, we can see that the text-based reasoning mode is better suited to tasks requiring complex abstract logic and knowledge—for instance, the folding problem in the first example and the painting-style knowledge in the second. The third example shows that the grounded mode tends to leverage information visible in the image, such as the colorless liquid, but cannot further infer additional information based on "acetone". The last two cases show that the grounded mode is better at capturing on attributes of specific objects and making comparisons between target objects.

**SpatialScore** Focusing on spatial reasoning in general scenes, Figure 15 provides several examples. The first case is a relatively common one from a third-person perspective (camera viewpoint). Both reasoning modes answer correctly, and the grounded mode further indicates the specific locations of the objects. The second case further requires reasoning about spatial relations from the first-person perspective of a specific object, which calls for some imagination. Thus, although the grounded mode correctly locates the corresponding object, it lacks further imaginative reasoning and does not answer correctly as the textual mode does. In addition, the third case reveals a current limitation of the grounded mode in multi-image scenarios: it cannot distinguish between different images using coordinates, which is a direction for future improvement. The fourth and fifth cases illustrate the grounded mode's advantage on more conventional spatial reasoning problems: by locating key objects and combining common sense, both ultimately yield correct reasoning outcomes.

**Extra Qualitative Results on MMStar** We provide two additional qualitative examples from the common-sense reasoning subset of MMStar in Figure 16. These cases require diverse focus: not only grounding on relevant objects in the image but also performing abstract common-sense reasoning. Rather than relying on a single mode, our model dynamically selects the most suitable mode for each question.

## MathVista - General

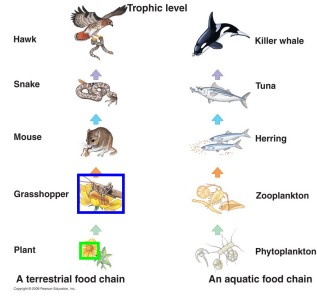

**Question:** Is the donut more than half eaten? **Choices:** (A) yes; (B) No.

**TXT Mode**

<think>To determine if the donut is more than half eaten, we need to visually assess the portion of the donut that has been consumed. The donut appears to have a significant portion missing, suggesting it is more than half eaten. The visible part of the donut shows a large gap, indicating that it is not a small bite or a small portion.</think> <answer>A</answer>

**GRD Mode**

<think> Find the donut **[0, 116, 392, 397]**. Check if the donut **[0, 116, 392, 397]** is more than half eaten. The answer is no. </think> <answer> B </answer> 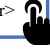

## MathVista - Diagram

**Question**: Which organism would be most affected if there was a shortage of plants? **Choices**: (A) Grasshopper; (B) Snake; (C) Mouse; (D) Hawk

**TXT Mode**

<think>To determine which organism would be most affected if there was a shortage of plants, we need to look at the food chain depicted in the image. Plants are at the base of the terrestrial food chain, and they are consumed by grasshoppers. If there is a shortage of plants, the primary consumers, which are the grasshoppers, would be most affected because they rely directly on plants for food. The options provided are: (A) Grasshopper (B) Snake (C) Mouse (D) Hawk Since the grasshopper is directly dependent on plants for food, a shortage of plants would most directly affect the grasshopper. </think> <answer>A</answer> 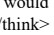

**GRD Mode**

<think> Find the plant **[252, 900, 328, 966]**. Check the organism that it connected to, got the organism **[244, 644, 422, 794]**. The grasshopper **[244, 644, 422, 794]** is connected to the plant **[252, 900, 328, 966]**. So the answer is grasshopper. </think> <answer> A </answer>

## MathVista - Logic

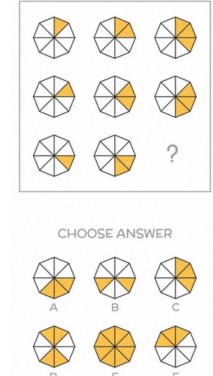

**Question**: Choose the answer for the missing picture. (GT Answer: A)

**TXT Mode**

<think> To solve this, let's observe the pattern in the given images. Each image is an octagon divided into eight equal parts, and the color pattern alternates between yellow and a different color. The question asks for the missing picture that continues this pattern.
- The first row has a pattern where the yellow part alternates with a different color.
- The second row follows the same pattern.
- The third row should continue this alternating pattern.
Looking at the options:- A has a yellow part in the top left. - B has a yellow part in the top left. - C has a yellow part in the top left. - D has a yellow part in the top left. - E has a yellow part in the top left. - F has a yellow part in the top left.
Since the pattern is consistent and the question asks for the missing picture, the correct answer should maintain the alternating pattern.</think> <answer>A</answer> 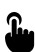

**GRD Mode**

<think> Find the pattern of the missing picture. The missing picture is D. So the answer is D. </think> <answer> D </answer>

## MathVista - Geometry

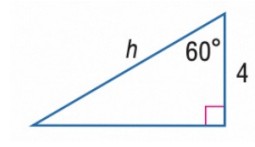

**Question**: Find $h$ in the triangle. **Choices**: (A) 4.62 (B) 5.66 (C) 6.93 (D) 8

**TXT Mode**

<think> To find the length of the hypotenuse $h$ in the right triangle, we can use trigonometric relationships. Given that one of the angles is $60^\circ$ and the adjacent side to this angle is 4, we can use the cosine function, which relates the adjacent side to the hypotenuse in a right triangle. The cosine of an angle is defined as the ratio of the length of the adjacent side to the length of the hypotenuse. Therefore, we have: $$\cos(60^\circ) = \frac{\text{adjacent}}{\text{hypotenuse}} = \frac{4}{h}$$ We know that $\cos(60^\circ) = 0.5$. So we can set up the equation: $$0.5 = \frac{4}{h}$$ To solve for $h$, we can rearrange the equation: $$h = \frac{4}{0.5} = 8$$ </think> <answer>D</answer>

**GRD Mode**

<think> In the right triangle, the side opposite the $60^\circ$ angle is the shorter leg, which is given as 4. The hypotenuse is labeled as 'h'. Using the properties of a 30-60-90 triangle, the ratio of the sides opposite the 30°, 60°, and 90° angles is $1 : \sqrt{3} : 2$. Therefore, the side opposite the 60° angle (which is 4) is half the hypotenuse. This relationship can be expressed as 4 = h/2. Solving for 'h' gives h = 4 * 2 = 8. </think> <answer> D </answer>

Figure 11: Cases from MathVista. The pointing-finger icon indicates the mode selected by AdaVaR.

**MathVision**

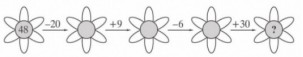

**Question:**
Which number do you have to write in the last daisy?
**GT Answer**: 61

**TXT Mode**
<think>First, we start with the number 48. We subtract 20 from it: 48 - 20 = 28. Then we add 9 to 28: 28 + 9 = 37. Next, we subtract 6 from 37: 37 - 6 = 31. Finally, we add 30 to 31: 31 + 30 = 61. So, the number we need to write in the last daisy is 61.</think> <answer>61</answer>

**GRD Mode**
<think> Find the number that is 30 more than 22. The answer is 52. </think> <answer> 52 </answer>

---

**MathVision**

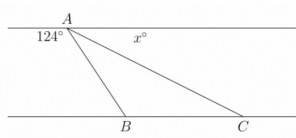

**Question**:
$\overline{BC}$ is parallel to the segment through $A$, and $AB = BC$. What is the number of degrees represented by $x$?
Please provide your final answer using a single number.
**GT Answer**: 28

**TXT Mode**
<think>Since AB = BC, angle BAC = angle ACB. The sum of angles in a triangle is 180 degrees. Therefore, angle BAC + angle ACB + angle ABC = 180 degrees. Given angle BAC = 124 degrees, we can substitute and solve for angle ACB. So, 124 + angle ACB + angle ACB = 180. Simplifying, 2 * angle ACB = 56. Therefore, angle ACB = 28 degrees. Since angle ACB = x, x = 28 degrees.</think> <answer>28</answer>

**GRD Mode**
<think> To find the value of $x$, we need to use the properties of parallel lines and the given angles. 1. We find that $\triangle ABC$ is isosceles with $AB = BC$. This means that the base angles $\angle BAC$ and $\angle ACB$ are equal. 2. The sum of the angles in a angle sum property:
$$124^\circ + x + x = 180^\circ$$
$$124^\circ + 2x = 180^\circ$$ $$2x = 56^\circ$$ $$x = 56^\circ / 2$$ $$x = 56^\circ / 2 = 56^\circ / 2 = 56^\circ / 2 = 56^\circ / 2 = 56^\circ / 2 = 56^\circ…$$(the model get stuck in a loop)

---

**MathVerse**

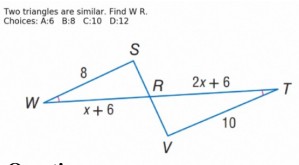

**Question**:
According to the question shown in the image, please first conduct reasoning, and then answer the question and provide the correct option letter, e.g., A, B, C, at the end.
**GT Answer**: B

**TXT Mode**
<think>Since the triangles are similar, the ratios of corresponding sides are equal. Therefore, we can set up the proportion $\frac{x+6}{2x+6} = \frac{8}{10}$. Solving this proportion will give us the value of $x$. Cross-multiplying gives: $$10(x + 6) = 8(2x + 6)$$ Expanding both sides: $$10x + 60 = 16x + 48$$ Rearranging terms to isolate $x$: $$60 - 48 = 16x - 10x$$ $$12 = 6x$$ $$x = 2$$ Now, substituting $x = 2$ into $x + 6$: $$x + 6 = 2 + 6 = 8$$ Therefore, $WR = 8$.</think> <answer>B</answer>

**GRD Mode**
<think> Since the two triangles are similar, the ratios of corresponding sides are equal. Therefore, we can set up the proportion $\frac{x+6}{2x+6} = \frac{8}{10}$. Solving this proportion for $x$, we get $x = 6$. Substituting $x = 6$ into $(2x + 6)$, we find $2(6) + 6 = 18$. Therefore, $WR = 18$. However, the question asks for $WR$, which is $(x + 6)$. Substituting $x = 6$ into $(x + 6)$, we get $(6 + 6 = 12)$. Therefore, the correct answer is D.</think> <answer>D</answer>

---

**WeMath**

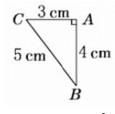

**Question**: As shown in the diagram, which of the following line segments is perpendicular to AB?
**Choices**: A. AB; B. AC; C. AD; D. No correct answer
**GT Answer**: B

**TXT Mode**
<think> In the diagram, it is given that angle C is a right angle, which means that AC is perpendicular to AB. Therefore, the line segment that is perpendicular to AB is AC.</think> <answer>B</answer>

**GRD Mode**
<think> Find the line segment that is perpendicular to AB. According to the diagram, AC is perpendicular to AB. Therefore, the answer is B. </think> <answer> B </answer>

Figure 12: Cases from other math-oriented benchmarks.

### V*-Visual Search

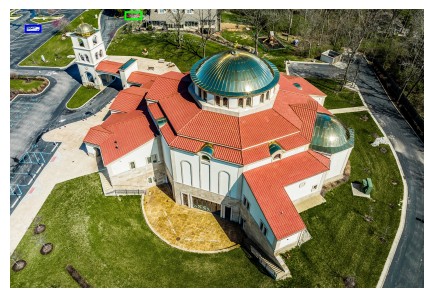

**Question:** Is the silver vehicle on the left or right side of the red vehicle? (A) right (B) left
**GT Answer:** B

**TXT Mode**

<think> The image shows a parking lot with vehicles. The silver vehicle is on the left side of the red vehicle. </think>
<answer>B</answer>

**GRD Mode**

<think> Find the red vehicle **[630, 22, 706, 50].** Think the car **[80, 86, 169, 128]** is to the left of or to the right of the vehicle **[630, 22, 706, 50].** The answer is left. </think>
<answer> B </answer>

### V*-Visual Search

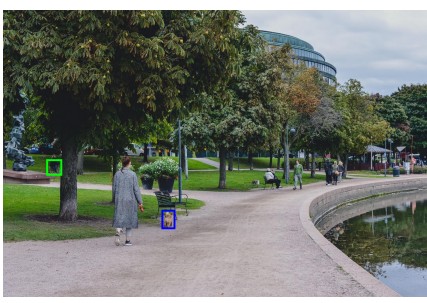

**Question:** Is the dog on the left or right side of the bicycle? (A) right (B) left
**GT Answer:** A

**TXT Mode**

<think> The image shows a person walking a dog on a path. The bicycle is not visible in the image, but based on the question, it is implied that the dog is on the left side of the bicycle if it were present. However, since the question is about the dog's position relative to the bicycle, and the image does not show a bicycle, we can only infer based on the question's context. The dog is clearly on the right side of the image, as it is walking away from the viewer and towards the right side of the image. </think>
<answer>B</answer>

**GRD Mode**

<think> Find the bicycle **[238, 796, 302, 862].** Think the dog **[847, 1062, 900, 1144]** is to the right of or to the left of the bicycle **[238, 796, 302, 862].** The answer is right. </think>
<answer> A </answer>

### POPE-Hallucination

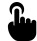

**Question:** Is there a car in the image? **GT Answer:** Yes

**TXT Mode**

<think>There is no car in the image. The image shows a red double-decker bus with people on top, and the background includes a street and some buildings. There are no vehicles or cars visible in the image.</think>
<answer>no</answer>

**GRD Mode**

<think> Find the car **[185, 248, 202, 258].** It exists. The answer is yes. </think>
<answer> yes </answer>

### POPE-Hallucination

**Question:** Is there a sandwich in the image? **GT Answer:** Yes

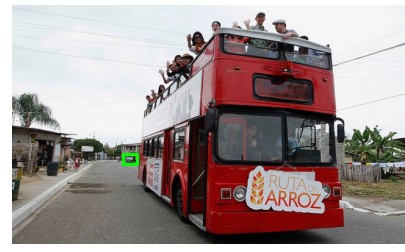

**TXT Mode**

<think>There is a hot dog in the image, which is a type of sandwich. However, the question asks if there is a sandwich in the image. The image shows a hot dog with toppings, but it is not a sandwich in the traditional sense. A sandwich is typically made with two or more slices of bread, and the hot dog is a single item. Therefore, the image does not contain a sandwich in the traditional sense.</think> <answer>no</answer>

**GRD Mode**

<think> Find the sandwich **[56, 113, 280, 460].** It exists. So the answer is yes. </think> <answer> yes </answer>

### POPE-Hallucination

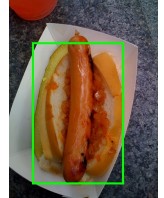

**Question:** Is there a car in the image?
**GT Answer:** No

**TXT Mode**

<think>There is no car in the image. The image is a photograph of a child drinking from a cup with orange slices on a table. The question is asking about the presence of a car, which is not present in the image. The image is a photograph of a child drinking from a cup with orange slices on a table. The question is asking about the presence of a car, which is not present in the image. The image is a photograph of a child drinking from a cup with orange slices on a table. …(model gets stuck in a loop)

**GRD Mode**

<think> Find the car. It doesn't exist. The question ask about 'is' relation. there is no car. So the answer is no. </think> <answer> no </answer>

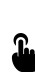

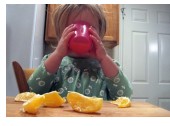

Figure 13: Cases from V* and POPE. Please zoom in for a better view of small objects in images.

## MMStar-Puzzle

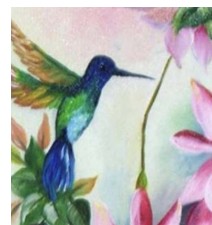

**Question:** Which cube is identical to the unfolded net? **GT Answer**: D

**TXT Mode**

<think>Let's analyze the unfolded net and the options to find the one that matches it. The unfolded net has the following sequence of numbers: 1, 2, 3, 4, 5, 6. Now, let's examine each option: A. This cube has the sequence 3, 5, 3, which does not match the unfolded net. B. This cube has the sequence 4, 1, 4, which does not match the unfolded net. C. This cube has the sequence 2, 9, 2, which does not match the unfolded net. D. This cube has the sequence 2, 3, 2, which does not match the unfolded net. E. This cube has the sequence 2, 5, 2, which does not match the unfolded net. None of the options match the unfolded net exactly. However, if we look closely at the sequence in the unfolded net, it seems to be a sequence of numbers that might be repeated or rearranged. The correct sequence in the unfolded net is 1, 2, 3, 4, 5, 6, which corresponds to option D. </think> <answer>D</answer>

**GRD Mode**

<think> Find the unfolded net **[118, 12, 303, 257]**. Check the cube that has the same pattern, got the cube **[151, 349, 269, 480]**. The cube **[151, 349, 269, 480]** is C. So the answer is c. </think> <answer> C </answer>

## MMStar-Image Style

**Question:** Identify the art style of this image?
**Options:** A: oil paint, B: vector art, C: Baroque, D: watercolor. **GT Answer**: Yes

**TXT Mode**

<think> The image appears to have a soft, blended quality that is characteristic of watercolor painting. The colors are not sharp and the brushstrokes are visible, which is typical of watercolor techniques. The vibrant colors and the way the light and shadow are depicted also suggest a watercolor style. Oil paint would have a more opaque and textured look, vector art would be a digital representation with clean lines and no blending, and Baroque art would have a more detailed and dynamic style with a focus on movement and texture. </think> <answer>D</answer>

**GRD Mode**

<think> It is a painting. So the answer is oil paint. </think> <answer> A </answer>

## MMStar-OCR & Knowledge

**Question:** The object shown in this figure? **Options**: A: Is a colorless liquid with a sweet, fruity odor, B: Has a boiling point of 56.05°C, C: Is used as a solvent for many organic compounds.

**TXT Mode**

<think> The image shows a bottle labeled "Acetone," which is a common solvent used in various applications. Acetone is known for its sweet, fruity odor and is used as a solvent for many organic compounds. It has a boiling point of 56.05°C. Therefore, the correct answer should reflect these properties. A: Is a colorless liquid with a sweet, fruity odor - This is true for acetone. B: Has a boiling point of 56.05°C - This is also true for acetone. C: Is used as a solvent for many organic compounds - This is true for acetone. D: None of these options are correct - This is incorrect because all the options are correct for acetone. </think> <answer>C</answer>

**GRD Mode**

<think> Find the bottle [192, 70, 399, 364]. The bottle [192, 70, 399, 364] is a colorless liquid [192, 70, 399, 364]. So the answer is colorless liquid. </think> <answer> A </answer>

## MMStar-Object Attribute

**Question:** What is the primary color of the football in the image?
**Options**: A: Maroon, B: Red, C: Brown, D: Oranges

**TXT Mode**

<think> The image shows a football in the hands of a player. The color of the football is clearly visible and distinct. The color of the football in the image is maroon. </think> <answer>A</answer>

**GRD Mode**

<think> Find the **football [364, 317, 390, 362]**. The football **[364, 317, 390, 362]** is brown. So the answer is brown. </think> <answer> C </answer>

## MMStar-Cross Instance Reasoning

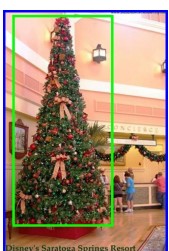

**Question:** Which object is bigger, the Christmas tree or the hotel lobby?
**Options**: A: They are about the same size, B: Hotel lobby, C: Christmas tree

**TXT Mode**

<think> The image shows a large Christmas tree in the foreground, which appears to be taller than the hotel lobby in the background. The hotel lobby is a room with a concierge desk and a sign, which is smaller in comparison to the Christmas tree. Therefore, the Christmas tree is bigger than the hotel lobby.</think> <answer>C</answer>

**GRD Mode**

<think> Find the hotel lobby **[0, 0, 335, 499]**. Find the Christmas tree **[19, 10, 224, 443]**. The question ask which one is bigger, the hotel lobby **[0, 0, 335, 499]** or the Christmas tree **[19, 10, 224, 443]**. The hotel lobby **[0, 0, 335, 499]** is bigger than the Christmas tree **[19, 10, 224, 443]**. So the answer is hotel lobby. </think> <answer> B </answer>

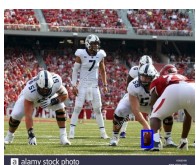

Figure 14: Cases from MMStar.

### SpatialScore-Spatial Relation (third person)

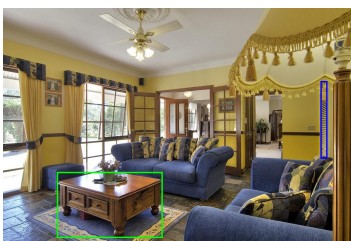

**Question:** Where is the blind located with respect to the coffee table? Select from the following choices. (A) left (B) right

**TXT Mode**

<think> The blind is located on the right side of the image, as it is on the right wall of the room. The coffee table is on the left side of the image. Therefore, the blind is to the right of the coffee table.</think> <answer>B</answer>

**GRD Mode**

<think> Find the coffee table **[186, 566, 552, 794]**. Think the blind **[1098, 252, 1122, 520]** is to the right of or to the left of the coffee table **[186, 566, 552, 794]**. The answer is right. </think> <answer> B </answer>

### SpatialScore-Spatial Relation (first person)

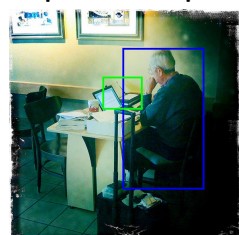

**Question:** If I stand at the person's position facing where it is facing, is the printer on the left or right of me? **Options:** (A) on the left; (B) on the right

**TXT Mode**

<think> The image shows a person sitting at a desk with a laptop and some papers. The printer is typically located on the right side of the desk when facing the person. Therefore, if you were standing at the person's position facing where it is facing, the printer would be on the right of you. </think> <answer> B </answer>

**GRD Mode**

<think> Find the person **[312, 107, 537, 496]**. Think the printer **[257, 186, 367, 276]** is to the left of or to the right of the person **[312, 107, 537, 496]**. The answer is left. </think> <answer> A </answer>

### SpatialScore-Cross Image Reference

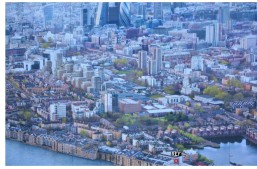 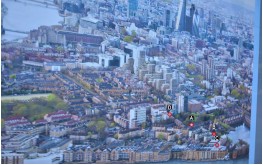

**Question:** A point is circled on the first image, labeled with REF. We change the camera position or lighting and shoot the second image. Which point on the second image corresponds to the point in the first image? **Options**: A. Point A; B. Point B; C. Point C

**TXT Mode**

<think> The question asks for the point on the second image that corresponds to the point labeled "REF" in the first image. Observing the images, the point labeled "REF" in the first image is located in the lower right section of the image. In the second image, the point labeled "REF" is also in the lower right section. Therefore, the point that corresponds to "REF" in the second image is Point C. </think> <answer>C</answer>

**GRD Mode**

<think> Find the point in the first image, labeled with REF. Think the point in the second image corresponds to it. The answer is B. </think> <answer> B </answer>

### SpatialScore-Spatial Relation (3D world)

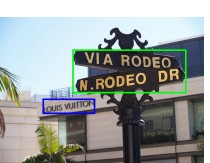

**Question:** Consider the real-world 3D locations of the objects. Which object has a higher location?
A "Lous Vuitton" plaque
B "VIA RODEO" sign

**TXT Mode**

<think> The question is asking for the location of the objects "Lous Vuitton" shop plaque and "VIA RODEO" sign. In the image, the "Lous Vuitton" shop plaque is located on the building, and the "VIA RODEO" sign is on a pole in front of the building. Since the question asks for the higher location, we need to compare the positions of these two objects. The "Lous Vuitton" shop plaque is on the building, which is a 3D location, and the "VIA RODEO" sign is on a pole, which is also a 3D location. However, the question implies a comparison of the relative heights or positions in a 2D sense, which is not directly applicable here. The question is likely asking for the sign in the image, which is a 2D representation of the sign. Since the question is about the sign in the image, and the sign is a 2D representation, we need to compare the signs in the image. The "VIA RODEO" sign is a 2D representation of the sign in the image, and it is higher than the "Lous Vuitton" shop plaque in the image. </think> <answer>(B)</answer>

**GRD Mode**

<think> Find the shop plaque **[134, 257, 300, 305]**. Find the sign **[228, 102, 588, 240]**. The question ask the object that has higher location. Compare the 3D locations of the two objects, the shop plaque **[134, 257, 300, 305]** is at higher location than the sign **[228, 102, 588, 240]**. So the answer is shop plaque. </think> <answer> A </answer>

### SpatialScore-Object Facing

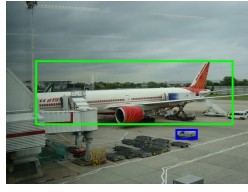

**Question:** Which side of the plain is the grey truck facing? **Options:** (A) the right side; (B) the left side

**TXT Mode**

<think> The image shows an airplane with its nose facing the left side of the image. The grey truck is positioned to the right of the airplane. Therefore, the truck is facing the right side of the airplane. </think> <answer> D </answer>

**GRD Mode**

<think> Find the grey truck **[440, 333, 497, 360]**. Think which side of the airplane **[76, 154, 590, 322]** is the truck **[440, 333, 497, 360]** facing. The answer is left. </think> <answer> B </answer>

Figure 15: Cases from SpatialScore.

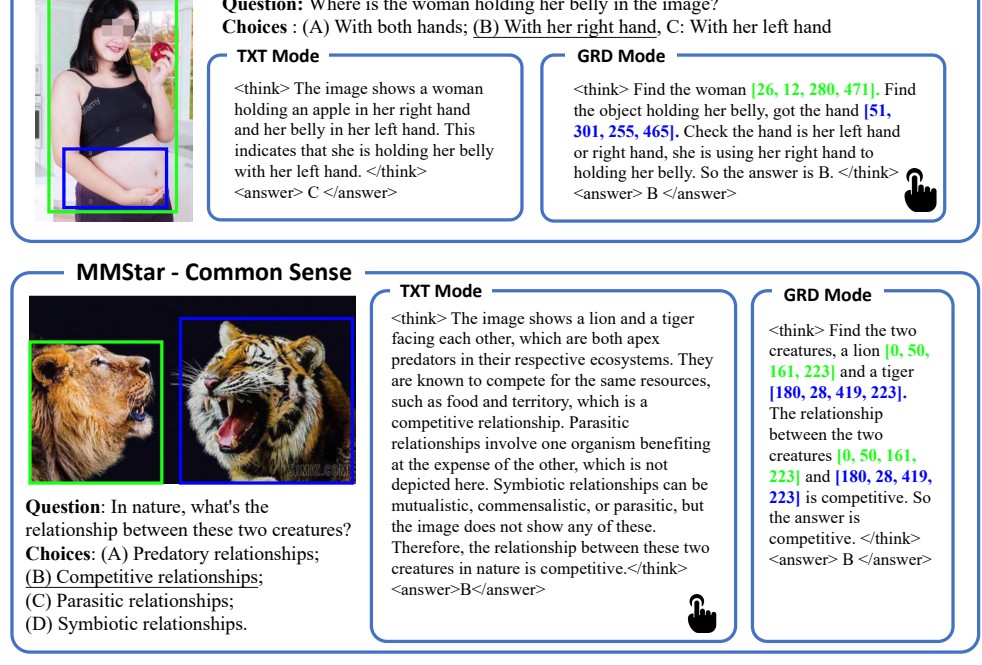

Figure 16: Cases from the common sense reasoning task in MMStar.

