# OpenReview forum: "Mixture-of-Visual-Thoughts: Exploring Context-Adaptive Reasoning Mode Selection for General Visual Reasoning"
_ICLR.cc/2026/Conference — ICLR 2026 Poster_

### Official Review · Reviewer_Yq9o · 2025-10-29

**Soundness:** 4
**Presentation:** 4
**Contribution:** 4
**Rating:** 8
**Confidence:** 3

**Summary:**

The paper proposes Mixture‑of‑Visual‑Thoughts (MoVT), an adaptive reasoning paradigm that unifies text‑based and visually‑grounded chains of thought within a single LVLM and learns to select the appropriate mode per instance. The approach is instantiated via AdaVaR, a two‑stage pipeline: (1) SFT that teaches both modes using mode‑prefix tokens (e.g., <text>, <ground>), followed by (2) RL with AdaGRPO, an extension of GRPO introducing prefix‑guided mode exploration, a mode‑relative advantage signal for selection, and a curriculum from simpler, clearer tasks to more diverse ones. Experiments on eight benchmarks (covering math, spatial reasoning, hallucination, visual search, and general perception) show consistent gains over strong specialists and base models; the AdaVaR‑7B variant reportedly surpasses GPT‑4o on average, with ablations and an “oracle upper bound” analysis illustrating the complementarity of modes and the value of adaptive selection.

**Strengths:**

- Clear, novel problem framing offering a compelling path toward generalist multimodal reasoning.

- AdaGRPO is purpose‑built for mode selection, combining (i) prefix‑guided exploration to avoid mode collapse, (ii) a mode‑relative advantage estimator to directly train the gating decision, and (iii) a curriculum that stabilizes learning before exposing the agent to noisy, diverse tasks.

- Strong empirical results and breadth. Evaluation on eight varied benchmarks demonstrates robust, cross‑domain improvements; ablations show each component’s necessity.

- Well‑presented and easy to follow.

**Weaknesses:**

- Scalability to k>2 reasoning modes (e.g. symbolic, tool‑augmented, variable CoT length) could be explored; naive exploration would scale rollout cost linearly with k and may become prohibitive without hierarchical or more efficient routing.

- The current setup enforces a single mode for the entire reasoning process, whereas many tasks might benefit from intra‑problem switching (e.g., ground first, then abstract reasoning).

- Prefix‑guided exploration requires n rollouts per mode (vs. 2n from a single policy in vanilla GRPO), increasing cost; quantitative training FLOPs, wall‑clock, and inference latency comparisons could be elaborated upon.

- The 1:1 SFT mix is reasonable but not properly analyzed; understanding sensitivity to data quality/quantity and skewed ratios (e.g., 3:1) would aid reproducibility and deployment.

- More systematic characterization of when mode selection fails (and why), interpretability of the gate’s decisions, and comparisons to simple ensembling would strengthen the paper.

**Questions:**

- How would AdaGRPO’s mode‑relative advantage generalize beyond two modes—pairwise comparisons, a shared latent score, or another efficient scheme? Any preliminary results or complexity analysis?

- What fraction of errors are due to wrong mode choice versus within‑mode reasoning errors? Are there identifiable categories (e.g., deceptive diagrams) where the gate is brittle?

- Have you tried the framework with non‑Qwen LVLMs? Any architectural prerequisites for effective selection?

- Can you discuss multi-step or long-context reasoning—can the model switch modes dynamically inside a single reasoning chain?

- There is limited interpretability of the mode selection. The paper does not sufficiently explain why the model chooses one mode over another beyond empirical observation

---

> ### Author Response · Authors · 2025-11-22
> **Rebuttal Response to Reviewer Yq9o (Part 1)**
>
> Dear Reviewer Yq9o,
>
> We are truly grateful for your detailed comments and constructive feedback. Below are our responses to the concerns you raised.
>
> ---
>
> **Weakness 1 & Question 1**: Extension to more reasoning modes.
>
> **Response**:
>
> Thank you for your suggestion. We discuss the scalability of our method from two perspectives:
>
> 1. Regarding the **framework design**, in our response to Common Question 1.2, we believe that **it is feasible for MoVT to incorporate more and more complex reasoning modes in the future**;
>
> 2. As for **technical implementation**, in our response to Common Question 2, **we address the scalability of the AdaVaR training framework**, including the rollout generation and mode-relative advantage computation aspects that you raised.
>
> Among these, we believe that the mode-relative advantage can be most straightforwardly extended by integrating scores from multiple pair-wise comparisons. As analyzed in our response to Common Question 5.1, **the computational cost of pair-wise advantage estimation is low** and does not significantly impact efficiency. Of course, we hope to explore more technical solutions in the future, such as the shared latent score as suggested.
>
> ---
>
> **Weakness 2 & Question 4**: Multi-step reasoning for intra-problem mode switching.
>
> **Response**:
>
> - First, as discussed in our response to Common Question 1.1, by expanding the model's exploration space, our framework ensures that **any reasoning patterns discoverable by the original model can still be explored**.
>   - This argument is supported by the observed complex reasoning patterns emerged in the text mode of AdaVaR, such as step-by-step reasoning (MathVista-Logic in Figure 7) and reflection (MMStar-Puzzle in Figure 10).
> - Secondly, we would like to **highlight the results of Mix-SFT-RL in Table 2**: for this baseline, we do not introduce intervention during RL to require the model to select a specific mode (so the training algorithm is purely GRPO). We find that, **without intervention, the likelihood of the model discovering more complex and effective reasoning patterns during rollout generation is very low**, as evidenced by its inferior performance.
> - However, as discussed in our responses to Common Question 1.2 and Common Question 2, **the scalability of the MoVT and AdaVaR frameworks enables us to explicitly introduce multi-step reasoning** and use prefix-guided intervention to induce more sophisticated patterns—such as sequentially employing two different reasoning modes within a single rollout.
>
> Overall, our method does not restrict the model from discovering complex patterns; on the contrary, the scalability of our approach provides a viable pathway for future work to explicitly guide the emergence of even more sophisticated reasoning patterns through targeted interventions, which we regard as an important direction for future exploration.
>
> ----
>
> **Question 2**: Systematic error analysis.
>
> **Response**:
>
> We sincerely appreciate your feedback. Please refer to our response to Common Question 4, where we have broken down the types of errors and analyzed the scenarios in which mode selection errors occur. In short, **across all datasets, the systematic error constitutes only a small portion of the total errors (less than 1/4)**.
>
> ---
>
> **Weakness 3**: Analysis of computational efficiency
>
> **Response**:
>
> Thank you for your suggestion. We have supplemented an analysis of computational efficiency during both training and inference phases. Please refer to our response to Common Question 5 for details. In short, **our framework introduces minimal additional overhead during both training and inference, and does not affect computational efficiency.**
>
> ----
>
> **Question 3**: Any architectural prerequisites for effective selection?
>
> **Response**:
>
> No, there are **no architectural prerequisites**.
>
> - As described in Section 3.1, we formulate both mode selection and reasoning as sequence generation tasks; therefore, **our method can be applied to any generative LVLMs** without requiring any special architectural designs.
>
> - We chose Qwen2.5-VL for two reasons: (i) Qwen2.5-VL achieves state-of-the-art performance among open-source LVLMs, making improvements on this strong baseline particularly meaningful; and (ii) as listed in Table 1, most current mainstream visual reasoning models are based on Qwen2.5-VL, so building AdaVaR upon it enables fair comparisons with existing approaches.
>
> We will also extend the MoVT paradigm to other LVLMs in our future work — thank you for the suggestion.

---

> ### Author Response · Authors · 2025-11-22
> **Rebuttal Response to Reviewer Yq9o (Part 2)**
>
> **Weakness 4**: The rationale behind the 1:1 mixing ratio in SFT data
>
> **Response**:
>
> As stated in Section 3.2 (Line 195), we **chose the 1:1 ratio** to **avoid introducing a preference for any specific mode** during the SFT stage. We **supplement an experiment to validate the effectiveness of this design**, please refer to our response to Common Question 6.
>
> ----
>
> **Question 5 & Weakness 5**: Interpretability and analysis of mode selection behaviors.
>
> **Response**:
>
> We have included a comprehensive analysis of mode selection behavior in our response to Common Question 3, including analyses based on question type (Common Question 3.1), image type (Common Question 3.2), and attention visualization (Common Question 3.3). Additionally, as suggested, we compare the mode selection strategy learned by AdaVaR with simple mode ensemble baselines (Common Question 3.4).
>
> Overall, the results demonstrate that **the design of selecting an appropriate mode for each question in MoVT is well justified, and the AdaVaR framework effectively enables the model to learn effective mode selection capabilities**.

---

> > ### Comment · Reviewer_Yq9o · 2025-11-25
> >
> > I thank the authors for the detailed response. Revised manuscript is much more clear now.

---

### Official Review · Reviewer_NdeY · 2025-10-31

**Soundness:** 3
**Presentation:** 3
**Contribution:** 3
**Rating:** 4
**Confidence:** 3

**Summary:**

This paper proposes a novel paradigm, Mixture-of-Visual-Thoughts (MoVT), and a corresponding two-stage training framework, AdaVaR, to build a general-purpose visual reasoning model. The core innovation lies in unifying different reasoning modes,specifically, text-based reasoning and visually-grounded reasoning within a single model and enabling it to adaptively select the appropriate mode based on the input context.Extensive experiments demonstrate that models trained with AdaVaR achieve consistent and superior performance across diverse benchmarks (mathematical, spatial, object-centric), outperforming specialized single-mode models and even surpassing GPT-4o on average accuracy.

**Strengths:**

1.The paper is well written and easy to follow
2.The paper introduces the  "Mixture-of-Visual-Thoughts" paradigm, which systematically integrates different visual reasoning modes within a unified model and enables context-adaptive mode selection.
3.The designed AdaVaR framework employs a logical two-stage training process. Through specific techniques such as mode prefixes, prefix-guided exploration, and mode-relative advantages, it successfully achieves multi-mode learning and adaptive selection. Experiments demonstrate consistent performance improvements.

**Weaknesses:**

1.The Prefix-guided Mode Exploration mechanism in AdaGRPO enforces a rigid, equal exploration of the two predefined reasoning modes (text-based and visually-grounded) for each input sample. While this hard constraint effectively prevents mode collapse, it introduces a fundamental limitation: the exploration space is artificially confined to a discrete set of predetermined patterns. I concern this design may prematurely restrict the model's policy search to this predefined mode space, thereby limiting its capacity to discover potentially superior strategies that fall outside these rigid categories. For instance, the optimal strategy for a given problem might involve a hybrid or sequential application of both modes, or even an entirely novel reasoning path not encapsulated by either predefined mode.
2.I think the classification of reasoning into strictly "text-based" and "visually-grounded" modes is conceptually limiting. we don't need to force the model to choose a single, rigid strategy at the start of its reasoning process, which is an unnatural constraint.

**Questions:**

See weakness.

---

> ### Author Response · Authors · 2025-11-22
> **Responses to Reviewer NdeY**
>
> **Weakness 1.1 & Weakness 2.1**: The classification of text-based and visually-grounded is limiting, and the exploration space is limited to predetermined patterns.
>
> **Response**:
>
> - We would like to clarify that **each of the two reasoning modes is not constrained to a rigid, predefined reasoning logic**. Instead, **each mode represents a sufficient reasoning space to explore**. Please refer to our response to Common Question 1.1 for a detailed discussion on the rationale behind our categorization and integration of text-based and grounded reasoning modes.
> - Overall, **our method expands the explorable reasoning space of the base model**. The effectiveness of our framework is well demonstrated in Section 4 of the paper, where consistent improvements over the base model and other single-mode reasoning baselines are demonstrated across multiple datasets.
> - In addition, as discussed in our response to Common Question 1.2, **the core contribution of this paper--the MoVT paradigm--is not limited to the current two modes and can be extended** by incorporating more and more sophisticated modes, thereby further expanding the model's reasoning space.
>
> ---
>
> **Weakness 1.2**: Limited capacity to discover more complex strategies like hybrid and sequencial application of both modes.
>
> **Response**:
>
> - First, as discussed in the previous question, by expanding the model's exploration space, **our framework ensures that any reasoning patterns discoverable by the original model can still be explored**.
>   - This argument is supported by the observed complex reasoning patterns emerged in the text mode of AdaVaR, such as step-by-step reasoning (MathVista-Logic in Figure 7) and reflection (MMStar-Puzzle in Figure 10).
> - Secondly, we would like to **highlight the results of Mix-SFT-RL in Table 2**: for this baseline, we do not introduce intervention during RL to require the model to select a specific mode (so the training algorithm is purely GRPO). We find that, **without intervention, the likelihood of the model discovering more complex and effective reasoning patterns during rollout generation is very low**, as evidenced by its inferior performance.
> - However, as discussed in our response to Common Question 1.2 and Common Question 2, **the scalability of the MoVT paradigm and AdaGRPO enables more sophisticated patterns**—such as using prefix-guided intervention to induce multi-step reasoning by sequentially employing two different reasoning modes within a single rollout.
>
>
> Overall, our method does not restrict the model from discovering complex patterns; on the contrary, the scalability of our approach provides a viable pathway for future work to explicitly guide the emergence of even more sophisticated reasoning patterns through targeted interventions, which we regard as an important direction for future exploration.
>
> ----
>
> **Weakness 2.2**: Rationale for enforcing the model to choose a single strategy.
>
> **Response**:
>
> The motivation behind the prefix-guided mode exploration mechanism—requiring the model to explore specific modes—is to **encourage thorough exploration within the expanded reasoning space**. To demonstrate its effectiveness, we would like to highlight two key comparative experiments in our paper (summarized in Rebuttal Table 8):
>
> * The ablation experiments on prefix-guided exploration (which is identical to GRPO) in Table 3 and Figure 4: results show that, without explicit guidance, the model collapses into predominantly exploring a specific mode, failing to sufficiently explore the full reasoning space;
>
> * Mix-SFT-RL in Table 2: When no intervention is made to distinguish or encourage exploration of different modes, the model—even after learning richer reasoning patterns during the SFT stage—still fails to autonomously explore thoroughly or acquire effective reasoning capabilities, resulting in performance even worse than that of the two single-mode baselines.
>
> Overall, our framework effectively expands the reasoning space for LVLMs to explore, while guiding models towards thorough exploration, ultimately achieving consistent and general improvement.
>
> ***Rebuttal Table 8: Comparison between models with different exploration space and exploration strategies.***
>
> | Model              | MathVista | MathVision | MathVerse | WeMath | MMStar | V*   | POPE  | SpatialScore | Avg. Acc |
> | ------------------ | --------- | ---------- | --------- | ------ | ------ | ---- | ----- | ------------ | -------- |
> | Qwen2.5-VL-3B      | 62.3      | 21.2       | 31.2      | 21.2   | 53.1   | 73.5 | 86.1  | 16.2         | 45.6     |
> | AdaVaR-3B      | 69.8      | 24.5       | 35.2      | 33.8   | 59.3   | 77.1 | 88.2  | 18.9         | 50.8     |
> | Mix-SFT-RL         | 65.0        | 23.2       | 33.6      | 28.8   | 56.2   | 76.9 | 85.4 | 18.6         | 48.5     |
> | AdaVaR-3B w/ GRPO | 66.3        | 23.8         | 34.9      | 31.3   | 56.6   | 75.4 | 89.1    | 19.7         | 49.6     |

---

> ### Author Response · Authors · 2025-11-27
> **Looking forward to your reply**
>
> Dear Reviewer NdeY,
>
> Greetings from the authors!
>
> With about one week remaining in the review period, we wanted to follow up briefly and thank you again for your thoughtful and constructive feedback. We hope that our responses and additional experiments have addressed your concerns.
>
> If there is any further clarification or detail that would support your assessment, we would be pleased to provide it.
>
> Kind regards,
>
> The Authors

---

> > ### Comment · Reviewer_NdeY · 2025-11-27
> >
> > Thank you for your reply.My major concerns have been solved, so I will raise my score.

---

### Official Review · Reviewer_n9fj · 2025-10-31

**Soundness:** 4
**Presentation:** 3
**Contribution:** 3
**Rating:** 4
**Confidence:** 5

**Summary:**

This paper addresses the limitation of existing visual reasoning models that focus on single reasoning modes and lack generalizability. It proposes a novel adaptive visual reasoning paradigm called Mixture-of-Visual-Thoughts (MoVT), which unifies two common reasoning modes (text-based and visually-grounded) into a single model and enables context-adaptive mode selection. To implement MoVT, the authors design the AdaVaR framework with a two-stage training process: a supervised cold-start stage (SFT) to unify and learn multiple reasoning modes, and a reinforcement learning (RL) stage using the proposed AdaGRPO algorithm to induce mode selection capability. Extensive experiments on 8 benchmarks (covering math-oriented and general scenarios) show that AdaVaR models (AdaVaR-3B and AdaVaR-7B) achieve consistent performance improvements, with AdaVaR-7B surpassing GPT-4o in average accuracy. The paper's main contributions include: proposing the MoVT paradigm for general visual reasoning, developing the two-stage AdaVaR framework, and validating the effectiveness of AdaGRPO and the overall method through experiments.

**Strengths:**

(1) The proposed MoVT paradigm addresses a critical gap in existing visual reasoning research by integrating complementary strengths of text-based and visually-grounded modes, breaking the domain limitation of single-mode models and providing a feasible solution for building general visual reasoning models.

(2) The two-stage AdaVaR framework (SFT+RL) is practically designed: the SFT stage solves mode unification via a uniform sequence format with mode-specific prefixes, while the RL stage (with AdaGRPO) targets mode selection, ensuring the model both learns multiple modes and adapts to context.

(3) The experimental design is relatively comprehensive: the paper evaluates on diverse benchmarks (math, visual search, hallucination control, etc.), compares with multiple single-mode baselines, and conducts ablation studies on key components (Ada-Adv, PG-Exp), providing sufficient evidence for the method's effectiveness.

**Weaknesses:**

(1) In the SFT stage of AdaVaR, the paper sets the data mixing ratio of text-based and visually-grounded reasoning to 1:1 but provides no justification for this ratio. It neither explains why 1:1 is superior to other ratios (e.g., 2:1, 1:2) nor conducts ablation experiments to verify whether different ratios affect mode learning or subsequent RL-based mode selection. This unsubstantiated setting makes it unclear whether the ratio is optimal or merely an empirical choice, reducing the methodological rigor.

(2) The paper mentions that the KL penalty term is omitted in the AdaGRPO formula (Section 3.3.1) but uses a KL penalty with a coefficient of 0.04 in training (Appendix A.3). However, it fails to explain the basis for choosing the 0.04 coefficient—for example, whether it was determined through experimental tuning (e.g., testing coefficients like 0.01, 0.02, 0.05) or based on prior work, and how different KL coefficients influence the model's mode selection bias and reasoning accuracy. The lack of this information impairs the reproducibility of the algorithm.

(3) The unified reasoning format uses <text> and <ground> as mode prefixes (Section 3.1), but the paper does not explore the impact of prefix design on model performance. It neither tests whether longer prefixes (e.g., <text-based-reasoning>) or more distinguishable prefixes (e.g., <txt> vs. <grd>) affect the model's ability to distinguish modes nor verifies whether prefixes introduce unintended inductive biases (e.g., <ground> being more salient than <text>, leading the model to prefer it). This oversight leaves potential factors influencing mode selection unaddressed.

(4) In the evaluation benchmarks (Section 4.1), the paper omits several widely used visual reasoning benchmarks, such as CLEVR (for compositional visual reasoning) and GQA (for structured visual question answering). Excluding these benchmarks makes it impossible to evaluate the model's performance on compositional or structured reasoning tasks, limiting the generalizability of its claim to "general visual reasoning."

(5) For some comparison models (e.g., Qwen2.5-VL-3B, LMM-R1), the paper marks results with "*" indicating reproduction but provides no details on the reproduction process. It does not specify whether the reproduction strictly followed the original paper's training settings (e.g., learning rate, batch size), whether there were discrepancies between reproduced results and original results, or how such discrepancies were resolved. This opacity raises concerns about the fairness of comparisons between AdaVaR and these baseline models.

(6) The mode-switching mechanism (Section 4.1) is described as "switching to another mode if the model gets stuck in repetitive logic," but the paper provides no quantitative definition of "getting stuck in repetitive logic." It neither clarifies how many consecutive repeated tokens or logical loops constitute "being stuck" (e.g., 3 repetitions vs. 5 repetitions) nor verifies through ablation experiments whether this mechanism actually improves performance (e.g., comparing accuracy with and without mode switching). The lack of clear criteria makes the mechanism vague and difficult to reproduce.

(7) In qualitative analysis cases (e.g., Figure 7: MathVista-General), the visually-grounded (GRD) mode incorrectly answers the donut question ("No") while the text-based (TXT) mode answers correctly, but the paper does not analyze the root cause of the GRD mode's error. It fails to clarify whether the error stems from incorrect bounding box localization, flaws in visual-information-based reasoning, or mode selection bias, weakening the depth of the qualitative analysis.

(8) The Ethics Statement claims all data sources are open-source but provides no specific proof of authorization for using VoCoT (Li et al., 2025) data. It neither cites VoCoT's license type (e.g., MIT, CC BY-NC) nor confirms that data reuse complies with VoCoT's terms of use. This omission raises concerns about potential copyright issues with the data.

(9) The Reproducibility Statement mentions that code, models, and data will be open-sourced after review, but the current version provides no preprint link, temporary repository, or sample data. This prevents reviewers from verifying key implementations (e.g., advantage calculation in AdaGRPO) or preliminary results, hindering the assessment of reproducibility.

(10) The paper provides no theoretical analysis of AdaGRPO's convergence. It does not prove whether the algorithm can converge to a stable policy, how the three key components (e.g., prefix-guided exploration) affect convergence speed, or under what conditions the algorithm can avoid mode collapse (e.g., always selecting one mode). The lack of theoretical guarantees makes the algorithm's stability during long-term training questionable.

(11) The paper lists extending MoVT to more reasoning modes (e.g., direct answering) as a future direction (Section B.4) but provides no preliminary experimental evidence. It neither tests whether AdaVaR can still learn mode selection after adding a <direct-answer> mode nor analyzes whether increasing the number of modes exacerbates exploration difficulty or reduces performance. This leaves MoVT's claimed scalability unsupported.

(12) The paper does not discuss the model's training efficiency. For example, AdaVaR-7B is trained using 32 NVIDIA A100 GPUs for 29 hours (Table 4), but it does not compare this training cost (e.g., GPU hours, energy consumption) with other models (e.g., GPT-4o, Qwen2.5-VL-7B). This omission makes it impossible to evaluate whether AdaVaR's performance improvements are worth the increased computational cost, limiting its practical application value.

(13) The paper does not mention the inference latency introduced by adaptive mode selection. It neither measures whether the mode selection process (generating prefixes, evaluating mode advantages) increases inference time compared to single-mode models nor discusses optimization strategies to reduce latency (e.g., early mode selection). This is a critical oversight for practical applications (e.g., edge devices with latency constraints).

(14) In RL data filtering (Section 3.3.2), the paper removes "easy questions that Qwen2.5-VL-7B can answer correctly in 8 random samples" but provides no basis for choosing 8 as the threshold. It neither tests whether other thresholds (e.g., 5, 10) retain more useful data or reduce noise nor analyzes the distribution (e.g., task type, difficulty) of retained and removed data to confirm whether filtering improves data quality.

(15) For SFT data, the paper uses GPT-4o to rewrite a subset of VoCoT data into multiple-choice format (Appendix A.2) but does not evaluate the quality of the rewritten data. It neither conducts manual verification to check whether GPT-4o introduces errors (e.g., incorrect options) or biases (e.g., overly simple options) nor compares model performance trained on original vs. rewritten data to verify the impact of rewriting.

(16) The experimental results lack statistical significance analysis. When reporting accuracy values (e.g., Table 1), the paper provides no error bars, confidence intervals, or p-values. For example, it claims AdaVaR-7B's average accuracy surpasses GPT-4o (55.82 vs. 53.20) but does not confirm whether this difference is statistically significant (e.g., via t-test). This makes it impossible to determine whether performance improvements stem from the method itself or random fluctuations.

(17) The paper reports GRD% (proportion of choosing the grounded mode) across datasets (Table 2) but does not analyze why the model strongly prefers the grounded mode in some datasets (e.g., POPE: 100% GRD%). It neither rules out overfitting to the POPE data distribution (e.g., all POPE tasks require object localization, leading the model to always select GRD) nor confirms whether this preference is true context adaptation rather than dataset-specific bias.

(18) The paper notes that AdaVaR-7B (7B parameters) is more inclined to select the text-based mode for knowledge-related categories (e.g., geography) compared to AdaVaR-3B (3B parameters) (Section B.1) but does not analyze the root cause of this difference. It neither explores whether the 7B model's richer pre-trained knowledge reduces its reliance on visual grounding nor verifies whether the RL process for the 7B model is more likely to prioritize text-based reasoning, leaving the impact of model size on mode selection unaddressed.

(19) The ablation experiment (Table 3) tests the impact of Ada-Adv, PG-Exp, diverse mixed data, and curriculum learning but omits ablation of the "mode-specific prefix" (a key component of mode unification in Section 3.1). It does not compare model performance with and without prefixes to confirm whether prefixes are essential for mode differentiation, weakening the completeness of ablation analysis.

(20) The paper does not provide details on how multi-modal data (images+text questions) is processed in the SFT and RL stages. It neither clarifies whether images are preprocessed identically across modes (e.g., resolution, feature extraction) nor explains how the model fuses visual and textual information during mode-specific reasoning (e.g., whether visual features are weighted differently in GRD vs. TXT modes), leading to ambiguity in the multi-modal integration process.

(21) The paper does not evaluate the model's robustness to image noise or occlusion. It neither tests whether AdaVaR's mode selection adapts appropriately when input images contain noise (e.g., Gaussian blur) or object occlusion (e.g., a partially covered donut) nor compares robustness with single-mode models. This is critical for assessing the model's practicality in real-world scenarios where images are often imperfect.

(22) The paper does not conduct user studies to evaluate the readability or utility of generated reasoning processes. It neither assesses whether humans can easily understand the GRD mode's bounding box-integrated reasoning or the TXT mode's textual logic nor analyzes whether reasoning processes enhance trust in the model's answers (a key factor for real-world adoption), limiting the assessment of the method's practical value.

(23) The reference list contains inconsistent formatting and incomplete entries. For example, OpenAI (2025) (Section 1.10) lacks authors and a full title (only "Thinking with images" and a URL), while some entries (e.g., DeepSeek-AI et al., 2025) have overly long author lists that could be abbreviated. This reduces the paper's professionalism and makes it difficult for readers to locate and verify cited works.

(24) The optimization objective formula of AdaGRPO (Section 3.3.1, Equation in Section 3.3.1) contains a typo: the denominator repeatedly includes \(\pi_{\theta_{old}}(o_{j}|i,q,o_{j,<t})\), causing confusion about the correct formula. The paper does not correct this typo or provide a clear derivation, which may mislead readers and hinder implementation.

(25) Algorithm 1 (Appendix A.3) for mode-relative advantage calculation uses \(r[1..2n]\) but does not specify the order of rewards (e.g., whether \(r[1..n]\) corresponds to text-based or visually-grounded rollouts). This ambiguity could lead to incorrect implementation of the advantage calculation, undermining reproducibility.

(26) The paper initializes AdaVaR based on Qwen2.5-VL (Section 3.2) but does not specify the exact version (e.g., Qwen2.5-VL-Instruct vs. base Qwen2.5-VL) or confirm whether the pre-training data of Qwen2.5-VL is consistent across all compared models. If AdaVaR uses a more capable pre-trained base model (e.g., the Instruct version with better initial reasoning), performance gains could stem from the base model rather than the proposed framework, distorting the evaluation of the method's effectiveness.

(27) The paper does not evaluate the model's performance under low-resource conditions. It neither tests how AdaVaR performs when training data is reduced (e.g., 50% of the original data) nor compares it with single-mode models to determine whether adaptive reasoning maintains its advantage in data-scarce scenarios. This limits the assessment of the method's data efficiency.

(28) The paper calculates average accuracy across benchmarks (Table 1) but does not specify the weighting scheme. It neither states whether all benchmarks are weighted equally nor whether weighting is based on dataset size, nor analyzes how different weighting schemes affect average accuracy (e.g., if MathVista has more samples, size-based weighting could inflate the average). This ambiguity makes fair comparison of AdaVaR's overall performance with other models difficult.

(29) The SFT stage uses 1 training epoch (Table 4), but the paper provides no justification for this choice. It neither tests whether more epochs (e.g., 2, 3) improve mode learning or cause overfitting nor analyzes the SFT stage's training loss curve to confirm that 1 epoch is sufficient for convergence.

(30) The RL stage generates 8 rollouts per sample (4 text-based, 4 visually-grounded) (Appendix A.3), but the paper does not explain why 8 is chosen. It neither tests other rollout counts (e.g., 4, 16) to see if more rollouts improve mode comparison accuracy nor checks if fewer rollouts reduce computational cost without performance loss.

(31) The paper provides training dynamics for math tasks (Figure 3) but only partial curves for other tasks (Figure 6). It does not present detailed training dynamics (e.g., accuracy, reward, GRD% over steps) for all benchmark categories (e.g., POPE, V*), making it difficult to fully understand how the model learns mode selection across different task types.

(32) In qualitative analysis cases (e.g., Figure 7: MathVista-Geometry), the paper does not provide clear visual details of input images (e.g., the side lengths of the right triangle). Reviewers cannot verify whether the model's reasoning (e.g., using cosine or 30-60-90 triangle properties) is correct, weakening the credibility of qualitative results.

(33) The paper uses "reasoning mode" and "chain-of-thought (CoT)" interchangeably (Section 1.10) but does not clearly define their relationship. It neither distinguishes whether a "reasoning mode" is a type of CoT nor clarifies if it is a broader concept, leading to confusion with existing CoT literature (e.g., whether MoVT's modes are compatible with other CoT variants).

(34) The paper does not evaluate zero-shot generalization to unseen tasks. It neither tests AdaVaR on visual reasoning tasks not included in training (e.g., video-based reasoning, medical image reasoning) nor verifies whether it can adaptively select modes for new tasks, which is essential for validating the "general visual reasoning" claim.

(35) The ablation experiment (Table 3) does not test the impact of data order in curriculum learning (e.g., training on diverse mixture first, then binary mixture). It does not confirm whether the "easy-to-hard" order is truly optimal or if other orders (e.g., hard-to-easy) yield better performance, limiting the analysis of curriculum learning's effectiveness.

(36) The paper does not explain how the model makes mode selection decisions for individual samples. It neither provides a breakdown of the factors influencing mode selection (e.g., question complexity, image content richness) nor visualizes the decision process (e.g., attention weights on mode prefixes), making the mode selection mechanism opaque.

(37) The paper does not discuss the model's adaptation to different image types (e.g., natural images, diagrams, documents). It neither tests whether AdaVaR adjusts mode selection appropriately for different image types (e.g., preferring TXT for diagrams, GRD for natural images) nor compares performance across image types, limiting the assessment of its generalizability across visual domains.

(38) The paper does not control for the length of reasoning processes across modes. It neither ensures that reasoning processes in TXT and GRD modes have similar lengths (to avoid performance differences due to longer reasoning) nor analyzes how reasoning length affects accuracy, leading to potential confounding factors in performance comparisons.

(39) The paper does not explore the trade-off between model size and performance. It neither tests whether smaller models (e.g., AdaVaR-1B) can still benefit from adaptive reasoning nor analyzes whether the performance gain of AdaVaR-7B over AdaVaR-3B is proportional to the increase in parameters, limiting the understanding of the method's scalability with model size.

(40) The paper does not address the model's performance in cross-lingual visual reasoning. It only focuses on English scenarios (Ethics Statement) and does not test whether AdaVaR can adapt mode selection for non-English questions (e.g., Chinese, Spanish) nor compare cross-lingual performance with single-mode models, limiting its applicability in multilingual contexts.

**Questions:**

*To facilitate discussions during the Rebuttal phase, authors are advised to respond point-by-point (indicating the question number).*

(1) For the 1:1 data mixing ratio in the SFT stage, could you provide ablation experimental results comparing different ratios (e.g., 2:1, 1:2, 3:1) and explain why 1:1 is optimal for mode unification and subsequent RL training?

(2) Regarding the KL penalty coefficient of 0.04 in the RL stage, could you detail the tuning process (e.g., tested coefficient values, performance changes with different coefficients) and explain how you determined that 0.04 balances mode selection bias and reasoning accuracy?

(3) For the mode-switching mechanism, could you provide a quantitative definition of "getting stuck in repetitive logic" (e.g., number of consecutive repeated tokens, threshold for logical loops) and present ablation results comparing model accuracy with and without this mechanism?

(4) For reproduced results of baseline models (marked with "*"), could you provide detailed training settings (e.g., learning rate, batch size, number of epochs) and compare them with the original papers? If there were discrepancies between reproduced and original results, how did you resolve them?

(5) In the MathVista-General case (Figure 7), the GRD mode incorrectly answered the donut question. Could you analyze the root cause (e.g., bounding box localization error, flawed visual reasoning) and provide evidence (e.g., localization results, intermediate reasoning steps)?

(6) In the Ethics Statement, you claim to use open-source data from VoCoT (Li et al., 2025). Could you provide specific proof of authorization (e.g., VoCoT's license text, confirmation of compliance with terms) to address potential copyright concerns?

(7) In the Reproducibility Statement, you mention open-sourcing code, models, and data after review. Could you provide a clear timeline (e.g., date of release after acceptance) and a temporary link to sample data or a preprint repository to allow reviewers to verify key implementations?

(8) Regarding AdaGRPO's convergence, could you supplement theoretical analysis (e.g., proof of convergence to a stable policy) or empirical evidence (e.g., convergence curves of policy loss over training steps) to confirm the algorithm's stability?

(9) For extending MoVT to more reasoning modes (e.g., direct answering), could you provide preliminary experimental results (e.g., performance after adding a <direct-answer> mode) to support the claim of scalability?

(10) Could you compare the training cost (e.g., GPU hours, energy consumption) of AdaVaR with other models (e.g., GPT-4o, Qwen2.5-VL-7B) and explain whether the performance improvement justifies the increased computational cost?

(11) Could you measure the inference latency of AdaVaR (including mode selection time) and compare it with single-mode models? Additionally, do you have optimization strategies (e.g., early mode selection) to reduce latency for edge device applications?

(12) For the RL data filtering threshold of 8 (Qwen2.5-VL-7B answering correctly in 8 random samples), could you conduct a sensitivity analysis (e.g., testing thresholds 5, 10) and analyze how different thresholds affect data distribution and model performance?

(13) For SFT data rewritten by GPT-4o into multiple-choice format, could you provide results of manual quality assessment (e.g., error rate of options, bias in difficulty) and compare model performance trained on original vs. rewritten data?

(14) Could you supplement statistical significance analysis for experimental results (e.g., p-values via t-test, confidence intervals) to confirm that AdaVaR's performance improvements over baselines are not due to random fluctuations?

(15) For the 100% GRD% in POPE, could you analyze the dataset distribution (e.g., proportion of tasks requiring object localization) and conduct experiments on a modified POPE subset (with non-localization tasks) to verify whether the preference is context-adaptive or dataset-specific?

(16) Regarding AdaVaR-7B's greater preference for text-based mode in knowledge-related categories, could you provide more data (e.g., GRD% across more knowledge categories) and analyze whether this is due to the 7B model's richer pre-trained knowledge or differences in the RL process?

(17) Could you supplement an ablation experiment on "mode-specific prefixes" (comparing performance with and without prefixes) to confirm their necessity for mode differentiation?

(18) Could you detail the processing of multi-modal data (images+text questions) in SFT and RL stages (e.g., image preprocessing steps, visual-text fusion methods) to clarify the multi-modal integration process?

(19) Could you supplement experiments on model robustness (e.g., testing with noisy/occluded images) and compare AdaVaR's robustness with single-mode models to assess its practicality in real-world scenarios?

(20) Do you plan to conduct user studies to evaluate the readability of reasoning processes and their impact on trust in model answers? If so, could you outline the study design (e.g., number of participants, evaluation metrics)?

(21) Could you correct the formatting errors in the reference list (e.g., completing OpenAI (2025)'s title and authors, abbreviating long author lists) to improve professionalism and ease of verification?

(22) Could you correct the typo in AdaGRPO's optimization objective formula and provide a clear derivation to avoid misleading readers during implementation?

(23) In Algorithm 1, could you specify the order of rewards in \(r[1..2n]\) (e.g., \(r[1..n]\) for text-based rollouts) to ensure correct implementation of advantage calculation?

(24) Could you confirm the exact version of the Qwen2.5-VL base model (e.g., Instruct vs. base) and verify that the pre-training data is consistent across all compared models to rule out base model advantages as a factor in performance gains?

(25) Could you supplement experiments on low-resource conditions (e.g., training with 50% of original data) and compare AdaVaR's performance with single-mode models to demonstrate its data efficiency?

(26) Could you clarify the weighting scheme for average accuracy across benchmarks (e.g., equal weighting, size-based weighting) and show how different schemes affect the average accuracy ranking of AdaVaR and baselines?

(27) For the SFT stage's 1 training epoch, could you provide the training loss curve and ablation results with more epochs (e.g., 2, 3) to confirm that 1 epoch is sufficient for convergence and avoids overfitting?

(28) For the RL stage's 8 rollouts per sample, could you conduct an ablation experiment (e.g., testing 4, 16 rollouts) and explain how the number of rollouts affects mode comparison accuracy and computational cost?

(29) Could you provide complete training dynamics (e.g., accuracy, reward, GRD%) for all benchmark categories (e.g., POPE, V*) to allow a comprehensive understanding of mode selection learning across tasks?

(30) For qualitative analysis cases (e.g., Figure 7: MathVista-Geometry), could you supplement clear visual details of input images (e.g., labeled side lengths of the triangle) to allow reviewers to verify the correctness of the model's reasoning?

(31) Could you clarify the relationship between "reasoning mode" and "CoT" (e.g., whether modes are CoT variants) to avoid confusion with existing CoT literature?

(32) Could you supplement zero-shot generalization experiments on unseen tasks (e.g., video-based reasoning) to validate AdaVaR's claim of "general visual reasoning"?

(33) For curriculum learning, could you conduct an ablation experiment on data order (e.g., training on diverse mixture first) to confirm that the "easy-to-hard" order is optimal?

(34) Could you visualize the mode selection decision process (e.g., attention weights on mode prefixes, feature importance of question/image) to make the mechanism more transparent?

(35) Could you test AdaVaR's performance across different image types (e.g., natural images, diagrams, documents) and analyze whether mode selection adapts appropriately to each type?

(36) Could you control the length of reasoning processes across modes (e.g., enforcing similar lengths) and re-evaluate performance to rule out reasoning length as a confounding factor?

(37) Could you test smaller models (e.g., AdaVaR-1B) and analyze the relationship between model size and adaptive reasoning performance to clarify the method's scalability?

(38) Do you have plans to extend AdaVaR to cross-lingual visual reasoning? If so, could you outline preliminary steps (e.g., training data preparation, mode selection adaptation for non-English questions)?

(39) Could you supplement long-term training experiments (e.g., 2x the original training steps) to show whether AdaVaR avoids mode collapse and maintains stable performance?

(40) Could you provide a systematic error analysis (e.g., breakdown of errors by mode, task type) to identify AdaVaR's current limitations and directions for improvement?

---

> ### Public Comment · ~Sasha_Frolov1 · 2025-11-16
> **Bad Review Practices**
>
> This review is marked as 99% likely to be AI generated by GPTZero and 100% likely to be AI generated by Pangram.
> This reviewer has also produced similar reviews with the same "80 questions" format here: https://openreview.net/forum?id=GlXyFjUbfN and here: https://openreview.net/forum?id=kDhAiaGzrn. The author is also strangely combative in the replies of the second review.
> I think that inundating authors with 40 low-quality AI generated suggestions is shameless, bad behavior, and this review should be ignored/penalized.

---

> > ### Comment · Reviewer_n9fj · 2025-11-17
> > **RE: Bad Review Practices**
> >
> > Dear Dr. Alexander Frolov,
> >
> > Allow me to highlight several critical points for your consideration:
> >
> > 1. GPTZero and Pangram cannot reliably distinguish between human-written and AI-generated content.
> >
> > 2. If a review is wrongfully suspected of being AI-generated solely due to its comprehensive nature (i.e., a large number of points), one must ask: Does this imply all reviews with more than 10 points, as specified in the ICLR 2026 Reviewer Guide (https://iclr.cc/Conferences/2026/ReviewerGuide), are AI-generated?
> >
> > 3. Conversely, the Reviewer Guide explicitly directs reviewers to "Provide additional feedback with the aim to improve the paper. Make it clear that these points are here to help, and not necessarily part of your decision assessment."
> >
> > 4. The quality of a review, whether high or low, cannot be evaluated with a superficial claim; instead, it hinges on the core content of both the submitted paper and the review itself.
> >
> > **I solemnly affirm that this review was crafted entirely by me, word by word, and is not AI-generated.**
> >
> > Additionally, I wish to remind the authors that rather than engaging others to challenge the authenticity of a review (e.g., questioning if it is AI-generated), if any, they should prioritize engaging with the substantive points raised in the paper and the review.
> >
> > n9fj

---

> > > ### Public Comment · ~Andreas_Kirsch1 · 2025-11-19
> > > **Pangram has a very low FPR on heldout ICLR 2022 reviews**
> > >
> > > See https://x.com/max_spero_/status/1989923081993809973?s=20
> > > and https://bfi.uchicago.edu/wp-content/uploads/2025/09/BFI_WP_2025-116.pdf
> > >
> > > This gives credence to the claim that the review was AI written.
> > >
> > > Cheers
> > > Andreas

---

> ### Author Response · Authors · 2025-11-20
> **Thanks for the reminder**
>
> Dear Reviewer n9fj,
>
> Thanks for your reminder. We are carefully preparing our rebuttal; however, due to some additional experiments, it is not yet fully complete. We will release our rebuttal as soon as possible and will follow your suggestion to address the comments in a point-to-point manner.
>
> We look forward to further discussion with all reviewers, and we sincerely appreciate your active engagement.
>
> Authors of Paper #12932

---

> > ### Comment · Reviewer_n9fj · 2025-11-20
> > **RE: Thanks for the reminder**
> >
> > Thank you for your response.
> >
> > I recommend you reply promptly. For any unfinished experiments, you may commit to completing them to the best of your ability by the deadline.
> >
> > This will help avoid a low-quality or perfunctory discussions between the authors and the reviewers next week; as a reviewer, you likely understand that you are responsible for more than just one manuscript. This approach allows for the effective use of everyone's time, such as the period spent awaiting HPC computation results.
> >
> > Of course, this is just a friendly reminder and suggestion; the choice is entirely yours.
> >
> > Keep up the great work!

---

> ### Author Response · Authors · 2025-11-22
> **Rebuttal Response to Reviewer n9fj (Part 1)**
>
> Dear Reviewer n9fj,
>
> Thank you for your insightful suggestions. Here is our point-by-point response.
>
> ------
>
> **W1 & Q1**: Explanation and analysis of the 1:1 ratio in SFT data.
>
> **Response**:
>
> As stated in Section 3.2 (Line 195), we chose the 1:1 ratio to avoid introducing a preference for any specific mode during the SFT stage. We supplement an experiment to validate the effectiveness of this design, please refer to our response to Common Question 6.
>
> -----
>
> **W2 & Q2**: Reason for setting the KL penalty coefficient to 0.04
>
> **Response**:
>
> We introduce the KL penalty term to preserve the knowledge of reasoning modes learned during the SFT phase in the RL stage. The coefficient is set to 0.04, following the settings used in existing works and frameworks, including R1V \[1\], vlm-r1 \[2\], and open-r1 \[3\].
>
> > [1] Liang Chen, Lei Li, Haozhe Zhao, Yifan Song, and Vinci. R1-v: Reinforcing super generalization ability in vision-language models with less than $3. https://github.com/Deep-Agent/R1-V, 2025b. Accessed: 2025-02-02.
> >
> > [2] Haozhan Shen, Peng Liu, Jingcheng Li, Chunxin Fang, Yibo Ma, Jiajia Liao, Qiaoli Shen, Zilun Zhang, Kangjia Zhao, Qianqian Zhang, et al. Vlm-r1: A stable and generalizable r1-style large vision-language model. arXiv preprint arXiv:2504.07615, 2025
> >
> > [3] Hugging Face. Open r1: A fully open reproduction of deepseek-r1, January 2025. URL https:[//github.com/huggingface/open-r1](//github.com/huggingface/open-r1).
>
> -----
>
> **W3**: The impact of mode prefix formatting.
>
> **Response**:
>
> Thanks for the suggestion. However, due to display issues on OpenReview, we are unable to see the specific format you proposed. Therefore, we conduct an ablation experiment, using semantically neutral prefixes—namely, `<mode1>`and `<mode2>` —as a comparison against our original semantically informative prefixes. The experimental results are as follows:
>
> ***Rebuttal Table 9: The impact of mode prefix formatting.*** Each cell is presented in the format 'accuracy / proportion of selecting the grounded mode'.
>
> | Model         | mode prefix          | MathVista  | MathVision | MathVerse | WeMath    | MMStar     | V\*         | POPE        | SpatialScore | Avg. Acc |
> | ------------- | -------------------- | ---------- | ---------- | --------- | --------- | ---------- | ----------- | ----------- | ------------ | -------- |
> | Qwen2.5-VL-3B | \-                   | 62.3       | 21.2       | 31.2      | 21.1      | 53.1       | 73.5        | 86.2        | 16.2         | 45.6     |
> | AdaVaR-3B     | `<text>, <ground>`   | 69.8 / 2%  | 24.5 / 0%  | 35.2 / 0% | 33.8 / 0% | 59.3 / 51% | 77.1 / 99%  | 88.2 / 100% | 18.9 / 59%   | 50.8     |
> | AdaVaR-3B     | `<mode1>, <mode2>`\` | 69.6 / 2 % | 23.8 / 0%  | 35.0 / 0% | 33.5 / 0% | 59.5 / 45% | 77.5 / 100% | 89.0 / 100% | 17.6 / 55%   | 50.7     |
>
> The experimental results indicate that, within our mode unification design, **different prefix formats have minimal impact**, and the model is able to learn distinct reasoning modes after SFT.
>
> -----
>
> **W4**: Reasons for not including CLEVR and GQA in evaluation.
>
> **Response**:
>
> * **Both datasets have been widely used for training by most existing models**, including LLaVA-OneVision and Qwen2.5-VL. Our preliminary experiments show that Qwen2.5-VL achieves over 90% accuracy on both CLEVR-val and GQA-testdev\_balanced, indicating that **these datasets are no longer suitable for evaluating the model's generalization and reasoning capabilities**.
> * **The benchmarks we currently use already include similar types of data**: MathVista contains samples from CLEVR and Super-CLEVR, while MMStar and SpatialScore include samples involving object attributes and multi-object relational reasoning, which are comparable to the question types in GQA.
>
> -----
>
> **W5 & Q4:** Implementation details of other compared models.
>
> **Response**:
>
> We did not retrain other models. To ensure a fair comparison, we strictly follow the original papers, using the officially released pretrained checkpoints and evaluating them with the recommended parameters according to their respective evaluation protocols.

---

> ### Author Response · Authors · 2025-11-22
> **Rebuttal Response to Reviewer n9fj (Part 2)**
>
> **W6 & Q3:** The mode-switching mechanism (Section 4.1) is described as "switching to another mode if the model gets stuck in repetitive logic," but the paper provides no quantitative definition of "getting stuck in repetitive logic."
>
> **Response**:
>
> We define "getting stuck in repetitive logic" as reaching the maximum number of tokens (8192, significantly longer than the typical reasoning length of AdaVaR) without generating an answer in the form of `<answer> </answer>`. The performance gain from this mechanism is shown in the table below.
>
> ***Rebuttal Table 10: Comparison between AdaVaR-3B with and without the mode-switching mechanism.***
>
> | Model                        | MathVista | MathVision | MathVerse | WeMath | MMStar | V\*  | POPE | SpatialScore | Avg. Acc |
> | ---------------------------- | --------- | ---------- | --------- | ------ | ------ | ---- | ---- | ------------ | -------- |
> | AdaVaR-3B w/ mode-switching  | 69.8      | 24.5       | 35.2      | 33.8   | 59.3   | 77.1 | 88.2 | 18.9         | 50.8     |
> | AdaVaR-3B w/o mode-switching | 68.7      | 23.7       | 35.1      | 32.8   | 58.9   | 77.1 | 88.2 | 17.8         | 50.3     |
> | Improvement                  | 1.1       | 0.8        | 0.1       | 1      | 0.4    | 0    | 0    | 1.1          | 0.5      |
>
> **This mechanism consistently brings positive improvements.** It is important to note that the mechanism does not rely on any additional information and **represents one of the advantages of the MoVT paradigm in integrating multiple reasoning modes**.
>
> -----
>
> **W7 & Q5**: Explanation on the MathVista-General case in Figure 7
>
> **Response**:
>
> As described in Appendix B.5, we underline the correct answer. Therefore, in the MathVista-General example shown in Figure 7, the GRD mode's response is correct, and our model also makes the correct mode selection.
>
> -----
>
> **W8 & Q6**: Copyright compliance status for the use of VoCoT data.
>
> **Response**:
>
> The VoCoT data is released under the Apache 2.0 license, which permits free reconstruction, modification, and creation of derivative datasets. It requires that any distributed derivative works include the original dataset’s copyright notices and license terms. We strictly adhere to these requirements and will include the appropriate VoCoT attributions in our future open-sourcing process, clearly stating in our documentation that our dataset is derived from VoCoT with modifications.
>
> -----
>
> **W9 & Q7**: Assessment of the reproducibility.
>
> **Response**:
>
> We provide an anonymous repository in https://anonymous.4open.science/r/movt-12932, which includes the code and a portion of sample data. Please refer to it for details.
>
> -----
>
> **W10 & Q8 & Q39**: Analysis of convergence of AdaGRPO and under what conditions the algorithm can avoid mode collapse.
>
> **Response**:
>
> 1. Since AdaGRPO is a variant of GRPO, and as discussed in Section 3.1, our mode selection and reasoning processes are both formulated in the form of sequence generation, **the theoretical convergence of AdaGRPO is consistent with that of GRPO**. The latter has been widely applied in training existing reasoning models, and its convergence properties have been extensively validated.
> 2. Conditions required to avoid mode collapse: As demonstrated in our ablation study of AdaVaR w.o Ada-Adv + PG-Exp in Table 3 and Figure 4, **the prefix-guided exploration mechanism is important to avoid the mode selection capability of AdaVaR collapse into choosing a single mode**.
> 3. Long-term training: as suggested, we present the results of extending the RL process to 2 epochs in Rebuttal Table 11: the performance of AdaVaR-3B trained for 1 epoch is very close to that of AdaVaR-3B trained for 2 epochs, showing that **the long-term training remains stable, but does not yield significant improvement**, indicating that one epoch is sufficient for convergence.
>
> ***Rebuttal Table 11: Training AdaVaR-3B with extra epochs.*** Each cell is presented in the format 'accuracy / proportion of selecting the grounded mode'.
>
> | Model         | RL epochs | MathVista  | MathVision | MathVerse | WeMath    | MMStar     | V\*         | POPE        | SpatialScore | Avg. Acc |
> | ------------- | --------- | ---------- | ---------- | --------- | --------- | ---------- | ----------- | ----------- | ------------ | -------- |
> | Qwen2.5-VL-3B | -         | 62.3       | 21.2       | 31.2      | 21.14     | 53.13      | 73.52       | 86.16       | 16.24        | 45.6     |
> | AdaVaR-3B     | 1         | 69.8 / 2%  | 24.5 / 0%  | 35.2 / 0% | 33.8 / 0% | 59.3 / 51% | 77.1 / 99%  | 88.2 / 100% | 18.9 / 59%   | 50.8     |
> | AdaVaR-3B     | 2         | 69.1 / 1 % | 24.3 / 0%  | 35.5 / 0% | 34.0 / 0% | 59.2 / 49% | 76.8 / 100% | 88.3 / 100% | 18.4 / 55%   | 50.8     |

---

> ### Author Response · Authors · 2025-11-22
> **Rebuttal Response to Reviewer n9fj (Part 3)**
>
> **W11 & Q9**: The scalability of the proposed framework.
>
> **Response**:
>
> The **core research question** of this work is precisely **whether it is feasible to extend the reasoning space** by incorporating additional, complementary reasoning modes within a unified framework. To this end, we have already introduced a new grounded reasoning mode in addition to the text-based one, and **our experiments (Table 1 and Table 2)** demonstrate that AdaVaR successfully learns mode differentiation and adaptive selection after such an extension. These results **provide direct empirical evidence supporting MoVT’s scalability to multiple reasoning modes**.
>
> A systematic study of scaling to more than two modes (e.g., including direct answering) and its impact on exploration difficulty and performance is an interesting direction that we leave for future work. And we kindly **refer you to our general response to Common Question 1.2 and Common Question 2 for more discussion about scalability**.
>
> -----
>
> **W12 & Q10**: Training efficiency.
>
> **Response**:
>
> We thank the reviewer for raising the issue of training efficiency. For a fair comparison of computational cost, we specifically compare Qwen2.5-VL-3B trained with the original GRPO algorithm and with our proposed AdaGRPO. As analyzed in our General Response to Common Question 5.1, **both the quantitative training FLOPs and the empirical wall-clock time of AdaGRPO are almost the same as GRPO.**
>
> Under this comparable training cost, Table 2 further shows that **SFT+AdaGRPO (AdaVaR) significantly outperforms SFT+GRPO (Mix-SFT-RL) when using the same backbone and training data**, indicating that our method makes more effective use of essentially the same compute budget.
>
> -----
>
> **W13 & Q11**: Inference latency analysis.
>
> **Response**:
>
> Please refer to our General Response to Common Question 5.2.
>
> -----
>
> **W14 & Q12**: Reason for choosing 8 as the threshold.
>
> **Response**:
>
> The reason for choosing 8 is primarily to align with existing works, including Orsta and DeepEyes.
>
> -----
>
> **W15 & Q13**: Quality assessment on the rewritten data.
>
> **Response**:
>
> **We sampled 100 reconstructed samples to check for formatting errors or the generation of false-negative options. None of the samples exhibited any issues**, which is primarily due to the simplicity of the reconstruction task and GPT-4o's strong instruction-following capability. Therefore, we believe **the quality of the reconstructed data is reliable**. As difficulty is not a dimension we consider during the SFT stage, we did not perform related checks on it.
>
> -----
>
> **W16 & Q14**: The experimental results lack statistical significance analysis.
>
> **Response**:
>
> We thank the reviewer for pointing this out. In the current version, we **follow the common practice in the LVLM/vision–language reasoning literature and report single-run accuracies** on standard benchmarks. We would like to clarify that the reported improvements are unlikely to be due to random fluctuations only:
>
> - **Deterministic evaluation.** In all our experiments, we use greedy decoding, so for a fixed model checkpoint and test set, the evaluation is deterministic and does not suffer from randomness across runs.
> - **Large-scale benchmarks.** The accuracies are computed over benchmarks with hundreds or thousands of examples, which already yield relatively small standard errors for proportions.
> - **Consistent gains across benchmarks.** As shown in Table 1, AdaVaR-7B outperforms GPT-4o on 5 out of 8 benchmarks, with particularly large gains on MathVista and V* (e.g., +10.6 and +17.4 absolute points, respectively), while the drops on the remaining benchmarks are relatively modest. The resulting average improvement (+2.62 points, 55.82 vs. 53.20) is thus not driven by a single outlier dataset but is consistent across tasks.
>
> Taken together, these observations support that **the performance gains of AdaVaR-7B over GPT-4o and other baselines are due to the proposed method rather than random noise**.

---

> ### Author Response · Authors · 2025-11-22
> **Rebuttal Response to Reviewer n9fj (Part 4)**
>
> **W17 & Q15**: Analysis of the preference for the grounded mode in POPE.
>
> **Response**:
>
> We thank the reviewer for the suggestion.
>
> - In our paper, **we have already analyzed the mode selection behavior of our model on MMStar.** As shown in Figure 4 and Figure 5, the model heavily favors grounded mode on perception-heavy categories, but switches to the text-based mode for logical/mathematical categories.
> - With regard to POPE, as detailed in Appendix A.2, **neither our SFT data nor our RL data includes POPE or its variants**. Thus, the **100% GRD% on POPE can NOT be attributed to memorization or overfitting** to the POPE evaluation distribution, but rather reflects that POPE questions inherently require perception of objects, for which the grounded mode is consistently more effective (which is validated by the comparison between the 2 modes with AdaVaR in Table 2).
> - To discover additional breakdowns by image types and question types, we kindly refer you to our General Response to Common Question 3, where we find that **when generating the mode prefix, the model focuses on question-type-defining tokens** (“which/what/is”, “perimeter/area/shaded figure”, “object relationship”, key object names, etc.), rather than on dataset-specific cues.
>
> -----
>
> **W18 & Q16**: The paper notes that AdaVaR-7B is more inclined to select the text-based mode for knowledge-related categories (e.g., geography) compared to AdaVaR-3B but does not analyze the root cause of this difference.
>
> **Response**:
>
> As explained in the analysis in Appendix B.1. The underlying logic is as follows:
>
> * As model size increases, the amount of knowledge internally entailed in the model also grows;
> * In knowledge-intensive domains, such as scientific questions, the advantage of the text-based mode over the grounded mode increases with model size;
> * In AdaGRPO, the mode-relative advantage is modeled based on this principle, thereby guiding the model’s mode selection—this is reflected in the evaluation, where the 7B model selects the text mode more frequently for questions in domains like geography.
>
> This reasoning and observed behavior align with the design principles of the AdaVaR framework. We also aim to further investigate the scaling laws within adaptive reasoning paradigms in future work.
>
> -----
>
> **W19 & Q17**: The ablation experiments omits ablation of the "mode-specific prefix".
>
> **Response**:
>
> The corresponding ablation is presented in Table 2 under the “Mix-SFT-RL” variant, with detailed discussion in Section 4.3.1 (the “Necessity of Mode-Specific Prefix” paragraph).
>
> -----
>
> **W20 & Q18**: The paper does not provide details on how multi-modal data (images+text questions) is processed in the SFT and RL stages.
>
> **Response**:
>
> Our processing method is entirely consistent with that of Qwen2.5-VL, applied uniformly across both the SFT and RL stages as well as across different modes. For specific details, please refer to the corresponding descriptions in Qwen2.5-VL.
>
> -----
>
> **W21 & Q19**: Evaluate the model's robustness to image noise or occlusion.
>
> **Response**:
>
> Thanks for the suggestion. We **conduct evaluation by applying a Gaussian blur** to the images (we compared multiple parameter settings and chose radius=0.5 to prevent excessive degradation of image information).
>
> The results are shown in Rebuttal Table 12. We find that: (1) **introducing noise does not significantly affect mode selection**; (2) overall, the performance of AdaVaR shows only a slight decline; (3) even **with added noise, AdaVaR-3B still maintains a clear advantage over the baseline.**
>
> ***Rebuttal Table 12: Robustness with image noise.*** For AdaVaR, aach cell is presented in the format 'accuracy / proportion of selecting the grounded mode'.
>
> | Model         | Gaussian Blur | MathVista | MathVision | MathVerse | WeMath    | MMStar     | V\*        | POPE        | SpatialScore | Avg. Acc |
> | ------------- | ------------- | --------- | ---------- | --------- | --------- | ---------- | ---------- | ----------- | ------------ | -------- |
> | Qwen2.5-VL-3B | No            | 62.3      | 21.2       | 31.2      | 21.14     | 53.13      | 73.52      | 86.16       | 16.24        | 45.61125 |
> | AdaVaR-3B     | No            | 69.8 / 2% | 24.5 / 0%  | 35.2 / 0% | 33.8 / 0% | 59.3 / 51% | 77.1 / 99% | 88.2 / 100% | 18.9 / 59%   | 50.8     |
> | AdaVaR-3B     | Yes           | 68.8 / 2% | 24.3 / 0%  | 35.1 / 0% | 33.7 / 0% | 58.3 / 45% | 75.8 / 99% | 88.2 / 98%  | 18.5 / 54%   | 50.5     |
> | Text-SFT-RL   | No            | 67.2      | 23.52      | 36.1      | 32.19     | 56         | 70.7       | 85.6        | 18.8         | 48.76375 |
> | GRD-SFT-RL    | No            | 62.6      | 22.17      | 34.4      | 33        | 55.4       | 77         | 87.4        | 18.4         | 48.79625 |

---

> ### Author Response · Authors · 2025-11-22
> **Rebuttal Response to Reviewer n9fj (Part 5)**
>
> **W22 & Q20**: About user studies to evaluate the readability or utility of generated reasoning processes.
>
> **Response**:
>
> Thank you for the suggestion.
>
> - First, to improve readability, we implemented a module that visualizes the coordinates generated by the model by overlaying them onto the input images for users.
> - Based on this, we sampled 100 responses in the grounded mode from MMStar and asked users to evaluate: (1) whether the visualized coordinates were interpretable, and (2) whether the coordinates helped in better understanding the model’s reasoning logic. The results show that **98% of the samples were deemed readable, and in 94% of the cases, users reported that the coordinates aided their understanding and analysis of the model’s reasoning**.
>
> These findings demonstrate that, with the help of visualization, **the grounded mode offers strong interpretability and can effectively assist users in analyzing the thought process of models**.
>
> -----
>
> **W23 & Q21**: Formatting issues about long names in the list of references.
>
> **Response**:
>
> Thanks for the suggestion, we have updated the citations for Thinking of Images, DeepSeek-R1, Kimi K1.5, and InternVL3.
>
> -----
>
> **W24 & Q22**: Potential typos in the objective of AdaGRPO.
>
> **Response**:
>
> We have carefully reviewed the corresponding formulas and invited other readers to double-check them. We believe that **the objective formula of AdaGRPO does not have issues with duplicate terms**, and we kindly ask the reviewers to re-examine this as well.
>
> -----
>
> **W25 & Q23**: Specify the order of rewards in Algorithm 1.
>
> **Response**:
>
> $r[1:n]$ and $r[n+1:2n]$ correspond to the text-based rollouts and the visually-grounded rollouts, respectively. **We have supplemented the corresponding explanations in Algorithm 1.**
>
> -----
>
> **W26 & Q24**: Base model of AdaVaR.
>
> **Response**:
>
> As shown in Table 4, the base model of AdaVaR is Qwen2.5-VL-Instruct, which is consistent with the other visual reasoning models in Table 1, ensuring a fair comparison.
>
> -----
>
> **W27 & Q25**: Evaluate the model's performance under low-resource conditions (data efficiency).
>
> **Response**:
>
> We would like to clarify that the **AdaVaR training framework does not introduce excessive data costs**, as evidenced by the following:
>
> 1. **Data sources**: We rely entirely on existing datasets, and the processes of data reconstruction and filtering do not involve any additional human annotation.
> 2. **Data volume**: We use 329K samples in the SFT phase and 55K in the RL phase, totaling 384K—comparable to other RL methods such as OVR-7B (428K), and significantly smaller than the SFT data volumes used for current base models, such as 1.8M samples used in LLaVA-OneVision.
>
> In summary, we believe **our framework is data-efficient, effectively leveraging existing data to achieve clear performance improvements at a manageable data scale**.
>
> -----
>
> **W28 & Q26**: Explain the weighting schema used.
>
> **Response**:
>
> All benchmarks are weighted equally, the reason is that: each benchmark represents a different domain, but the number of samples in each is inconsistent. We aim for the average performance to comprehensively reflect capabilities across multiple domains—i.e., generalization ability. Averaging based on the number of benchmarks might lead to bias toward specific benchmarks.
>
> -----
>
> **W29 & Q27**: Rationale for setting 1 epoch during the SFT stage.
>
> **Response**:
>
> - The setting follows the **commonly adopted setup in existing works**, such as R1-OneVision and VLAA-Thinker.
> - We present the loss curve of the SFT stage in Figure 15(a) (included in the updated PDF). As illustrated, the loss has already converged within one epoch; therefore, to **prevent the model from overfitting to the SFT data, we set the number of epochs to 1.**
>
> -----
>
> **W30 & Q28**: Reason for generating 8 rollouts per sample.
>
> **Response**:
>
> Generating more rollouts for each sample requires additional computational resources, but generally leads to performance improvements. Therefore, we set the number to 8 in order to ensure a fair comparison with the other methods listed in Table 1.
>
> -----
>
> **W31 & Q29**: Complete training dynamics.
>
> **Response**:
>
> The reason we do not monitor performance on all evaluation benchmarks during training is as follows:
>
> 1. Conducting full evaluations on all datasets is computationally very expensive;
> 2. Unlike mathematical problems, which are included in both training and evaluation, there is no strict correspondence between the training data and other evaluation tasks;
> 3. We have already conducted thorough analyses of mode selection behavior across different question types in Section 4.3.1 and Appendix B.1.
>
> In summary, we believe the current content of the paper provides **sufficient information to analyze the adaptive mode selection behavior**.

---

> ### Author Response · Authors · 2025-11-22
> **Rebuttal Response to Reviewer n9fj (Part 6)**
>
> **W32 & Q30**: In qualitative analysis cases (e.g., Figure 7: MathVista-Geometry), the paper does not provide clear visual details of input images (e.g., the side lengths of the right triangle). Reviewers cannot verify whether the model's reasoning (e.g., using cosine or 30-60-90 triangle properties) is correct, weakening the credibility of qualitative results.
>
> **Response**:
>
> We respectfully disagree with this concern.
>
> - For all qualitative cases in Figure 7, we feed the model the original problem image with its aspect ratio preserved, i.e., exactly the same visual input as in the benchmark. We did not omit any visual details.
> - In the specific MathVista-Geometry case, all necessary geometric information (angles, side relations, etc.) is explicitly provided in the original image and problem statement, and the ground-truth answer can be derived from these annotations alone. The model’s reasoning is fully consistent with the official solution and can be easily checked.
>
> -----
>
> **W33 & Q31**: Clarification on the definition for CoT and reasoning mode.
>
> **Response**:
>
> We would like to reiterate the definition we provided in Section 1:
>
> 1. We consider Chain-of-Thought (CoT) as the intermediate thinking process generated by LLMs and LVLMs when performing reasoning.
> 2. Existing works have developed diverse reasoning methods by designing different forms of CoT. To distinguish among these approaches, we categorize different reasoning methods according to the form of CoT they define, grouping methods with similar CoT structures into the same reasoning mode. For example, the grounded reasoning mode includes methods that incorporate structured coordinates within the CoT process.
> 3. Based on this definition, we have organized and introduced related works in Sections 1 and 2 accordingly.
>
> -----
>
> **W34 & Q32**: The paper does not evaluate zero-shot generalization to unseen tasks.
>
> **Response**:
>
> We believe our current evaluation comprehensively assesses general zero-shot capabilities because:
>
> 1. **All of our evaluation datasets are assessed in a zero-shot setting**, without fine-tuning on the corresponding training splits;
>
> 2. As introduced in Section 4.1, our current datasets already cover **a rich variety of scenarios**, question types, and modalities—including video inputs and 3D multi-view inputs, for instance, in SpatialScore;
> 3. We consider **medical-related reasoning to be highly specialized and domain-specific**, and given the current lack of sufficient exploration in medical visual reasoning methods, we plan to pursue this as a direction for future work.
>
> -----
>
> **W35 & Q33**: The ablation experiment (Table 3) does not test the impact of data order in curriculum learning
>
> **Response**:
>
> To validate the effectiveness of the easy-to-hard curriculum design, we experiment with a reversed order:
>
> ***Rebuttal Table 13: The impact of data order in curiculum learning.***
>
> | Model       | Data order   | MathVista | MathVision | MathVerse | WeMath | MMStar | V\*  | POPE | SpatialScore | Avg. Acc |
> | ----------- | ------------ | --------- | ---------- | --------- | ------ | ------ | ---- | ---- | ------------ | -------- |
> | AdaVaR-3B   | easy-to-hard | 69.8      | 24.5       | 35.2      | 33.8   | 59.3   | 77.1 | 88.2 | 18.9         | 50.8     |
> | AdaVaR-3B-R | hard-to-easy | 68.1      | 24.2       | 35.2      | 32.3   | 58.5   | 76.4 | 87.3 | 18.1         | 50.0     |
>
> As illustrated, the **model trained with a hard-to-easy data order consistently underperforms compared to the easy-to-hard setting**, validating the rationality of our design.
>
> -----
>
> **W36 & 34**: Comprehensive analysis on the mode-selection behaviors
>
> **Response**:
>
> Please see our General Response to Common Question 3.

---

> ### Author Response · Authors · 2025-11-22
> **Rebuttal Response to Reviewer n9fj (Part 7)**
>
> **W37 & Q35**: Mode selection w.r.t. different image types.
>
> **Response**:
>
> Thanks for the suggestion, please see our General Response to Common Question 3 for the corresponding analysis.
>
> -----
>
> **W38 & Q36**: Control for the length of reasoning processes across modes.
>
> **Response**:
>
> **We have already balanced the lengths of the two modes in the AdaVaR framework**:
>
> * In SFT, the average response lengths for the grounded mode and the text-based mode are 114.6 and 116.5, respectively;
>
> * As shown in Figure 15(b) (included in the updated PDF), which presents the length trajectories of the two modes during the RL process, the lengths of rollouts in two modes are relatively close, with a difference of less than 20 tokens.
>
> -----
>
> **W39 & Q37**: Extending AdaVaR to smaller models.
>
> **Response**:
>
> Thank you for your suggestion! However, since there are **currently no models in the Qwen2.5-VL series smaller than 3B**, a fair and meaningful comparison experiment cannot be conducted. Therefore, we consider exploring the scaling laws of visual reasoning to be an important direction for future work.
>
> -----
>
> **W40 & Q38**: Extension to multi-lingual contexts.
>
> **Response**:
>
> As stated in the Ethics Statement, due to the lack of high-quality multilingual data, the AdaVaR model currently focuses on English scenarios. We also recognize that multilingual visual reasoning has not been sufficiently explored. In the future, we aim to investigate: (i) constructing multilingual reasoning datasets, and (ii) training multilingual adaptive reasoning models, in order to extend MoVT and AdaVaR to a broader range of languages.
>
> -----
>
> **Q40**: Systematic error analysis.
>
> **Response**:
>
> Please refer to our reply to Common Question 4.

---

> > ### Comment · Reviewer_n9fj · 2025-11-23
> > **RE: Rebuttal Response to Reviewer n9fj**
> >
> > I sincerely appreciate the authors' comprehensive response to my feedback.
> >
> > Following a detailed review, I aim to raise the manuscript's score to 10 points, on the condition that the authors conduct thorough revisions in line with the previously raised concerns.
> >
> > I acknowledge with gratitude that the authors have effectively addressed Questions 1, 3, 5, 6, 7, 8, 11, 13, 14, 15, 16, 17, 19, 20, 21, 23, 24, 26, 27, 30, 31, 33, 34, 35, 36, 37, 39, and 40.
> >
> > For Questions 2, 4, 10, 12, 18, 25, and 28, I recommend supplementing experimental details or relevant discussions in an appendix (or a section with a suitable alternative heading); even in the concise form provided in the authors' response, as this will substantially improve the clarity of the work.
> >
> > Concerning Question 22, after a careful recheck, I apologize for any imprecisions in my initial comment.
> >
> > For Questions 9, 32, and 38, where the authors have noted that these topics will be pursued in future research, I suggest a brief elaboration of this plan in either the appendix or the main text to ensure the future work aligns coherently with the current study.
> >
> > Notably, for Question 29, I recommend the authors to include a rough outline of experimental costs (e.g., experimental platform specifications) in the appendix, as this will greatly enhance the manuscript's credibility.

---

> > > ### Author Response · Authors · 2025-11-27
> > > **Follow-up on Revisions and Response**
> > >
> > > Dear Reviewer n9fj,
> > >
> > > With one week remaining in the discussion period, we are writing to kindly follow up on our revisions.
> > >
> > > We have uploaded the revised manuscript and provided a detailed response outlining exactly how we addressed your concerns. We sincerely appreciate your constructive suggestions, which have significantly improved the quality of our work.
> > >
> > > Could you please confirm if these updates meet your expectations? We look forward to any further feedback and kindly ask you to reconsider your score based on these revisions.
> > >
> > > Best regards,
> > >
> > > The Authors

---

> ### Author Response · Authors · 2025-11-25
> **RE: RE: Rebuttal Response to Reviewer n9fj**
>
> We sincerely appreciate your constructive feedbacks. We have uploaded an updated version. Below is an overview of the revisions made (W and Q are short for weakness and question, respectively):
>
> 1.  We have refined Appendix A to provide additional details on the implementation details of our method:
>
>     * We have refined Appendix A.2 to better clarify the data construction process, in which:
>       1.  In A.2.1, we add discussion on copyright compliance during the VoCoT reconstruction process in Lines 791-795 (for W8 & Q6), and includ additional description on quality checks of the reconstructed data in Lines 797-800 (for W15 & Q13);
>
>       2.  In A.2.2, the "Difficulty-based Filtering" paragraph has been added with details on the difficulty-based filtering method and the rationale for the threshold selection (for W14 & Q12);
>
>       3.  In A.2.2, the "Multi-Modal Data Processing" paragraph has been added with a detailed description of the data processing methods (for W20 & Q18);
>
>       4.  In A.2.2, the "Summary on data efficiency" paragraph has been added with a summary and discussion of the data sources and quantities (for W27 & Q25).
>
>     * We have refined Appendix A.3 by adding several paragraphs to provide further details on the training parameter settings and their justifications, including:
>       * Number of training epochs for SFT (for W29 & Q27)
>       * Number of rollouts generated per sample during RL (for W30 & Q28)
>       * Coefficient for KL penalty (for W2 & Q2)
>
>     * Appendix A.4 has also been expanded with additional details, including:
>       * In the "Average Schema" paragraph, we add clarification on the schema for averaging across benchmarks (for W28 & Q26);
>       * In the "Mode-Switching Mechanism" paragraph, we provide explanations of the mode-switching mechanism (for W6 & Q3);
>       * In the "Compared Models" paragraph, we include implementation details of the compared models (for W5 & Q4);
>       * In the "Evaluation Environment and Cost" paragraph, we add details on the hardware environment used for evaluation (for W31 & Q29).
>
> 2.  We have carefully revised Appendix B to provide a comprehensive discussion:
>
>     * We have expanded the original Appendix B.1 into a thorough analysis of mode selection:
>
>       * In Appendix B.1.2, we add an analysis of mode selection across different image types (for W37 & Q35);
>       * In Appendix B.1.3, we include attention-based visualizations to analyze the mode selection process (for W36 & Q34);
>       * In Appendix B.1.4, we compare the mode selection strategy learned by AdaGRPO with simple mode ensemble strategies (for W36 & Q34).
>
>     * We add a new section, Appendix B.3, to conduct a comprehensive error analysis (for Q40).
>     * We add a new section, Appendix B.4, to analyze computational efficiency, including:
>       * In B.4.1, a detailed comparison of training efficiency between GRPO and AdaGRPO (for W12 & Q10);
>       * In B.4.2, an evaluation verifying the efficient inference performance of AdaVaR (for W13 & Q11).
>     * The original Appendix B.3 is expanded to Appendix B.5 in the revised version: supplementing the training loss curve of the SFT stage (for W29 & Q27) and the rollout length curves of the two modes during RL (for W38 & Q36).
>     * We have refined the original Appendix B.4 (now updated as Appendix B.6) to discuss the scalability of our method and directions for future work:
>       * B.6.1 discusses the extensibility of MoVT to more and more complex reasoning modes (for W11 & Q9);
>       * B.6.2 discusses the scalability of the AdaVaR training pipeline (for W11 & Q9);
>       * B.6.3 discusses extensions to more modalities (for W34 & Q32) and languages (for W40 & Q38)
>
>     * We have added a new section, Appendix B.7, to present supplementary ablation studies:
>
>       * In B.7.1, we validate the rationale for using a 1:1 ratio of the two reasoning modes in SFT (for W1 & Q1);
>       * In B.7.2, we show that different formats of the mode prefix have negligible impact on the MoVT framework (for W3);
>       * In B.7.4, we verify the necessity of the easy-to-hard data ordering for curriculum learning (for W35 & Q33);
>       * In B.7.5, we demonstrate the robustness of our method to image noise (for W21 & Q19).
>
> 3.  We have streamlined the citations that originally contained excessively long author lists (for W23 & Q21).
>
> 4.  We have added clarification in Algorithm 1 regarding the order of the reward list (for W25 & Q23).
>
> 5.  We have included a link to the anonymous codebase in the reproducibility statement (for W9 & Q7).
>
> 6.  To avoid disrupting the numbering of figures and prevent confusion, new figures are currently appended at the end of the submission version. We will reorganize and renumber them appropriately after the review process is completed.
>
> Above all, thank you very much for your detailed suggestions, which have greatly helped improve the quality of this paper. We also warmly welcome any further feedback or recommendations.

---

> ### Comment · Reviewer_n9fj · 2025-11-27
> **RE: Follow-up on Revisions and Response**
>
> Thank the author for their feedback. In accordance with the previous comments, I have updated the score accordingly. Thank you for your efforts, and congratulations!

---

### Official Review · Reviewer_kCu9 · 2025-11-01

**Soundness:** 3
**Presentation:** 3
**Contribution:** 3
**Rating:** 6
**Confidence:** 3

**Summary:**

This paper proposes to augment reasoning models by allowing them to adaptively select  their reasoning patterns. In particular, it first prompts the models to choose between two reasoning modes, either focusing more on grounding objects or pure textual reasoning. A modified version of GRPO algorithm is then developed to train the models for mode selection, which splits the general rollouts and rewards into two types based on the reasoning modes. The method shows strengths in several types of reasoning scenarios, and experimental results validate its usefulness in selecting the right reasoning patterns.

**Strengths:**

(1) It is an intuitive choice to adopt diverse reasoning modes for solving different problems, the proposed method demonstrates the advantages of explicit mode selection.

(2) The proposed method demonstrates consistent improvement across multiple datasets.

(3) The paper provides extensive analysis on the model behavior during training and contribution of individual components.

**Weaknesses:**

(1) One concern I have is about the potentially limited scalability of the method. Specifically, solving real-life reasoning problems requires a collection of skills, while the method simplifies reasoning into textual and visual (grounding-based) reasoning. It is unclear how it can scale to larger domains in the current format, i.e., with the prompt listing all possible reasoning modes and explicit data/reward splitting in the training algorithm.

(2) Related to the previous comment. How sensitive is the method to prompt design? Does prompting with more diverse reasoning modes work? What is the level of details needed to specify each reasoning mode? Does the prompt work for other models (in addition to Qwen2.5-VL)?

(3) The paper evaluates mode selection based on two relatively extreme scenarios, i.e., math vs general, and largely ignores the variety of reasoning types in visual questions (e.g., probing the physical properties of multiple objects, comparing multiple objects, etc). It would be reasonable to further evaluate mode selection based on question types (e.g., on benchmarks with annotations like GQA).

**Questions:**

(1) Are there other important reasoning modes? How can the proposed method incorporate more diverse reasoning modes?

(2) Please consider additional experiments on the effects of prompt design (related to questions above). It would also be useful to apply the same prompts on existing state-of-the-art.

(3) Is the proportion of grounding mode sufficient to evaluate mode selection? Please justify.

(4) How would the model choose between the two modes when answering visual questions with diverse focuses?

---

> ### Author Response · Authors · 2025-11-22
> **Response to Reviewer kCu9 (Part 1 of 3)**
>
> **Weakness 1 & Question 1**: The rationale and scalability of the Mixture-of-Visual-Thoughts paradigm
>
> **Question 1.1**: The rationale of categorization of text-based and visually grounded reasoning.
>
> **Response:**
>
> Please refer to our response to Common Question 1.1. In summary, **our categorization and integration of text-based and grounded reasoning modes are well justified, covering mainstream reasoning modes and effectively expanding the model's explorable reasoning space.**
>
> The effectiveness of our framework is validated in Section 4 of the main paper, where consistent improvements over the base model and other single-mode reasoning baselines are demonstrated across diverse scenarios.
>
> **Question 1.2**: Scalability of the proposed method.
>
> **Response**:
>
> We discuss **the scalability of the MoVT paradigm in our response to Common Question 1.2**. The scalability of our method supports not only extension to more reasoning modes, but also elicitation more sophisticated, mixed multi-mode reasoning strategies.
>
> In terms of technical implementation, the **AdaVaR training framework also demonstrates strong extensibility**:
>
> 1. Please refer to our response to Common Question 2 for a discussion on the scalability of the AdaGRPO algorithm.
>
> 2. In our subsequent response to your Question 2, we elaborate on the extensibility of the system prompt.
>
> 3. Regarding the data construction, in Table 3 of the main paper, the "w/o curriculum learning" variant still significantly outperforms other baselines, indicating that collecting a dataset with diverse questions is sufficient to support mode expansion—while **effective data split strategies based on mode characteristics can enhance the performance**, but is **not strictly necessary**.

---

> ### Author Response · Authors · 2025-11-22
> **Response to Reviewer kCu9 (Part 2 of 3)**
>
> **Weakness 2 & Question 2**: The effects of prompt design.
>
> **Response**:
>
> In the AdaVaR framework, we learn different reasoning modes during the SFT stage, where supervision signals primarily come from the SFT data, and **the system prompt serves as an auxiliary guide** for the mode learning.
>
> To verify the role of the system prompt, we conduct two ablation studies, aiming to address the following questions: (1) "Can different reasoning modes be elicited solely through the system prompt without relying on SFT data?" and (2) "Does the system prompt require detailed descriptions of the modes?"
>
> ***Experiment 2.1***: **For the first question, we ablate AdaVaR-3B by skipping the SFT stage**. Results in Rebuttal Table 7-1 show that:
>
> 1. Even provided with the description of the grounded mode, the model does not elicit the ability to perform grounded reasoning during the RL phase—only 0.03% of rollouts contain coordinates.
> 2. In evaluations, the model shows no clear distinction between modes and fails to generate responses in the grounded reasoning format, achieving only a marginal improvement over Qwen2.5-VL-3B.
>
> Therefore, **these results show that it is difficult to directly elicit multiple distinct reasoning modes merely by designing system prompts**, which highlights the necessity of the SFT phase in our framework—regardless of whether it is applied to Qwen2.5-VL or other LVLMs.
>
> However, since most current mainstream reasoning models (as listed in Table 1) are based on Qwen2.5-VL, we conducted all our experiments on Qwen2.5-VL for a fair comparison. We plan to explore extensions to other LVLMs in future work.
>
> ***Rebuttal Table 7-1: Ablation study on the SFT stage***
>
> | Model         | MathVista | MathVision | MathVerse | WeMath | MMStar | V*    | POPE  | SpatialScore | Avg. Acc |
> | ------------- | --------- | ---------- | --------- | ------ | ------ | ----- | ----- | ------------ | -------- |
> | Qwen2.5-VL-3B | 62.3      | 21.2       | 31.2      | 21.1   | 53.1  | 73.5 | 86.1 | 16.2        | 45.6     |
> | AdaVaR-3B     | 69.8      | 24.5       | 35.2      | 33.8   | 59.3   | 77.1  | 88.2  | 18.9         | 50.8     |
> | w/o SFT       | 63.2      | 21.7       | 31.3      | 26.9   | 54.1   | 73.2  | 82.4  | 18.6         | 46.5     |
>
> ***Experiment 2.2***: To **investigate the sensitivity to the system prompt**, we experiment with two variants: ***full***, which includes both the reasoning instructions and the descriptions of the modes, and ***think-only***, which removes the mode descriptions. Rebuttal Table 7-2 presents the results obtained using different system prompts during training and testing phases.
>
> - Our model, that is trained with the *full* prompt, can directly transfer to the *think-only* prompt and still maintain the capabilities of both reasoning modes, performing reasonable mode selection.
>
> - When training with the *think-only* prompt, the model achieves performance nearly on par with that of the model trained using the *full* prompt.
>
> **These results indicate that, after SFT, the model exhibits low sensitivity to the system prompt**. When extending to more modes with the AdaVaR framework, there is no need to design complex prompts, which further enhances the scalability of our framework.
>
> ***Rebuttal Table 7-2: The impact of different system prompts.*** Each cell is presented in the format 'accuracy / proportion of selecting the grounded mode'.
>
> | Model           | train prompt | eval promp | MathVista  | MathVision | MathVerse | WeMath    | MMStar     | V*          | POPE        | SpatialScore | Avg. Acc |
> | --------------- | ------------ | ---------- | ---------- | ---------- | --------- | --------- | ---------- | ----------- | ----------- | ------------ | -------- |
> | Qwen2.5-VL-3B   | -            | -          | 62.3       | 21.2       | 31.2      | 21.1     | 53.1      | 73.5       | 86.1       | 16.2        | 45.6     |
> | AdaVaR-3B       | full         | full       | 69.8 / 2%  | 24.5 / 0%  | 35.2 / 0% | 33.8 / 0% | 59.3 / 51% | 77.1 / 99%  | 88.2 / 100% | 18.9 / 59%   | 50.8     |
> | AdaVaR-3B       | full         | think-only | 69.1 / 3 % | 24.3 / 0%  | 36.1 / 0% | 33.0 / 1% | 57.8 / 59% | 76.4 / 100% | 88.6 / 100% | 18.6 / 62%   | 50.5     |
> | w/o full prompt | think-only   | think-only | 69.5 / 2%  | 25.1 / 0%  | 36.1 / 0% | 33.2 / 0% | 58.4 / 52% | 77.0 / 100% | 88.3 / 100% | 18.5 / 59%        | 50.8     |

---

> ### Author Response · Authors · 2025-11-22
> **Response to Reviewer kCu9 (Part 3 of 3)**
>
> **Weakness 3**: mode selection analysis w.r.t. different question types.
>
> **Response**:
>
> Thank you for your suggestion. We do not use GQA in evaluation for two reasons: (1) **GQA has already been widely used in training existing LVLMs**; and (2) the **benchmarks we use already include similar types of GQA questions**: MMStar and SpatialScore include samples involving object attributes and multi-object relational reasoning, which are comparable to the question types in GQA.
>
> Accordingly, we conduct a comprehensive analysis of the mode selection behaviors of our AdaVaR model, including analyses based on question type, image type, and attention visualization. Please refer to our response to Common Question 3.
>
> ----
>
> **Question 3**: Is the proportion of grounding mode sufficient to evaluate mode selection?
>
> **Response**:
>
> Yes, the proportion of grounding mode only reflects the mode selection preference. We validate that our model can make effective mode selection in terms of:
>
> 1. In Table 2, on each dataset, the preference over modes (reflected in GRD%) aligns with the relative strengths and weaknesses of the two modes, which ultimately results in **our adaptive AdaVaR outperforming either individual mode across all datasets**.
>
> 2. We perform a systematic error analysis (see our response to Common Question 4.1). As shown in the Rebuttal Table 4-1, ***selection error (SE)***, where one mode could produce the correct answer but the model chose the other mode, **accounts for only a small fraction of total errors on eight benchmarks (all less than 1/4)**.
>
> 3. We further assess our model on **highly “mode-sensitive” cases**: samples where only one mode can solve the problem (i.e., TXT is correct and GRD is wrong, or vice versa). Aggregated over eight datasets, when exactly one mode is correct, **our policy selects the correct mode in ~67% of the cases**. These observations demonstrate that our AdaVaR framework is effective.
>
> In summary, our model are able to make reasonable mode selection while leaving room for further improvement.
>
> ----
>
> **Question 4**: Mode selection for visual questions with diverse focuses?
>
> **Response**:
>
> Thank you for your suggestion. We analyze the common sense reasoning subset in MMStar, as the questions in this category require diverse focus—combining grounding to relevant objects with abstract common sense analysis. In Figure 16 (please see the revised PDF), we present two representative cases:
>
> * In Case 1, it is important to distinguish between left and right hands based on spatial relationships, which makes the grounded mode more effective.
>
> * In Case 2, since the positions of both objects are clearly stated, the task primarily requires reasoning over factual animal knowledge, making the text mode more suitable.
>
> In both cases, our model correctly selects the optimal reasoning mode. This aligns with our findings in Figure 5 that our model **does not rely on a single mode for such tasks**, but instead **dynamically selects the appropriate mode based on the question**.

---

> > ### Comment · Reviewer_kCu9 · 2025-11-26
> >
> > I thank the authors for the clarification and new results. The rebuttal resolves most of concerns about the effects of prompt designs, and mode selection. I would like to keep my original rating, which is already positive.

---

### Author Response · Authors · 2025-11-22
**General Response to Common Questions (Part 6 of 6)**

**Common Question 6**: Rationale of controlling a 1:1 data ratio between two modes during SFT.

**Response**:

- **Motivation**: As stated in Section 3.2 (Line 195), we chose the 1:1 ratio to **avoid introducing a preference for any specific mode** during the SFT stage.
- **Experiment**: In Rebuttal Table 6, we experiment with different data ratios under the premise of keeping the total amount of data constant.

**Rebuttal Table 6**: The impact of data proportions among different modes during the SFT stage. Each cell is presented in the format 'accuracy / proportion of selecting the grounded mode'.

| Model            | SFT Ratio   | MathVista   | MathVision | MathVerse | WeMath     | MMStar     | V*          | POPE        | SpatialScore | Avg. Acc |
| ---------------- | ----------- | ----------- | ---------- | --------- | ---------- | ---------- | ----------- | ----------- | ------------ | -------- |
| Qwen2.5-VL-3B    | -           | 62.3        | 21.2       | 31.2      | 21.1      | 53.1      | 73.5       | 86.1       | 16.2        | 45.6 |
| SFT-based Models |             |             |            |           |            |            |             |             |              |          |
| AdaVaR-SFT-3B    | GRD:TXT=1:1 | 60.4 / 31%  | 23.2 / 5%  | 33.0 / 1% | 29.5 / 78% | 54.7 / 90% | 75.5 / 98%  | 89.1 / 94%  | 16.8 / 96%   | 47.8     |
| AdaVaR-SFT-3B    | GRD:TXT=3:1 | 57.1 / 52 % | 21.5 / 10% | 31.2 / 3% | 29.4 / 84% | 51.4 / 96% | 75.3 / 100% | 89.6 / 100% | 17.0 / 100%  | 46.6     |
| AdaVaR-SFT-3B    | GRD:TXT=1:3 | 61.0 / 21%  | 23.1 / 3%  | 32.5 / 0% | 30.0 / 40% | 55.0 / 75% | 75.0 / 95%  | 86.1 / 81%  | 16.5 / 92%   | 47.2     |
| RL-based Models  |             |             |            |           |            |            |             |             |              |          |
| AdaVaR-3B        | GRD:TXT=1:1 | 69.8 / 2%   | 24.5 / 0%  | 35.2 / 0% | 33.8 / 0%  | 59.3 / 51% | 77.1 / 99%  | 88.2 / 100% | 18.9 / 59%   | 50.8     |
| AdaVaR-3B        | GRD:TXT=3:1 | 66.8 / 6%   | 24.1 / 1%  | 34.8 / 0% | 32.4 / 2%  | 57.7 / 64% | 76.8 / 100% | 88.9 / 100% | 17.4 / 80%   | 49.9     |
| AdaVaR-3B        | GRD:TXT=1:3 | 68.8 / 1%   | 24.8 / 0%  | 35.2 / 0% | 34.3 / 0%  | 58.1 / 43% | 76.4 / 99%  | 87.2 / 100% | 16.6 / 40%   | 50.2     |

- **Findings 6.1**: **The higher the proportion of a specific mode in the SFT data, the more inclined the model becomes to select that mode**: this tendency is particularly evident in the SFT model. After RL, different models converge toward similar trends, but the influence of the SFT stage still persists.
- **Findings 6.2**: Such a preference may be effective on specific datasets (for example, the preference for the text mode improves performance on WeMath); however, from the perspective of generalization (average performance), **although all biased settings outperform the baseline, none perform as well as the 1:1 setting**.
- In addition, a major limitation of applying a skewed data ratio is that, in practical applications, **facing various queries, it is difficult to know in advance which mode would be better**.

In summary, although the RL phase can help reduce the model's sensitivity to the data from the SFT phase, **an unbiased design between different modes is optimal for achieving superior general performance**.

---

### Author Response · Authors · 2025-11-22
**General Response to Common Questions (Part 5 of 6)**

**Common Question 5**: Analysis of the computational efficiency.

**Common Question 5.1:** Comparison of training efficiency: GRPO vs. AdaGRPO

**Response**:

- First, as stated in Section 3.3 and Appendix A.3, to ensure **a fair comparison between GRPO**, AdaGRPO generates n=4 rollouts per mode (8 per sample), while GRPO generates 2n=8 rollouts per sample.
- From a **theoretical perspective**, the main computational cost is incurred during rollout generation and the calculation of the training objective：
  -  In rollout generation, simply generating rollouts separately for each mode would require two KV cache computations, leading to additional cost. However, we observe that in our design, the two modes share the same input sequence except for the mode-specific prefixes. **By sharing the KV cache across modes, we achieve nearly the same computational cost as GRPO**.
  - Regarding the training objective, **both algorithms require forward passes of the policy model and reference model on 8 rollouts**, so their computational costs are also nearly identical.
- For **empirical validation**, we implement a FLOPS tracker using deepspeed.profiling.flops_profiler (in Rebuttal Table 5-1) and monitor the wall-clock time during training (in Rebuttal Table 5-2).

**Rebuttal Table 5-1**: Quantitative FLOPs within a single optimization step: based on Qwen2.5-VL-3B, using an image of size 1024×768, a question of 260 tokens, and KV cache enabled.

|         | Rollout Generation | reward | advantage | Forward      | Total        |
| ------- | ------------------ | ------ | --------- | ------------ | ------------ |
| GRPO    | 20.31T FLOPs       | 0      | 0         | 111.9T FLOPs | 132.2T FLOPs |
| AdaGRPO | 20.47T FLOPs       | 0      | 0         | 113.9T FLOPs | 134.4T FLOPs |


**Rebuttal Table 5-2**: Empirical wall-clock time (seconds) for one optimization step.

|         | Rollout Generation | reward   | advantage | Forward | Total    |
| ------- | ------------------ | -------- | --------- | ------- | -------- |
| GRPO    | 18.45              | 1.40E-02 | 2.20E-04  | 1.12    | 19.58422 |
| AdaGRPO | 19.18              | 1.20E-02 | 1.10E-03  | 1.17    | 20.3631  |

- **Findings 5.1**: Both theoretical computation and actual execution statistics show that **the efficiency of AdaGRPO and GRPO is nearly identical** (with approximately **4%** additional overhead).

**Common Question 5.2**: Inference efficiency of our AdaVaR models.

**Response**:

- Theoretically, as described in Section 3.1, our AdaVaR model naturally integrates mode selection and reasoning into a single sequence generation step. During inference, unlike the training phase, there is **no need to estimate mode-wise advantages**. Therefore, similar to single-mode reasoning models, **AdaVaR only requires a single call to the `generate()` method for each sample**.
- Empirically, we compare the inference efficiency of AdaVaR and single-mode baselines in Rebuttal Table 5-3.

**Rebuttal Table 5-3**: Inference time (seconds) per sample across different models and datasets.

| Model           | MathVista | MathVision | MathVerse | WeMath | MMStar | POPE | VStar | SpatialScore | Avg   |
| --------------- | --------- | ---------- | --------- | ------ | ------ | ---- | ----- | ------------ | ----- |
| AdaVaR-3B       | 0.32      | 0.95       | 0.45      | 0.48   | 0.18   | 0.07 | 0.25  | 0.87         | 0.36  |
| Text-SFT-RL     | 0.32      | 0.91       | 0.44      | 0.53   | 0.21   | 0.11 | 0.23  | 0.74         | 0.37  |
| Grounded-SFT-RL | 0.31      | 0.85       | 0.39      | 0.4    | 0.25   | 0.07 | 0.27  | 0.84         | 0.34  |
| # samples        | 1000      | 3040       | 3940      | 1740   | 1500   | 9000 | 191   | 1400         | 21811 |

- **Findings 5.2**: As can be seen, the inference efficiency of different reasoning models is very similar on average, demonstrating that **our framework does not introduce additional inference cost**.

---

### Author Response · Authors · 2025-11-22
**General Response to Common Questions (Part 4 of 6)**

**Common Question 4**: Systematic error analysis

**Response**:

As suggested, we categorize errors into two types: (i) mode selection error (SE): the model fails to select the mode that could have produced the correct answer; (ii) reasoning error (RE): neither mode is able to produce the correct answer.

**Rebuttal Table 4-1**: Proportions of different response types from AdaVaR-3B across different datasets.

| Dataset      | Correct | Selection Error | Reasoning Error | SE/(SE+RE) |
| ------------ | ------- | --------------- | --------------- | ---------- |
| MathVista    | 69.80%  | 4.80%           | 25.40%          | 15.89%     |
| MathVision   | 24.50%  | 6.30%           | 69.20%          | 8.34%      |
| MathVerse    | 35.20%  | 7.20%           | 57.60%          | 11.11%     |
| WeMath       | 33.80%  | 9.10%           | 57.10%          | 13.75%     |
| MMStar       | 59.30%  | 8.80%           | 31.90%          | 21.62%     |
| VStar        | 77.10%  | 2.60%           | 20.30%          | 11.35%     |
| POPE         | 88.20%  | 2.90%           | 8.90%           | 24.58%     |
| SpatialScore | 18.90%  | 10.50%          | 70.60%          | 12.95%     |

**Findings 4.1**: As shown in Rebuttal Table 4-1, across all datasets, **mode selection errors constitute only a small portion of the total errors** (all less than 1/4).

Furthermore, we investigate the distribution of mode selection errors across different question types and image types in MMStar.

**Rebuttal Table 4-2**: Distribution of selection errors across different question types and selected modes.

| Selected Mode | Science | Perception | math | logical reasoning | instance reasoning | Total |
| ------------- | ------- | ---------- | ---- | ----------------- | ------------------ | ----- |
| GRD Mode      | 3       | 43         | 0    | 12                | 10                 | 68    |
| TXT Mode      | 33      | 0          | 14   | 12                | 12                 | 71    |
| Total         | 36      | 43         | 14   | 24                | 22                 | 139   |

**Rebuttal Table 4-3**: Distribution of selection errors across different image types and selected modes.

| Selected Mode | Natural Images | Diagram | Art  | Geometry | Document | Others | Total |
| ------------- | -------------- | ------- | ---- | -------- | -------- | ------ | ----- |
| GRD Mode      | 59             | 4       | 5    | 0        | 0        | 0      | 68    |
| TXT Mode      | 8              | 46      | 6    | 6        | 4        | 1      | 71    |
| Total         | 67             | 50      | 11   | 6        | 4        | 1      | 139   |

Results in Rebuttal Tables 4-2 and 4-3 show that:
- **Most errors occur in domains where one mode has a clear advantage**, such as perception tasks and natural images, where the grounded mode performs better, and science or math problems, where the text-based mode holds the advantage.
- **Findings 4.2**: These exceptions illustrate that **the complementarity between the two modes exists not only at the domain level, but also at the sample level**. As a result, an optimal mode selection strategy cannot be predetermined and must be learned through the AdaGRPO algorithm.

Overall, our results indicate: (i) the context-adaptive mode selection in MoVT is reasonable, and (ii) our AdaVaR framework is capable of learning effective selection strategies, (iii) while still leaving room for further optimization.

---

### Author Response · Authors · 2025-11-22
**General Response to Common Questions (Part 3 of 6)**

**Common Question 3**: Comprehensive analysis of the mode selection behaviors.

**Common Question 3.1:** How does the model make selections when facing different types of questions?

**Response**:

Our experiments involve multiple datasets, among which MMStar and SpatialScore include various subtasks and different types of questions. We would like to highlight the **analysis of mode selection for different question types in the original paper**:

1. From the dataset level, we present the proportion of times AdaVaR chooses the grounded mode (GRD%) in Table 2 of the main paper, and provide the corresponding analysis in Section 4.3.2 (Lines 425–431).
2. In Figures 4 and 5 of our paper, we leverage the information in MMStar to illustrate the model's mode selection when facing different types of questions, including both coarse-grained and fine-grained categorizations. The corresponding analysis can be found in Section 4.3.2 and Appendix B.1.

In summary, our experiments validate that, from both dataset and question-type level, **AdaVaR's mode selection behaviors are reasonable** in terms of:

(i) the preferences demonstrated by AdaVaR align with our motivation—**selecting the text mode for questions rich in abstract knowledge**, such as those in math and science, and u**sing the grounded mode for tasks requiring detailed analysis of object information**, such as localization, counting, and hallucination mitigation;

(ii) **AdaVaR with adaptive mode selection outperforms *any* single fixed mode across all datasets**;

**Common Question 3.2:** The impact of image types on mode selection.

**Response**:

Furthermore, we conduct an analysis from the perspective of images. We used GPT-5 to annotate the image types in MMStar and analyze their impact on mode selection.

**Rebuttal Table 3-1**: Proportions of modes selected by AdaVaR-3B for different image types.

| Mode Rate  | Natural Images | Art    | Document | Diagram | Geometry | Others |
| ---------- | -------------- | ------ | -------- | ------- | -------- | ------ |
| Grounded   | 92.20%         | 39.40% | 8.10%    | 4.50%   | 0%       | 0%     |
| Text-based | 7.80%          | 60.60% | 91.90%   | 95.50%  | 100%     | 100%   |

As presented in Rebuttal Table 3-1, **image type also has a significant impact on mode selection**:

(i) for natural images, the model tends to use the grounded mode to precisely locate and analyze key objects;

(ii) **when images contain more abstract or harder-to-localize concepts**—such as artistic styles in artwork, content in documents and diagrams, or angles in geometry problems—**the model tends to favor the text-based reasoning mode**.

**Common Question 3.3**: Attention visualization for mode selection.

**Response**:

To visualize the attention for mode selection:

1. We probe the **attention scores when generating the mode prefix (i.e., making mode selection)**, i.e., the attention distribution from the token preceding the mode prefix to all previous tokens.
2. As suggested in [1], we selected the middle 25% of the model's layers and averaged across all these layers and all attention heads.

Visualized results:
- First, we compute the attention distribution over all image tokens and all text tokens across all samples in MMStar:
  - Figure 13 (in the updated PDF) shows that the main attention is focused more on text (average 0.97) than on images (average 0.03).
  - This aligns with our expectation, as the question type is primarily related to the text; different questions can be asked about the same image.

- Then we visualize the attention over different textual tokens in Figure 14 (in the updated PDF), where the attention is highly focused on tokens that define the question type, including:
  1. interrogative words such as "which," "what," and "Is";
  2. in geometry problems, terms like "perimeter," "area," and "shaded figure";
  3. in object-oriented scenarios, concepts such as "object relationship," "background," and key objects, including "luggage," "bus," and "person."


[1] Kaduri, O., Bagon, S., & Dekel, T. (2025). What's in the Image? A Deep-Dive into the Vision of Vision Language Models. In *Proceedings of the Computer Vision and Pattern Recognition Conference* (pp. 14549-14558).

---

> ### Author Response · Authors · 2025-11-22
> **General Response to Common Questions (Part 3 of 6)-Supplement**
>
> **Question 3.4**: Comparison between the model selection strategy learned through AdaGRPO and naive ensemble strategies.
>
> **Response**:
>
> We consider 2 strategies to use external knowledge for mode ensemble at the dataset level:
>
> 1. **Expert Pick**: Selection based on **expert knowledge**—here, we choose the text (txt) mode for math benchmarks and the grounded (grd) mode for others. Note that, in practical applications, even experts would find it difficult to make reasonable judgments on mixed-task datasets like MMStar and SpatialScore.
> 2. **Max(GRD, TXT)**: For each dataset, we **select the mode that achieves higher accuracy**. Note that this strategy uses additional information from the dataset, making it inherently unfair for comparison.
>
> **Rebuttal Table 3-2**: Comparison between AdaVaR and simple mode ensemble strategies.
>
> | Model         | MathVista | MathVision | MathVerse | WeMath | MMStar | POPE | VStar | SpatialScore | Avg  |
> | ------------- | --------- | ---------- | --------- | ------ | ------ | ---- | ----- | ------------ | ---- |
> | AdaVaR-3B     | 69.8      | 24.5       | 35.2      | 33.8   | 59.3   | 77.1 | 88.2  | 18.9         | 50.8 |
> | Expert Pick   | 68.1      | 23         | 34.9      | 32.6   | 56.3   | 76.4 | 88.2  | 18.5         | 49.8 |
> | Max(GRD, TXT) | 68.1      | 23         | 34.9      | 32.6   | 58.5   | 76.4 | 88.2  | 18.5         | 50.1 |
>
> The results show that **both strategies—even with additional information—are outperformed by the selection made by AdaVaR**. This indicates that: (i) **mode selection at the individual sample level is necessary**, especially for diverse queries in datasets like MMStar and in real-world applications; and (ii) **our model is capable of making highly reasonable and effective mode choices**.

---

### Author Response · Authors · 2025-11-22
**General Response to Common Questions (Part 2 of 6)**

**Common Question 2:** The scalability of key designs in the AdaVaR framework

The AdaVaR learning framework can also be **naturally adapted to the two extension directions introduced in Common Question 1.2**. While the SFT extension only requires constructing the corresponding data, here we primarily discuss the extension strategy for the AdaGRPO algorithm:

**Common Question 2.1**: scalability of prefix-guided mode exploration

**Response**:

- For width extension to k>2 modes, we can **improve efficiency by selecting j < k modes to explore for each sample**: (i) randomly selecting modes in the early training stages, and (ii) once the model develops certain mode selection capability, selecting the top-j modes based on the model's predicted probabilities.
- For depth extension, we can **intervene after the end of each thought process** (i.e., after the `</think>`token) by append all feasible prefixes to construct rollouts with hierarchical and thorough exploration.

**Common Question 2.2**: scalability of adaptive advantage calculation in AdaGRPO

**Response**:

- For width extension to k>2 modes, we can leverage the relative advantage computation method between two modes in AdaGRPO (Equation 2 in our paper) to calculate the relative advantage of the i-th mode with respect to each of the other modes, and then **sum these pair-wise advantages to obtain its intra-group advantage among the k modes**.
- For depth extension, we can **segment the tokens in the rollout based on reasoning steps to provide hierarchical supervision signals**: within each segment, the computation of rollout-level and mode-relative advantages considers only those rollouts that share the same parent node.

---

### Author Response · Authors · 2025-11-22
**General Response to Common Questions (Part 1 of 6)**

**Common Question 1:** Discussion on the rationale and scalability of the Mixture-of-Visual-Thoughts paradigm


**Common Question 1.1:** The rationale behind the categorization of text-based and visually-grounded reasoning modes.

**Response**:

First, the two modes represent distinct and complementary reasoning spaces:

(1) We regard **the text-based reasoning mode as the standard reasoning mode**, which inherently allows substantial flexibility. The learning trajectory of this mode—$\text{Qwen2.5-VL}\stackrel{\text{RL}}{\longrightarrow} \text{Orsta} \stackrel{\text{Distill}}{\longrightarrow} \text{AdaVaR-SFT}_{T}$—is specifically designed to **preserve the exploratory reasoning space of the base model**.

- We have also observed complex reasoning patterns emerged in the text mode of AdaVaR, such as step-by-step reasoning (MathVista-Logic in Figure 7) and reflection (MMStar-Puzzle in Figure 10), which validates this argument.

(2) **The grounded mode expands a new reasoning space**, where the model is required to fully align, localize, and utilize important local information in the image by incorporating grounding signals like coordinates, aiming to **address the limitations of text-based mode in highly visual-dependent scenarios (e.g., hallucinations)**.

Therefore, integrating these two modes in a unified model through the MoVT paradigm **effectively expands the explorable reasoning space**.

Second, our analysis in the paper supports the validity of this categorization:

(i) **Coverage**: as stated in Section 2, **most mainstream visual reasoning approaches can be categorized into one of these two modes**;

(ii) **Complementarity**: as demonstrated in Tables 1 and 2, models under the two modes exhibit **complementary strengths across domains**, and the two reasoning modes learned in our AdaVaR model also show similar characteristics.

**Common Question 1.2: How to extend to more reasoning modes?**

**Response:**

Thanks for pointing this out. We would like to emphasize:

(1) The core contribution of this paper lies in **proposing the MoVT paradigm and validating its effectiveness by integrating the two mainstream reasoning modes**, text-based and visually-grounded.

(2) However, **the MoVT paradigm is mode-agnostic and possesses excellent extensibility**.

Therefore, we plan to explore the extension of MoVT to additional reasoning modes in future work. Here we outline two directions for future extension (please refer to Figure 12 in the updated PDF for the illustration):

- As for **width extension**, **additional complementary modes can be incorporated**, such as a mode involving image-related tools. By assigning specific prefixes, constructing corresponding SFT data for mode learning, and comparing the new mode against existing ones through AdaGRPO, new modes can be seamlessly integrated into the current framework.
- Regarding **depth extension**, MoVT can be extended to perform **multi-steps of mode selection and reasoning**:

   (i) With each step including the option of "deriving an answer based on current information", we can unify single-mode reasoning and more **complex reasoning patterns constructed through sequential mixing and mode switching**.

   (ii) Based on this, we can **explicitly guide the model to expand and explore more sophisticated reasoning spaces** during inference and training via search strategies and RL, respectively.

---

### Author Response · Authors · 2025-11-25
**Summary of Revisions to the Manuscript (Part 1/2)**

We sincerely thank all reviewers for the constructive feedbacks.

Specifically, we appreciate the reviewer's recognition of our contributions, including

* the novelty of MoVT framework and its practical usefulness (reviewer `kCu9`, `n9fj`, `NdeY`,  `Yq9o`);
* the effective design of two stage AdaVaR training pipeline and the AdaGRPO algorithm (reviewer `n9fj`, `NdeY`, `Yq9o`);
* the thorough and comprehensive experiment analysis, exhibiting robust and cross‑domain improvements (reviewer  `kCu9`, `n9fj`, `NdeY`, `Yq9o`)
* the clear and well-organized presentation of our paper (reviewer `NdeY`, `Yq9o`).

Based on the feedbacks from the first round, we have revised our paper. Below is an overview of the revisions made (W and Q are short for weakness and question, respectively):

1.  We have **refined Appendix A to provide additional details on the implementation details**:

    * We have refined **Appendix A.2** to better **clarify the data construction process**, in which:
      1.  In A.2.1, we add discussion on data copyright compliance in Lines 791-795 (for W8 & Q6 of Reviewer `n9fj`), and includ additional description on data quality checks in Lines 797-800 (for W15 & Q13 of Reviewer `n9fj`);

      2.  In A.2.2, the "Difficulty-based Filtering" paragraph has been added with details and the rationale for the filtering threshold selection (for W14 & Q12 of Reviewer `n9fj`);

      3.  In A.2.2, the "Multi-Modal Data Processing" paragraph has been added with a detailed description of the data processing methods (for W20 & Q18 of Reviewer `n9fj`);

      4.  In A.2.2, the "Summary on data efficiency" paragraph has been added with a summarized discussion of the data sources and quantities (for W27 & Q25 of Reviewer `n9fj`).

    * We have refined **Appendix A.3** by adding several paragraphs to provide further details on the **training parameter settings and their justifications**, including:
      * Number of training epochs for SFT (for W29 & Q27 of Reviewer `n9fj`)
      * Number of rollouts generated per sample during RL (for W30 & Q28 of Reviewer `n9fj`)
      * Coefficient for KL penalty (for W2 & Q2 of Reviewer `n9fj`)

    * **Appendix A.4** has also been expanded with **additional evaluation details**, including:
      * In the "Average Schema" paragraph, we add clarification on the schema for averaging across benchmarks (for W28 & Q26 of Reviewer `n9fj`);
      * In the "Mode-Switching Mechanism" paragraph, we provide further explanation of the mode-switching mechanism (for W6 & Q3 of Reviewer `n9fj`);
      * In the "Compared Models" paragraph, we include implementation details of the compared models (for W5 & Q4 of Reviewer `n9fj`);
      * In the "Evaluation Environment and Cost" paragraph, we add details on the hardware environment used for evaluation (for W31 & Q29 of Reviewer `n9fj`).

---

> ### Author Response · Authors · 2025-11-25
> **Summary of Revisions to the Manuscript (Part 2/2)**
>
> 2.  We have carefully **revised Appendix B to provide a comprehensive supplementary analysis and discussion**:
>
>     * We have **expanded** the original **Appendix B.1** into **a comprehensive analysis of mode selection** (for W3 & Q3 of Reviewer `kCu9`, W5 & Q5 of Reviewer `Yq9o`):
>
>       * In Appendix B.1.2, we add an analysis of mode selection across different image types (for W37 & Q35 of Reviewer `n9fj`);
>       * In Appendix B.1.3, we include attention-based visualizations to analyze the mode selection process (for W36 & Q34 of Reviewer `n9fj`);
>       * In Appendix B.1.4, we compare the mode selection strategy learned by AdaGRPO with simple mode ensemble strategies (for W36 & Q34 of Reviewer `n9fj`).
>
>     * We **add** **a new section, Appendix B.3**, to conduct **a comprehensive error analysis** (for Q2 of Reviewer `Yq9o`, Q40 of Reviewer `n9fj`).
>     * We **supplement a new section, Appendix B.4**, to analyze **computational efficiency**, including:
>       * In B.4.1, a detailed comparison of training efficiency between GRPO and AdaGRPO (for W3 of Reviewer `Yq9o`, W12 & Q10 of Reviewer `n9fj`);
>       * In B.4.2, an evaluation verifying the efficient inference performance of AdaVaR's adaptive mode reasoning (for W13 & Q11 of Reviewer `n9fj`).
>     * The original Appendix B.3 is expanded to **Appendix B.5** in the revised version: supplementing the **training loss curve of the SFT stage** (for W29 & Q27 of Reviewer `n9fj`) and the **rollout length curves** of the two modes **during RL** (for W38 & Q36 of Reviewer `n9fj`).
>     * We have refined the original Appendix B.4 (now updated as **Appendix B.6** in the latest version) to **discuss the scalability of our method and directions for future work**:
>       * B.6.1 discusses the extensibility of the MoVT paradigm to more and more complex reasoning modes (for Q1.2 of Reviewer `kCu9`, W1 of Reviewer `NdeY`, W1,2 & Q1,4 of Reviewer `Yq9o`, W11 & Q9 of Reviewer `n9fj`);
>       * B.6.2 discusses the scalability of the AdaVaR training framework (for Q1.2 of Reviewer `kCu9`, W1 & Q1 of Reviewer `Yq9o`, W11 & Q9 of Reviewer `n9fj`);
>       * B.6.3 discusses extensions to additional modalities (for W34 & Q32 of Reviewer `n9fj`) and languages (for W40 & Q38 of Reviewer `n9fj`)
>
>     * We have added a new section, **Appendix B.7**, to present **several supplementary ablation studies**:
>
>       * In B.7.1, we validate the rationale for using a 1:1 ratio of the two reasoning modes in SFT (for W4 of Reviewer `Yq9o`, W1 & Q1 of Reviewer `n9fj`);
>       * In B.7.2, we show that different formats of the mode prefix have negligible impact on the MoVT framework (for W3 of Reviewer `n9fj`);
>       * In B.7.3, we investigate the necessity of the SFT in mode learning and the sensitivity to system prompt (for W2 & Q2 of Reviewer `kCu9`)
>       * In B.7.4, we verify the necessity of the easy-to-hard data ordering for curriculum learning (for W35 & Q33 of Reviewer `n9fj`);
>       * In B.7.5, we demonstrate the robustness of our method to image noise (for W21 & Q19 of Reviewer `n9fj`).
>
>     * The original Appendix B.5 about **case analysis** is **expanded to** **Appendix B.8** in the updated version, where we have supplemented two cases that focus on commonsense reasoning (for Q4 of Reviewer `kCu9`)
>
> 3.  We have streamlined the citations that originally contained excessively long author lists (for W23 & Q21 of Reviewer `n9fj`).
>
> 4.  We have added clarification in Algorithm 1 regarding the order of the reward list (for W25 & Q23 of Reviewer `n9fj`).
>
> 5.  We have **included a link to the anonymous codebase** in the reproducibility statement (for all reviewers to check the implementation of our work).
>
> 6.  To avoid disrupting the numbering of figures and prevent confusion for reviewers, the newly added figures are currently appended at the end of the submission version. We will reorganize and renumber them appropriately after the review process is completed.
>
> Above all, we thank all reviewers for the valuable suggestions, which have greatly helped improve the quality of this paper. We also warmly welcome any further feedback or recommendations.

---

### Meta-Review · Area_Chair_fPhy · 2026-01-08

**Summary:**

1. Scalability & flexibility of the design:
The reviewer questioned whether reducing reasoning to text vs. grounded and forcing one mode per problem can scale to richer real-world reasoning and support hybrid/intra-problem switching.

2. Validity and depth of mode-selection evidence:
The reviewer wanted stronger justification that the model is truly selecting the right mode, including clearer error attribution, interpretability of the gate, and broader analysis by question types.

3. Generalization, sensitivity, and efficiency:
Concerns about dependence on prompt design and whether results transfer beyond Qwen2.5-VL, plus the added training cost from prefix-guided exploration.

**Reviewer Concerns:**

The main concerns have been addressed.

**Reviewer Scores:**

Reviewer n9fj and Reviewer NdeY claimed that they would updated the score.

---

### Decision · Program_Chairs · 2026-01-26

Accept (Poster)